# Atlas of Fshr expression from novel reporter mice

Hongqian Chen[1†], Hui-Qing Fang[1,2†], Jin-Tao Liu[1], Shi-Yu Chang[1], Li-Ben Cheng[1], Ming-Xin Sun[1], Jian-Rui Feng[1], Ze-Min Liu[1,3], Yong-Hong Zhang[1,3], Clifford J Rosen[4], Peng Liu[1]*

[1]Laboratory of Bone and Adipose Biology, Shanxi Medical University, Taiyuan, China; [2]Department of Dentistry, The 980th Hospital of the PLA Joint Logistic Support Force, Shijiazhuang, China; [3]Shanxi Medical Universityersity, The Second Hospital, University Shanxi Medical University, Taiyuan, China; [4]Maine Medical Center Research Institute, Scarborough, United States

**eLife assessment**

These **valuable** findings develop a mouse model with trackable fusion Fshr protein, which will be of use to the field. The animal model helps to elucidate the expression and function of the FSH receptor in extra-gonadal tissues. The strength of the evidence is **solid** in most parts, although additional validation of the localization data would strengthen the study considerably.

**\*For correspondence:**
liupossible@gmail.com

[†]These authors contributed equally to this work

**Abstract** The FSH-FSHR pathway has been considered an essential regulator in reproductive development and fertility. But there has been emerging evidence of FSHR expression in extragonadal organs. This poses new questions and long-term debates regarding the physiological role of the FSH-FSHR, and underscores the need for reliable, in vivo analysis of FSHR expression in animal models. However, conventional methods have proven insufficient for examining FSHR expression due to several limitations. To address this challenge, we developed Fshr-ZsGreen reporter mice under the control of Fshr endogenous promoter using CRISPR-Cas9. With this novel genetic tool, we provide a reliable readout of Fshr expression at single-cell resolution level in vivo and in real time. Reporter animals were also subjected to additional analyses,to define the accurate expression profile of FSHR in gonadal and extragonadal organs/tissues. Our compelling results not only demonstrated Fshr expression in intragonadal tissues but also, strikingly, unveiled notably increased expression in Leydig cells, osteoblast lineage cells, endothelial cells in vascular structures, and epithelial cells in bronchi of the lung and renal tubes. The genetic decoding of the widespread pattern of Fshr expression highlights its physiological relevance beyond reproduction and fertility, and opens new avenues for therapeutic options for age-related disorders of the bones, lungs, kidneys, and hearts, among other tissues. Exploiting the power of the Fshr knockin reporter animals, this report provides the first comprehensive genetic record of the spatial distribution of FSHR expression, correcting a long-term misconception about Fshr expression and offering prospects for extensive exploration of FSH-FSHR biology.

## Introduction

Follicle-stimulating hormone (FSH), secreted by anterior pituitary gonadotrophs, is recognized as a crucial regulator of male and female gonadal function; indeed, reproductive biology textbooks strongly emphasize its role in normal physiology (*Simoni et al., 1997*). FSH exerts its effects via a specific receptor, follicle-stimulating hormone receptor (FSHR), which belongs to the highly conserved

family of G-protein-coupled receptors (*Chrusciel et al., 2019*). Traditional views hold that in females, FSHR is expressed in granulosa cells and controls the maturation of Graafian follicles, granulosa cell proliferation, and estrogen production (*Simoni et al., 1997*), while in males, FSHR is expressed in testicular Sertoli cells and regulates their metabolic functions, which are essential for proper spermatogenesis and germ cell survival (*Asatiani et al., 2002*). The large FSHR gene is located on chromosome 2p21 and comprises 10 exons (*Minegishi et al., 1991*).

However, accumulating evidence demonstrates that FSHR is also expressed in extragonadal tissues, such as endothelium (*Stilley and Segaloff, 2018*), monocytes (*Robinson et al., 2010*), developing placenta (*Stilley et al., 2014*), endometrium (*Sacchi et al., 2018*), malignant tissues (*Papadimitriou et al., 2016*), bone, adipose, and neural cells (*Bhartiya and Patel, 2021*). Intriguingly, blocking the interaction of Fsh with Fshr mitigates some degenerative disorders in mice, such as low bone mass (*Sun et al., 2006*), obesity (*Liu et al., 2017*), and neurocognitive decline (*Xiong et al., 2022*). Fshr was reported to be expressed in osteoclasts in vitro, and global Fsh or Fshr knockout resulted in an increase in bone mass by inhibiting bone resorption (*Sun et al., 2006*). Blocking the Fsh-Fshr interaction with either Fsh polyclonal or monoclonal specific antibodies triggered thermogenesis in adipose tissues, significantly reduced body weights (*Liu et al., 2017*), and reduced Alzheimer's symptoms in mice (*Xiong et al., 2022*). Fshr expression is also found in the vasculature of tissues (*Stilley et al., 2014*) and is particularly high in solid tumors (*Ghinea, 2018*). Nevertheless, the relevance of its extragonadal expression and functions have been debated (*Chrusciel et al., 2019*; *Kumar, 2014*). Therefore, precisely defining the localization of *Fshr* expression remains an imperative challenge in FSH-FSHR biology due to the concern of the 'non-specificity' of available antibodies used to localize FSHR expression and the 'quick turnover' and 'rapid degradation' of Fshr transcripts (*Chrusciel et al., 2019*; *Bhartiya and Patel, 2021*; *Tedjawirja et al., 2023*). Thus, a suitable genetic approach to resolve this issue is warranted (*Kumar, 2018*).

To address this challenge, we can utilize a GFP reporter driven by the endogenous promoter of *Fshr*, which enables the visualization of the expression of low-abundance transcripts in more accurate and context-specific ways (*Chudakov et al., 2005*; *Cheng et al., 1996*), when other approaches, e.g., antibodies and RT-PCR, are limited in their ability to detect *Fshr* expression. Therefore, in this study, we employed CRISPR/Cas9-mediated technology to create a novel Fshr-ZsGreen reporter mouse model for precisely clarifying *Fshr* expression. We believe that this powerful approach, as a reliable readout of *Fshr* expression at the single-cell level, should allow us to accurately capture *Fshr* expression in real time in a tissue-specific manner. We also used other techniques to ensure the accuracy and reliability of the results. Our results challenge the current understanding of *Fshr* expression and refine that Fshr has a wider expression profile than previously thought. The findings of Fshr in reproductive and nonreproductive tissues/organs will provide significant insights into new roles of Fsh and Fshr in physiology and pathology and have implications for the development of new therapies for reproductive and nonreproductive disorders, particularly metabolic diseases and degenerative diseases.

## Results

To precisely define Fshr expression in mice, we utilized several complementary strategies, including *Fshr* endogenous promoter driving ZsGreen knockin reporter mice, immunofluorescence (IF) staining with antibodies against tissue/cell type-specific markers and Fshr, droplet digital RT-PCR (ddRT-PCR) and single RNA-fluorescence in situ hybridization (RNA-smFISH), to comprehensively examine Fshr expression.

### Generation of Fshr-ZsGreen knockin reporter mice

CRISPR/Cas9-mediated homologous recombination was used to generate embryonic stem cell clones, in which a P2A-ZsGreen cassette was precisely inserted before the stop codon of the Fshr gene followed by the 3' UTR of the *Fshr* allele, as described in the Methods (*Figure 1A*). This P2A-ZsGreen expression cassette under the control of endogenous *Fshr* regulatory elements ultimately generates ZsGreen proteins without disrupting *Fshr* expression. The 19 amino acids P2A sequences function to cause ribosomal 'skipping' during translation, resulting in a missing peptide bond and effectively separates the two proteins, e.g., Fshr and ZsGreen (*Zhu et al., 2023*; *Trichas et al., 2008*; *Luke and Ryan, 2018*). The injected zygotes were transferred into oviducts of Kunming pseudopregnant females to

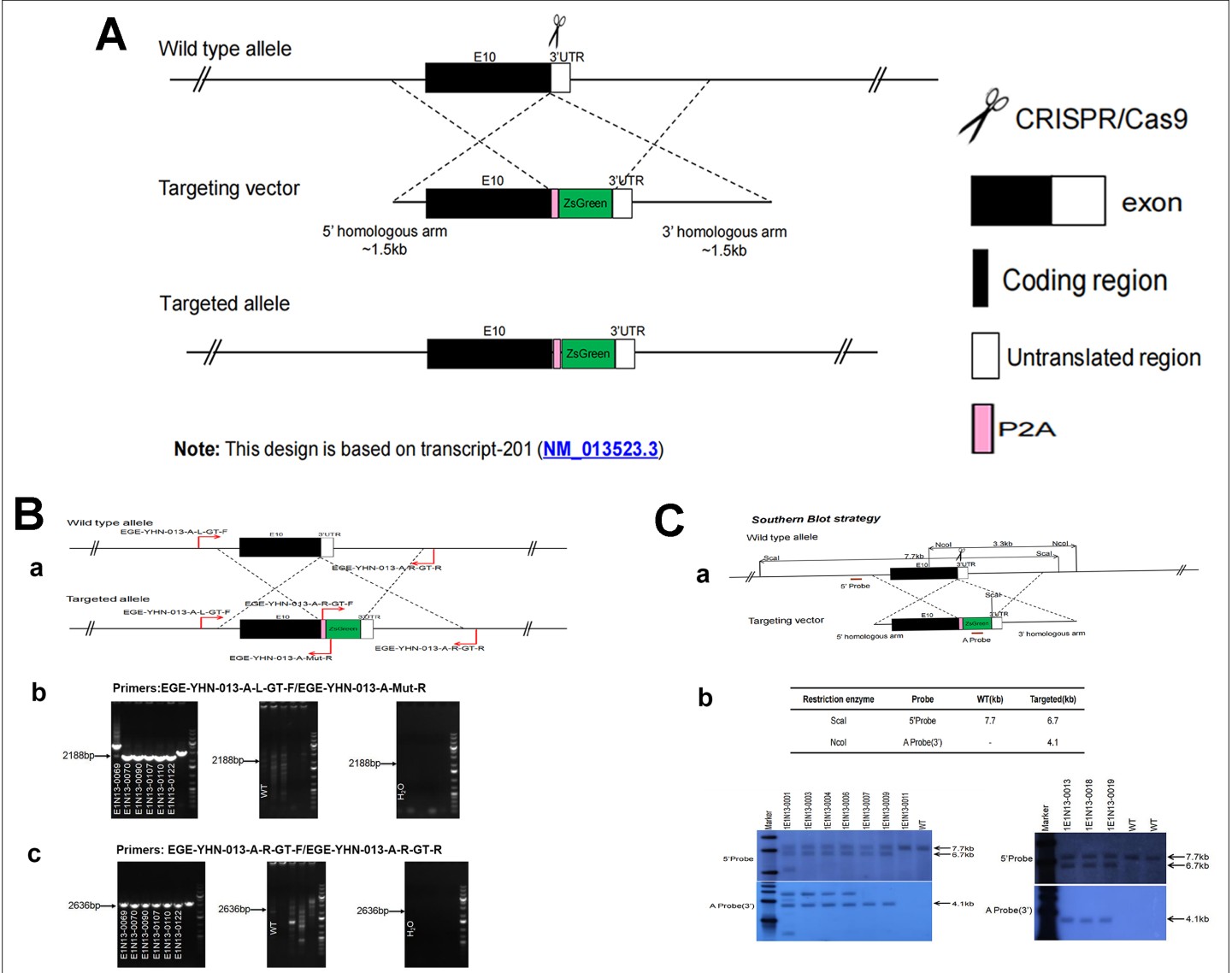

**Figure 1.** Generation of CRISPR/Cas9-mediated Fshr-ZsGreen knockin reporter mice. (**A**) CRISPR/Cas9-mediated targeting strategy to generate Fshr-P2A-ZsGreen knockin mice. (**B**) Detection of integration by PCR in F0 and F1 mice: (**a**) Schematic of PCR primer design specific to Fshr-P2A-ZsGreen and the wild-type allele. The results of genomic DNA PCR genotypes using the primer pair EGE-YHN-013-A-L-GT-F/EGE-YHN-013-A-Mut-R (**b**) and another primer pair EGE-YHN-013-A-R-GT-F/EGE-YHN-013-A-R-GT-R (**c**). (**C**) Southern blot confirmation of the correct integration of the P2A-ZsGreen allele in F0 and F1 mice. The Southern blot results demonstrated the successful generation of the targeted P2A-Fshr-ZsGreen allele: (**a**) Restriction sites in the wild-type sequence and targeted vector; (**b**) Southern blot analysis probes, expected restriction fragment lengths as indicated and blotted images.

The online version of this article includes the following source data for figure 1:

**Source data 1.** Gel images for genotyping PCR of mouse tail DNA.

**Source data 2.** Original TIFF images for Southern blots.

generate F0 mice. F0 mice with the expected genotype were confirmed by tail genomic DNA PCR, DNA sequencing, and Southern blotting (*Figure 1B and C*) and then mated with C57BL/6J mice to establish F1 heterozygous mice with the germline-transmitted transgene. F1 heterozygous mice were further genotyped by tail genomic PCR, DNA sequencing and Southern blotting (*Figure 1B and C*). The results from these tail genomic PCR, DNA sequencing, and Southern blots in both F0 and F1 pups demonstrated that the targeted P2A-ZsGreen cassette was accurately inserted into the designed site between exon 10 and the stop codon of the *Fshr* gene before the 3' UTR. We maintained heterogeneous Fshr-ZsGreen mice and used homogeneous mice for experiments. These mice were genotyped

using primer sets specific to P2A-ZsGreen, as shown in *Figure 1D*. Fshr and GFP are transcribed in a bicistronic mRNA but subsequently translated into two independent proteins rather than as a fusion protein. This design ensures unaffected Gαi3 transcription and function and simultaneous GFP expression as a reporter protein. All animals were fertile and showed normal behavior and no obvious abnormal phenotypes.

## Examination of Fshr expression in Fshr-ZsGreen reporter mice

With confirmation that the *Fshr*-P2A-ZsGreen targeting vector was successfully inserted into the *Fshr* locus, we investigated Fshr expression by confocal microscopy to detect ZsGreen expression and immunostaining for tissue/cell markers in frozen sections of tissues/organs of Fshr-ZsGreen reporter mice. To ensure that there was no nonspecific expression of Fshr-ZsGreen in the examined tissues/ organs, we took frozen sections derived from wild-type mice (B6) as negative controls. The negative controls were imaged under the conditions used for examining Fshr-ZsGreen expression. The representative results are shown in *Appendix 1—figure 1*, showing no nonspecific expression of Fshr-ZsGreen in the negative controls. On this basis, we performed the following imaging to examine Fshr-ZsGreen expression in the major organs and tissues.

### Reproductive organs

As the reproductive system is well known to express *Fshr*, we first tested Fshr-ZsGreen expression in the ovary and testis to ensure ZsGreen expression driven by the endogenous *Fshr* promoter. In the ovary, we observed Fshr-ZsGreen expression in the different stages of follicles from primordial follicles to primary follicles, secondary follicles, and the corpus luteum (*Figure 2A-b and e*). In the ovarian/Graafian follicles, expression was observed in the oocytes, granulosa cells/follicle cells and theca (interna and externa) (*Figure 2A-b, c, e, and f*). We also found Fshr-ZsGreen expression in the ciliated epithelial cells in the oviduct (*Figure 2A-d and h*). Furthermore, we employed an antibody against Stra8 (*Miyauchi et al., 2017*) to perform IF staining to identify reproductive cells and observed the colocalization of Fshr-ZsGreen with Stra8 staining as a marker for reproductive cells (*Figure 2A*).

In the testis, we found Fshr-ZsGreen expression and its colocalization with Stra8 (*Heinrich et al., 2020*) staining in the cells of seminiferous tubules (STs), including primary spermatocytes, Sertoli cells, and spermatids, and particularly in interstitial Leydig cells between STs (*Figure 2B*). *Figure 2B-a* shows an image of the whole sectioned testis. A representative ST is displayed at two magnifications (*Figure 2B-b and c*), demonstrating strong expression of Fshr-ZsGreen in Leydig cells, as indicated by empty white arrows. In addition, we also applied an antibody against Set to identify testis cells, whose expression was reported in multiple cell types of the mouse testis at different developmental stages (*Dai et al., 2014*). In *Figure 2C*, a representative image of ST with Leydig cells is shown at a lower magnification (×400) and a higher magnification (×1000) (*Figure 2C-a and b*), showing colocalization of Fshr-ZsGreen and Set staining in testis cells. We found strong Set staining in Fshr-ZsGreen-positive spermatogonia, as indicated by empty white arrowheads (*Figure 2C-b*). Fshr-ZsGreen was also observed in the arterioles of the testis (*Figure 2C-c*).

Because of RNA in situ hybridization as the gold standard method for visualizing RNA expression and localization in cells, tissue sections, and whole organs (*Young et al., 2020*; *Levsky and Singer, 2003*), we further carried out single RNA molecule-fluorescence in situ hybridization to confirm Fshr expression in the Leydig cells, using antisense probe for identifying *Fshr* expression in the sections of Fshr-ZsGreen and B6 mice, whereas its sense probe was used as a negative control (*Figure 2C and D*). Additionally, we also examined *Fshr* expression by IF in Leydig cell line TM3 (*Mather, 1980*) and found that the majority of TM3 cells stained for lower Fshr, only a few cells stained relatively higher *Fshr* (*Figure 2E-a and b*). Interestingly, we also noticed *Fshr* expression located in the nuclei (*Figure 2E-c to h*).

We then performed ddRT-PCR to compare *Fshr* expression in TM3 cells, testis, and ovaries of Fshr-ZsG and B6 mice at 3 months of age (*Figure 2G-a*, representative of three experiments). The results showed that *Fshr* expression was similar in the testes and ovaries of both types of mice, indicating that the insertion of P2A-ZsGreen did not disrupt *Fshr* expression in Fshr-ZsGreen mice. However, *Fshr* expression was significantly higher in the testes compared to the ovaries, by almost 40-fold. Interestingly, TM3 cells exhibited much lower *Fshr* expression levels than the testes and

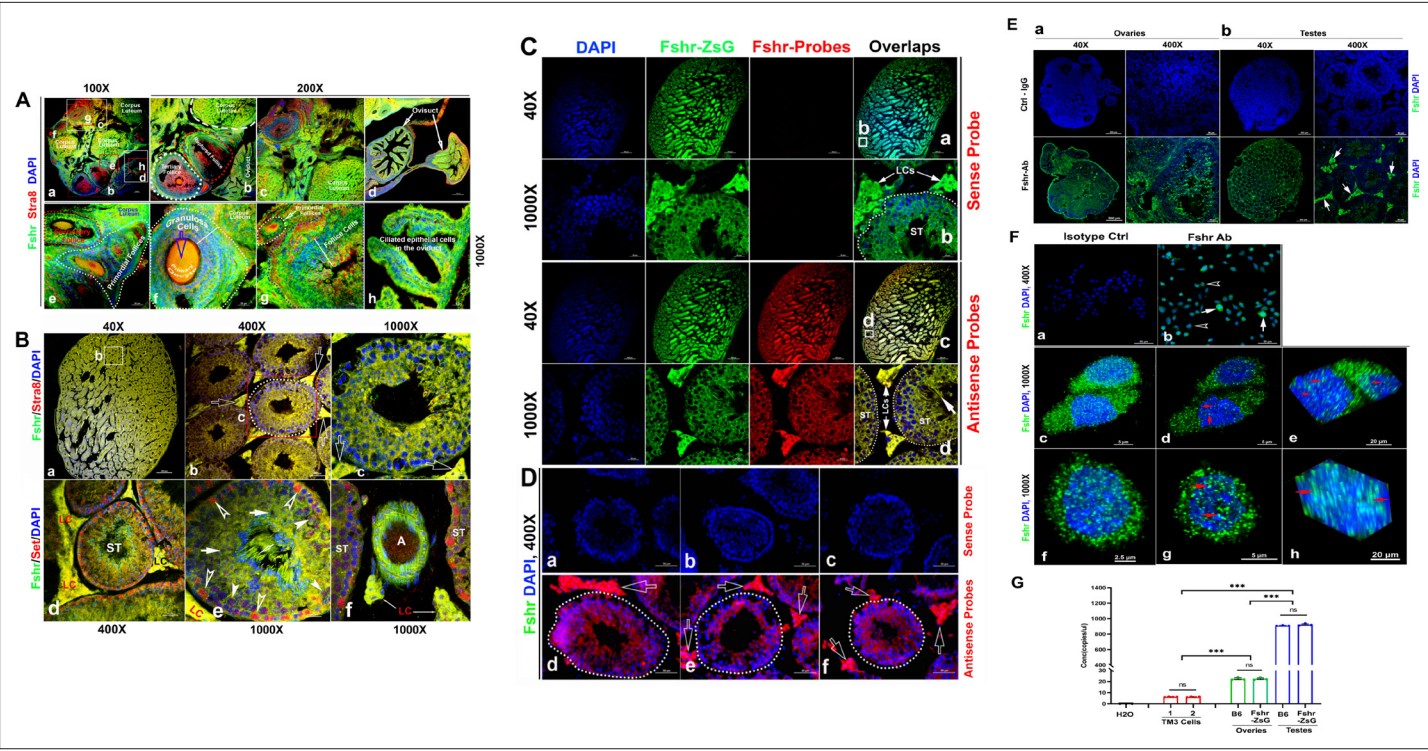

**Figure 2.** Imaging of Fshr-ZsGreen expression in the reproductive system. (**A**) Fshr expression in the ovary. Frozen sections of the ovary were immunostained with an antibody against mouse Stra 8, followed by imaging of Fshr-ZsGreen expression and its colocalization with Stra8 staining. The entire picture of the sectioned ovary is shown in (**A-a**). The representative images of the section were taken at ×400 magnification. (**A-b**) An area containing a corpus luteum, tertiary and ruptured follicles, and a partial oviduct; (**A-c**) an area with more corpora lutea; (A-d) oviducts. Furthermore, representative images at ×1000 magnification; (**A-e**) primordial and secondary follicles; (**A-f**) a mature follicle; (**A-g**) follicle cells in a corpus luteum; and (A-h) ciliated epithelial cells in the oviduct. Scale bars: 100 µm for (a, b, c, and d), 20 µm for (e–h). (**B**) Fshr-ZsGreen expression and its colocalization with Stra8 (**a–c**) and Set (**d–f**) in the testis. (**B-a**) The whole image of the sectioned testis; (**B-b**) the representative area of seminiferous tubes with interstitial cells (Leydig cells); and (**B-c**) an seminiferous tube. White empty arrowheads indicate Leydig cells; (**B-d**) the representative area of seminiferous tubes with interstitial cells (Leydig cells) stained for Set; (**B-e**) an seminiferous tube stained for Set and an artery with partial areas of seminiferous tubes stained for Set. Arrows: empty white arrows indicate Leydig cells; empty white arrowheads indicate spermatogonia; white arrowheads indicate Sertoli cells; white arrows indicate spermatocytes. Magnifications: ×40 for (**a**), ×400 for (**b and d**), and ×1000 for (**c, e, and f**). Scale bars: 500 µm for (**b and d**), 50 µm for (**c and d**), and 20 µm for (**c, e, and f**). (**C**) Single RNA-fluorescence in situ hybridization (RNA-smFISH) confirmation of Fshr expression in Fshr-ZsG mice. Mixed antisense probes were applied for detection of Fshr while a sense probe was taken as a negative control. The entire image of the sectioned testis was taken at ×40 magnification (**a and c**), and two representative areas of seminiferous tubes with Leydig cells are demonstrated at ×1000 magnification (**b and d**). Abbreviations: LCs, Leydig cells, and ST, seminiferous tubule. Scale bars: 50 µm. (**D**) RNA-smFISH confirmation of Fshr expression in B6 mice. Mixed antisense probes were applied for detection of Fshr while a sense probe was taken as a negative control. Three representative areas of seminiferous tubes with Leydig cells were imaged at ×400 magnification (**a–c** for the sense probe; **d–f** for antisense RNA probes). White empty arrows indicate Leydig cells. Scale bars: 50 µm. (**E**) Fshr expression in the ovary (**a**) and testis (**b**) of B6 mice. Frozen sections of the ovary and testis were immunofluorescence stained for Fshr. A representative area of the whole ovary or the testis image is present at ×400 magnification. White arrows indicate Leydig cells. (**F**) Immunofluorescence staining for Fshr in TM3 cells. (**a**) Isotype control (IgG); (**b**) positive staining for Fshr with mouse Fshr antibody, empty arrowheads indicate lower Fshr expression cells and white arrows indicate higher Fshr expression cells; (**c and f**) higher magnification of Fshr positively stained cells (maximum intensity projected images); (**d and g**) single layer images across the centers of nuclei of the cells in (**c and f**; **e and h**). 3D images of the nuclei showing Fshr located in the nuclei; (i) a SIM image of Fshr located in the nuclei. (**G**) Comparison of Fshr expression in TM3 cells, testes and ovaries of Fshr-ZsGreen and B6 mice assessed by droplet digital RT-PCR (ddRT-PCR). Two groups of TM3 cells (1 and 2) for measurement of Fshr expression (n=4 per group). Three samples for each organ (n=3 mice/each for the testis or ovary). ***p<0.001; ns, no significant difference in the comparisons.

ovaries. In fact, the levels were approximately 145-fold lower than in the testes and 3.6-fold lower than in the ovaries.

Last, we also searched for *FSHR* expression in single-cell RNA-seq (scRNA-seq) databases DISCO (https://immunesinglecell.org/genepage/FSHR) and BioGPS (http://biogps.org/#goto=genereport&id=2492), showing FSHR expression in human Leydig cells (*Appendix 1—figure 3*).

Overall, we observed Fshr-ZsGreen expression in the reproductive system, demonstrating that ZsGreen is a reliable readout for *Fshr* expression. In addition to its sole expression in granulosa cells and Sertoli cells, as reported previously, our findings clearly reveal that Fshr is also expressed in other cell types in the reproductive system, particularly in Leydig cells.

## Skeletal tissues

Fsh has been thought to have a direct role in bone (*Sun et al., 2006*) , therefore we next examined the expression pattern of *Fshr* in femoral sections, as shown in *Figure 3A*, under confocal fluorescence microscopy. The representative areas are presented at two magnifications of ×400 and ×1000. In the epiphyseal growth plate, we observed lower expression of Fshr in chondrocytes, as indicated by two dotted lines, compared to its expression in cells located in the sponge area above or under the dotted lines (*Figure 3A-a*). At higher magnification, *Fshr* was expressed in the columns of chondrocytes from the resting zone to the transformation zone (*Figure 3A-e*).

In contrast, we found that *Fshr* was brightly expressed in osteoblasts and osteoclasts in the metaphyseal trabeculae, which were recognized based on nuclear DAPI staining - osteoblasts were stained with a single DAPI nucleus, while osteoclasts contained more than two DAPI-stained nuclei (*Figure 3A-c and d*). Similarly, we further observed that these cells on the surfaces of trabeculae in bone marrow and cortical bone also clearly expressed Fshr-ZsGreen, as shown in *Figure 3A-e and f* (trabeculae) and *Figure 3A-g and h* (cortex). Importantly, we noted Fshr-ZsGreen expression in osteocytes (indicated by empty white arrowheads, *Figure 3A-d, g, and h*), the most abundant cell in the skeleton. These were embedded in trabecular and cortical bone, as well as in the periosteum (P) (*Figure 3A-h*).

To confirm the identification of osteoblasts/osteocytes and osteoclasts that expressed Fshr, we performed IF staining using an antibody against osteocalcin, a marker of osteoblasts (*Karsenty, 2017*). As shown in *Figure 1B*, we observed colocalization of Fshr expression with osteocalcin staining in chondrocytes in the transforming zone, as indicated by white arrows (*Figure 3B-a*), in cuboid and nucleated osteoblasts and bone lining cells on the trabecular and endosteal surfaces (*Figure 3B-b and c*), cortical bone, and osteocytes within the mineralized matrix (*Figure 3B-b, c, and e*).

To identify osteoclasts, we performed fluorescence immunostaining with an antibody against Trap, an osteoclast marker (*Lindunger et al., 1990*). Osteoclasts were recognized by positive staining for TRAP with more than two DAPI-stained nuclei. Multinucleated Fshr-ZsGreen-positive cells were stained positively for Trap located on the surface of the resorptive bays or areas adjacent to the trabecular bone (*Figure 3B-e and f*).

To examine whether skeletal stem/progenitor cells express *Fshr*, we performed IF staining for stem markers with antibodies against CD34 or CD133 (*Chan et al., 2015*; *Kuroda et al., 2014*; *Handgretinger et al., 2003*; *Handgretinger and Kuçi, 2013*; *Pozzobon et al., 2009*). Using these well-known stem markers, we identified Fshr-ZsGreen-positive cells as stem/progenitor cells located in the bone marrow, growth plate/articular cartilage, and periosteum, as shown in *Figure 2C-a, b, c, e, and d*, respectively. These cells also featured an increased nuclear-cytoplasmic ratio, except for these cells on the trabecular surface and in the periosteum.

We also examined *Fshr* expression in osteocytes from DMP1-CreERT$^{2+}$:Fshr$^{fl/fl}$ mice (osteocyte-specific Fshr cKO) where the control mice (DMP1-CreERT$^{2-}$:Fshr$^{fl/fl}$) as a positive control. We noticed a significant drop of Fshr expression in osteocytes in Fshr cKO mice when tamoxifen was intraperitoneally administered to the mice (*Figure 3D*). We also found an increased thickness of cortex triggered by Fshr cKO (*Figure 3D* - the left panels, the thickness indicated by dotted lines). The finding further demonstrated Fshr present in osteocytes, the largest cell population in bones, and the specificity of the Fshr antibody.

## Adipose tissues

Because our previous works and others have provided evidence on the role of Fsh in adipose tissues, we then examined Fshr-ZsGreen expression in adipose tissues. As expected, we found Fshr-ZsGreen expression in adipocytes of the frozen sectioned inguinal WAT, as shown in the left panel of *Figure 4A* at a low magnification (×40), which was further confirmed under a higher magnification (×400) in the three representative areas, demonstrating that Fshr-ZsGreen was expressed in the cellular membranes of individual adipocytes (*Figure 4A-a, b, and c*).

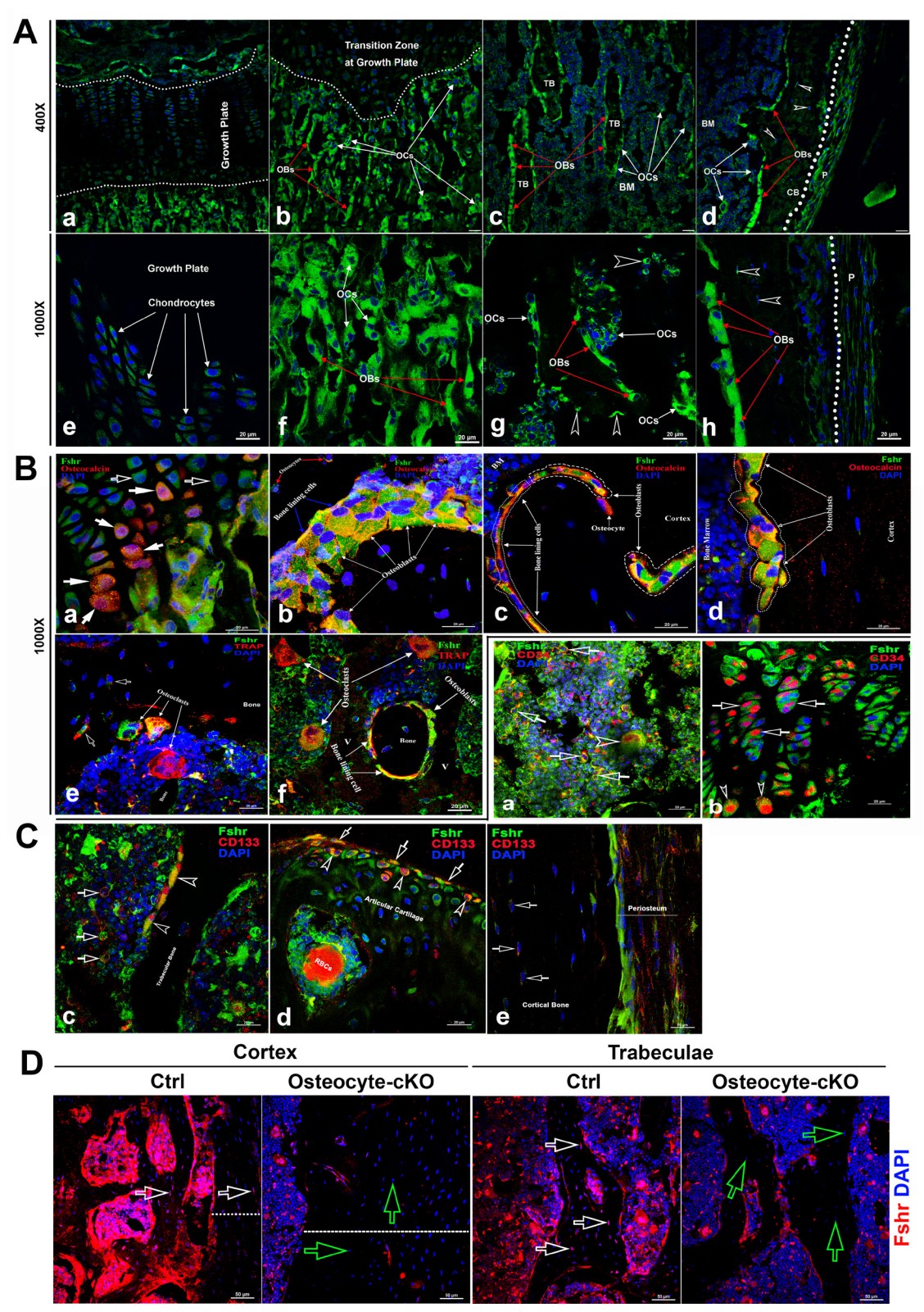

**Figure 3.** Fshr-ZsGreen expression in skeletal tissues. (**A**) Detection of Fshr-ZsGreen expression in frozen sectioned skeletal tissues by fluorescence confocal microscopy. The upper panels show Fshr-ZsGreen-positive cells in the representative areas of skeletal tissues at low magnification (×400), while the lower panels show their corresponding cells at high magnification (×1000). These representative areas demonstrate the Fshr-ZsGreen-positive cells in chondrocytes of the growth plate (**a and e**), on the surfaces of sponge bone under the growth plate (**b and f**), in trabecular bone in the bone marrow

*Figure 3 continued on next page*

*Figure 3 continued*

(**c and g**), and in the cortex (**d and h**). Abbreviations: OBs, osteoblasts; OCs, osteoclasts; BM, bone marrow; CB, cortical bone; and P, periosteum. Arrows: empty white arrowhead indicates GFP-positive osteocytes. Magnifications: ×400 for (**a–d**) and ×1000 for (**e–h**); scale bars: 50 µm for (**a–d**) and 20 µm for (**e–h**). (**B**) Confirmation of GFP-positive cell identities with antibodies against either osteocalcin or Trap as markers for osteoblasts or osteoclasts. Colocations of GFP expression with osteocalcin in mature chondrocytes (**a**), osteoblasts, bone lining cells, and osteocytes on the surface of or within sponge bone (**b**), trabecular bone (**c**), and cortical bone (**d**). Colocations of Trap with multinucleated GFP-positive cells are shown in (**e** and **f**), which represent osteoclasts on the resorptive areas and in bone marrow adjacent to bone surfaces and bone lining cells over trabecular bone. Arrows: empty white arrows indicate early chondrocytes with GFP expression but undetectable osteocalcin expression, while white arrows point to mature chondrocytes with both strong GFP and osteocalcin expression. Magnifications: ×1000 for (**a–f**). Scale bars: 20 µm for (**a–f**). (**C**) Identification of stem/progenitor cells. (**a**) Colocalization of Fshr-ZsGreen with CD34-positive staining in bone marrow, indicated by empty white arrows; an empty white arrowhead indicates multinucleated Fshr-ZsGreen cells with CD34-positive staining. (**b**) CD34-positive stained cells in growth plates. Empty white arrows indicate spindle-shaped ZsGreen-positive chondrocytes stained positively for CD34, where empty white arrowheads point to round cells with CD34-positive staining located in the bottom of the growth plate. (**c**) Colocalization of Fshr-ZsGreen with CD133 staining in the cells of bone marrow and on the bone surface. Empty white arrows indicate bone marrow cells positive for both Fshr-ZsGreen and CD133 staining, and empty white arrowheads indicate Fshr-ZsGreen-positive cells on the trabecular bone surface with positive staining for CD133. (**d**) Positive staining for CD133 on Fshr-ZsGreen-positive chondrocytes on the surface of articular cartilage. (**e**) CD133 staining in the weak Fshr-ZsGreen-positive fibroblast-like cells in the periosteum. (**D**) Reduced Fshr expression by Fshr cKO in osteocytes. Immunofluorescence staining with Fshr antibody was performed in decalcified frozen sections of femurs from the control and inducible osteocytes Fshr cKO mice (DMP1-CreERT$^{2+}$:Fshr$^{fl/fl}$ treated tamoxifen and DMP1-CreERT$^{2-}$:Fshr$^{fl/fl}$ as the control treated with corn oil). White empty arrows, osteocytes in the control; green empty arrows, osteocytes with reduced Fshrr expression in the cKO. Dotted lines indicate the thickness of the cortex. Magnifications: ×400. Scale bars: 50 µm.

To confirm adipocyte identification, we performed IF with an antibody against mouse Ucp1 (*Liu et al., 2017*) to recognize adipocytes, showing that the majority of Fshr-ZsGreen-positive cells were stained positively for Ucp1, and found two types of Fshr$^+$ adipocyte populations: one with colocalization of the two markers only in the membrane (indicated by empty white arrowheads) and another with two markers in both the membrane and cytoplasm (indicated by white arrows) (*Figure 4B-a and b*). In addition, we observed that Fshr-ZsGreen was expressed in arterioles in adipose tissue, denoted by white empty arrowheads in *Figure 4B-c and a* dotted circle in *Figure 4B-d*.

We further examined Fshr-ZsGreen expression in BAT. Fshr-ZsGreen was observed across the whole section of examined BAT at a lower magnification (*Figure 3A-a*). At a higher magnification (×400), three representative areas are presented, as shown in *Figure 3A-b, c, and d*, in which Fshr-ZsGreen was expressed not only in the cellular membranes but also in areas of cytoplasm close to the membranes, as indicated by white arrows (*Figure 4C-b, c, and d*).

Furthermore, we also performed IF with three antibodies against Ucp1, Th, and Peri to identify brown cells and peripheral fibers in the Fshr-positive section. We found that several areas in the section were strongly stained for Ucp1 at low magnification (×40), as shown in *Figure 4D-a*. Under higher magnifications of ×400 and ×1000, we colocalized Fshr-ZsGreen with Ucp1 in the three representative areas, in which some locations had higher Fshr-ZsGreen expression and others had higher Ucp1 expression (*Figure 4D-a, b, and d*).

To examine whether peripheral neural fibers express Fshr, we used antibodies against Th and Peri that can recognize peripheral neural fibers. We found that neural fibers stained positively and surrounded Fshr-ZsGreen-positive arterioles in BAT (*Figure 4E-a and b*). In addition, Fshr-ZsGreen was expressed in the nodes of Ranvier of TH-stained small neural fibers (indicated by empty red arrows, *Figure 4E-e*). We further confirmed Fshr-ZsGreen expression in peripheral neural fibers by IF with an antibody against peripherin (Peri), which is a type III intermediate filament protein found predominantly in peripheral nerves, specifically in sensory and autonomic neurons (*Figure 4F*). We noted colocalization of Fshr-ZsGreen and Peri staining in large neural fibers (*Figure 4F-b, d, and e*). Interestingly, we observed that both markers for peripheral neural fibers were also expressed in the cytoplasm of BAT cells, in which Fshr-ZsGreen was strongly expressed (*Figure 4E-b, c, and f*; *Figure 4F-e, c, and f*).

To further ensure *Fshr* expression in adipose tissues, we also performed IF staining with Fshr antibody in white and BAT adipose sections of B6 mice. The results showed the similar expression pattern as seen in Fshr-ZsGreen mice (*Figure 4G*).

Taken together, the above-described findings on Fshr-ZsGreen expression in reproductive, skeletal, and adipose tissues convincingly demonstrate that Fshr-ZsGreen is a reliable readout of Fshr expression. Furthermore, we identified Fshr expression in Leydig cells and follicles at different developmental

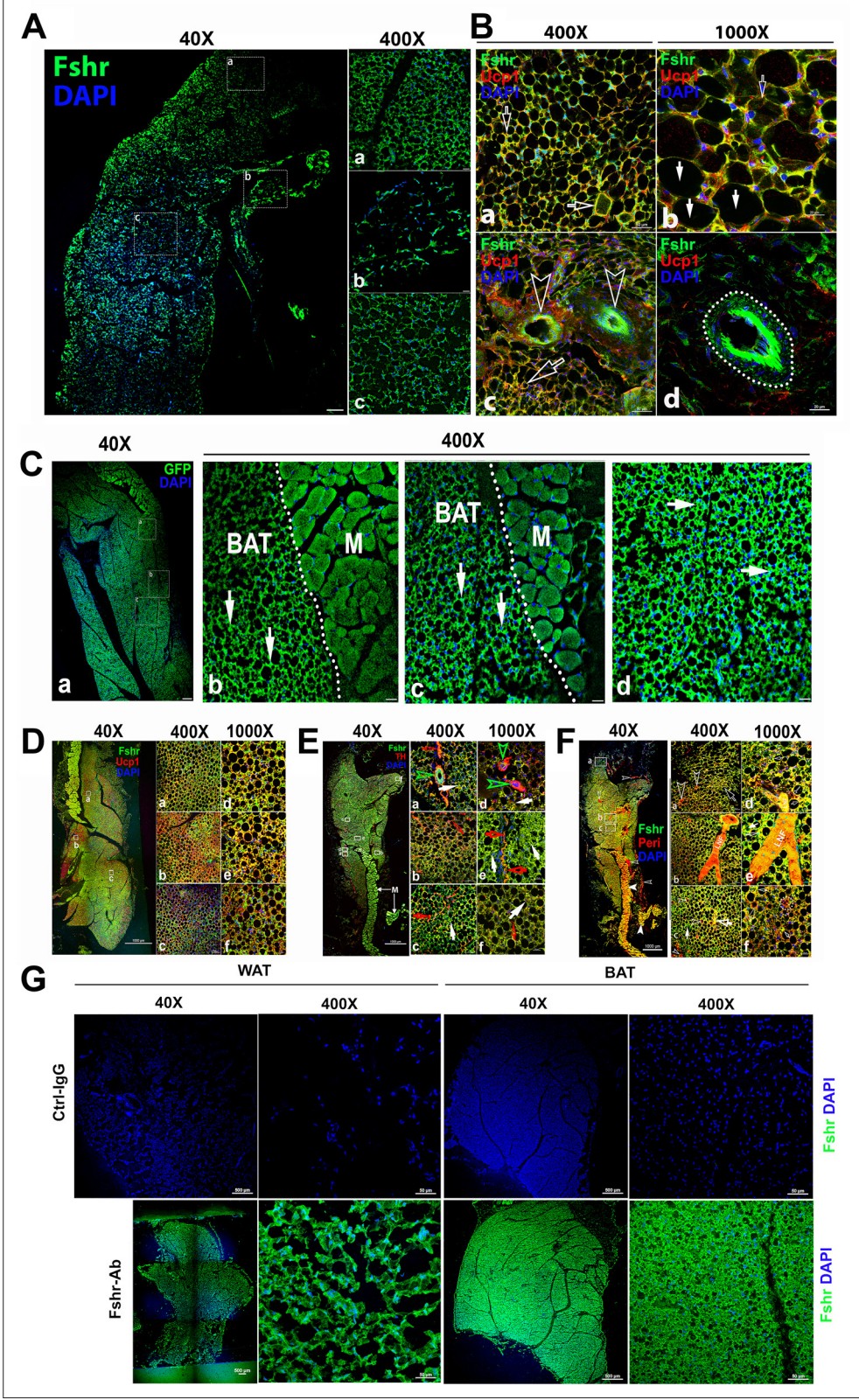

**Figure 4.** Fshr-ZsGreen expression in adipose tissues. (**A**) Confocal imaging of Fshr-ZsGreen expression in frozen sections of inguinal WAT. The whole image of a piece of the inguinal WAT was imaged under confocal microscopy with a low magnification (×40), and three representative areas were also presented with a higher magnification (×400) showing low and high cellular densities (low – **b** and high – **a** and **c**). These images demonstrate Fshr-

*Figure 4 continued on next page*

*Figure 4 continued*

ZsGreen expression in the cellular membranes of adipocytes. (**B**) Immunofluorescence (IF) staining of Fshr-ZsGreen-positive adipocytes with an antibody against mouse Ucp1. To identify the beige cells in WAT, we performed IF staining with an antibody against mouse Ucp1. Positive staining for Ucp1 was colocalized within Fshr-ZsGreen-positive cells in the areas with a higher cell density (**a and b**) imaged at low magnification, as shown in the left panels, while higher magnification images are shown in the right panels (**a and d**). In addition, strong Fshr-ZsGreen expression was observed in the arterioles of WAT (**c and d**). Arrows: empty white arrows indicate beige cells; white arrows point to white adipocytes, and empty arrowheads indicate GFP-positive arterioles. Magnifications: ×400 for (**a and c**); ×1000 for (**b and d**). Scale bars: 50 µm for (**a and c**) and 20 µm for (**b and d**). (**C**) Fluorescence images of Fshr-ZsGreen expression in frozen sectioned BAT. A whole image of BAT was imaged under confocal fluorescence microscopy at low magnification (**a**, ×40). Three representative areas are also presented at a higher magnification (**b, c, and d**, ×400), which clearly show the Fshr-ZsGreen expressed in the cells of BAT and skeletal muscles in the right parts of the images (**b and c**). Abbreviation: M, muscle. Arrows: white arrows indicate strong Fshr-ZsGreen expression in BAT cells. Magnifications: ×40 for (a) and 40°C for (**b, c, and d**). Scale bars: 100 µm for (a) and 50 µm for (**b, c, and d**). (**D**) Identification of brown adipocytes in BAT by IF staining. Brown fat cells were identified using an antibody against mouse Ucp1. Ucp1 staining is shown at lower magnification (×40) in the left panel covering a whole piece of BAT. Three representative areas are imaged at higher magnifications (**a, b, and c**, ×400 and ×1000), showing Fshr-ZsGreen colocalization with Ucp1-positive staining. Magnifications and scale bars are indicated in the figure. (**E**) Examination of peripheral neural fibers within BAT. To determine whether Fshr-ZsGreen is expressed in the peripheral nerves in BAT, we performed IF staining with an antibody against tyrosine hydroxylase (TH), a marker for peripheral sympathetic neurons. A whole piece of BAT was imaged at a lower magnification (×40) after being stained for TH, as shown in the left panel. Representative areas are presented in the right panels with higher magnifications (×400 - **a, b, and c** and ×1000 - d, e, and f, respectively). Arrows: red outlined arrowheads indicate Fshr-ZsGreen- and TH-positive large peripheral nerves; green outlined arrowheads point to Th-stained nerve fibrils accompanying Fshr-ZsGreen-positive nerve fibrils around an Fshr-ZsGreen-positive arteriole; red arrows indicate Fshr-ZsGreen- and TH-positive nerve fibrils; and white arrows indicate brown adipocytes with both Fshr-ZsGreen and TH expression. Magnifications: ×40 for the whole image of BAT in the left panel; ×400 for (**a, b, and c**); and ×1000 for (**a, e, and f**). Scale bars: indicated in the images. (**F**) Further confirmation of Fshr-ZsGreen expression in the peripheral nerves. To confirm Fshr-ZsGreen expression in peripheral neurons, we also employed another antibody against peripherin (Peri), a 57 kD type III intermediate filament that is a specific marker for peripheral neurons, to further identify peripheral neurons in BAT. The left panel shows an entire image of the BAT stained for Peri at a lower magnification (×40). The three representatives are shown in the right panel: (**a**) the first area located in the edge of the BAT with more white adipocytes; (**b**) the second area with more brown adipocytes and a large peripheral nerve; (**c**) the last area enriched with brown adipocytes. Their corresponding higher magnifications (×1000) are shown in (**a, e, and f**), respectively. Abbreviation: LNF, large never fibril. Arrows: empty white arrows - small peripheral fibrils; white arrow - Fshr-ZsGreen-positive fibrils, and empty white arrows - both Fshr-ZsGreen- and Peri-positive brown adipocytes. Magnifications: ×40 for the t panel; ×400 for (**a, b, and c**) and ×1000 for (**d, e, and f**). Scale bars: 1000 µm for the whole images in the left panel; 50 µm for (**a, b, and c**), and 20 µm for (**d, e, and f**). (**G**) Detection of Fshr expression in adipose tissues of B6 mice. Immunofluorescence staining for Fshr expression was carried out in frozen sections of white adipose tissue (the left panels) and brown fat (the right panels) from B6 mice at age of 3 months. Magnifications: ×40 and ×400. Scale bars: 500 µm and 50 µm.

stages, cells of osteoblast lineage, and peripheral nerve fibers. With confidence that Fshr-ZsGreen is a reliable readout, we used this powerful tool to further examine Fshr expression in other tissues and organs.

## Heart and aorta

To examine *Fshr* expression in the cardiovascular system, we used the heart and aorta as key organs/tissues to detect Fshr-ZsGreen expression. As expected, we observed strong Fshr-ZsGreen expression in the myocardium (*Figure 5A-a*) and large muscular arteries (a representative is shown in *Figure 5A-b*). Then, we further confirmed Fshr-ZsGreen expression by IFs with two antibodies against α-SMA and EMCN (*Chai et al., 2023*) that recognize alpha smooth muscle actin of smooth muscle and endomucin of endothelial cells at higher magnifications. With IF staining for α-SMA, we imaged several areas of both heart and blood vessels, as shown in the left image of the upper panel (×40). In the heart, we observed that Fshr-ZsGreen was highly expressed in cardiomyocytes in longitudinal and transverse orientations of the myocardium, which were positively stained for α-SMA (*Figure 6B-a, b, and c*). At a magnification of ×1000, it was also expressed in the endothelial layer of arterioles between muscle fibers (*Figure 5B-i*).

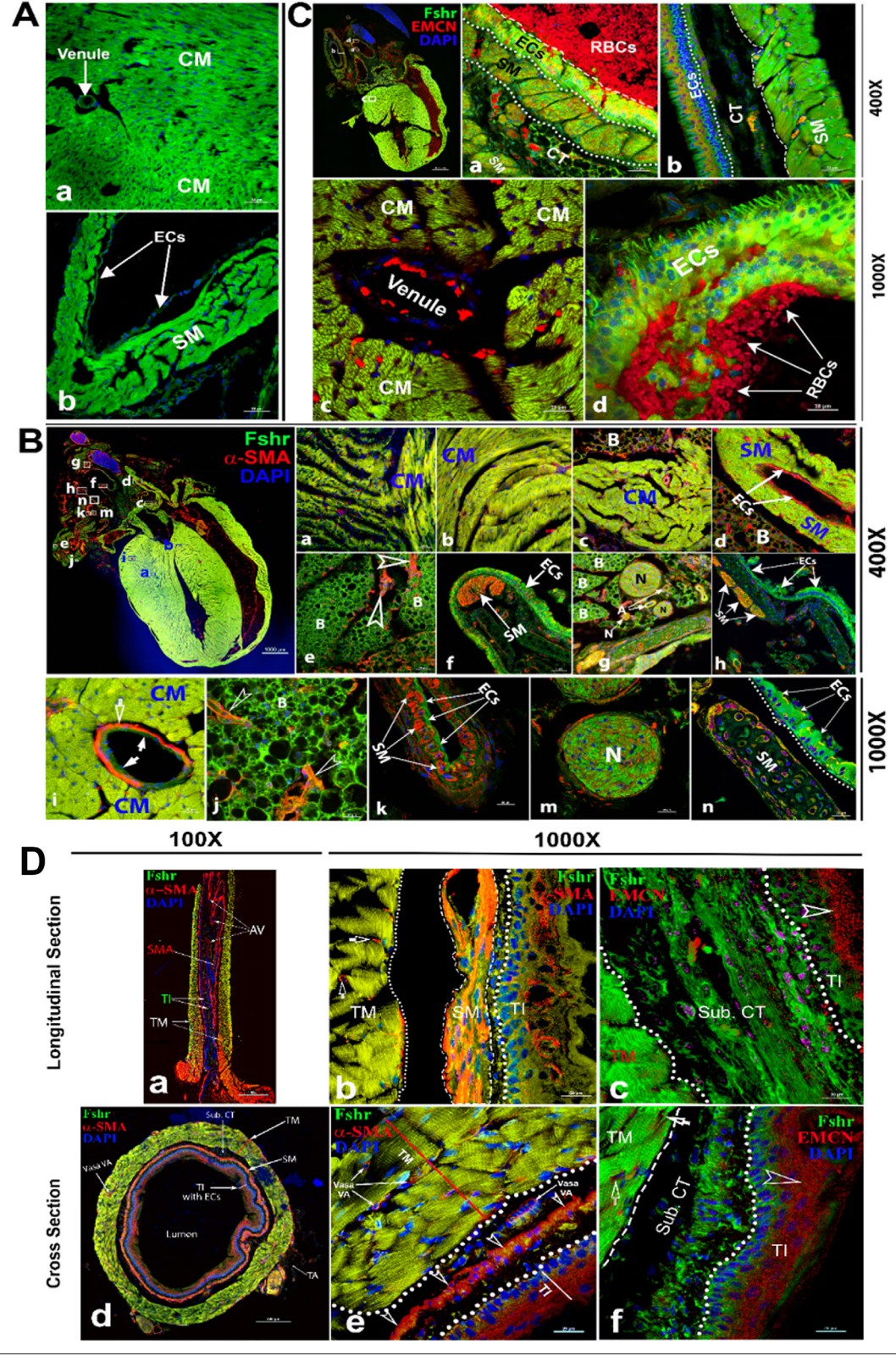

**Figure 5.** Imaging of Fshr-ZsGreen expression in the heart and aorta. Fshr-ZsGreen expression was imaged in frozen sections of the heart (**A**): cardiomyocytes (**a**) and smooth muscles (**b**). Then, immunofluorescence (IF) staining with antibodies against α-SMA (**B**) and EMCN (**C**) was carried out to identify cardiomyocytes and smooth muscles. The whole image of the heart with Fshr-ZsGreen expression and staining for α-SMA is shown in the left

*Figure 5 continued on next page*

*Figure 5 continued*

upper panel of (**B**). Its representative areas are presented at two magnifications of ×400 (**B-a** to **h**) and ×1000 (**B-i** to **n**), respectively, in the following: (1) ×400 magnification: (**a**) cross-oriented cardiomyocytes; (**b**) longitudinally oriented cardiomyocytes; (**c**) cross-oriented smooth muscle of the ascending aorta with brown adipose tissue; (**d**) smooth muscle of the left pulmonary artery with brown adipose tissue; (**e**) brown adipose tissue attached to a large artery; (**f**) layer of endothelial cells of the superior vena cava (SVC); (**g**) connective tissues between arteries containing brown adipose, different sized nerves and arteries; (**h**) another part of the SVC with layers of smooth muscles and connective tissue. (2) ×1000 magnification: (**i**) cardiomyocytes with an arteriole; (j) brown adipose tissue attached to the circulation system above the heart; (**k**) the layers of endothelial cells and smooth muscles of a bronchial artery; (**m**) transverse section of nerve fibers; (**n**) the layers of ECs and SM of the SVC. In addition, imaging of the IF staining for EMCN is shown in (**C**). The whole image of Fshr-ZsGreen and EMCN staining in the heart at a magnification of ×40 in the left upper panel, from which a representative of pulmonary artery at a magnification of 400 is shown in (**a**) at a magnification of ×400 (**a**); a representative of aorta is presented in (**b**). Representative images at higher magnification of ×1000: (**c**) cardiomyocytes in transverse orientation with a venule and (**d**) a layer of endothelial cells of pulmonary artery. Abbreviations: CM, cardiomyocytes; SM, smooth muscle; ECs, endothelial cells; N, never; B, brown adipose. Magnifications and scale bars are indicated as in the figure. Two orientations of the ascending aorta were examined for Fshr-ZsGreen after staining for α-SMA or EMCN: the longitudinal section (**D-a** to **c**) and the cross section (**D-d** to **f**). (**D-a**) shows the entire image of the ascending aorta stained for α-SMA at a magnification of ×100, while representative images at a higher magnification of ×1000 are presented in (**D-b**) for α-SMA staining and (**D-c**) for EMCN staining. The entire image of the transversely sectioned aorta (**D-d**) and representative images at a higher magnification of ×1000 for α-SMA (**D-e**) and EMCN (**D-f**). Abbreviations: TI, tunica intima, TM, tunica media, AV, aortic valve, SM, smooth muscles, Sub. CT, subendothelial connective tissue. Arrows: empty white arrows indicate arterioles positively stained for α-SMA in (b); empty arrowheads point to endothelial cells positively stained for either α-SMA or EMCN. Scale bars: 100 µm for (**a and d**) and 20 µm for (**b, c, e, and f**).

In addition to the cardiomyocytes, we also observed Fshr-ZsGreen in α-SMA-stained smooth muscles and endothelial cells of large blood vessels above the heart (*Figure 7B-d*). Interestingly, we found Fshr-ZsGreen in adipose tissue around the blood vessels, and the adipocytes morphologically appeared to be brown adipose cells, as the majority of these brown-like cells were full of ZsGreen-positive cytoplasm, instead of single large fat droplets with Fshr-ZsGreen expression in cellular membranes (*Figure 5B-c, d, e, and j*). The adipose tissue was stained positively for α-SMA, suggesting that the ZsGreen⁺ structures costained for α-SMA are blood vessels within the beige tissues, which are indicated by white empty arrowheads (*Figure 5B-e and j*).

Furthermore, we observed bright Fshr-ZsGreen expression with slightly weak staining for α-SMA in the layer of endothelial cells of the large vein in *Figure 5B-f and h*. Under the endothelial cells, a cluster of smooth muscles showed colocalization of Fshr-ZsGreen with positive staining for α-SMA (*Figure 5B-f and h*). In contrast to the vein, we observed a thin layer of endothelial cells (tunica intima) and stronger Fshr-ZsGreen expression with positive staining for α-SMA in the large artery (tunica media); Fshr-ZsGreen was also present in the cells in the tunica adventitia (*Figure 5B-k*). In frozen sections of hearts, we observed strong Fshr-ZsGreen expression in neural fiber clusters that were in adipose tissue, as shown in *Figure 5B-g and m*.

Using an antibody against EMCN, a marker for endothelial cells, we further confirmed Fshr-ZsGreen expression in the layer of endothelium (tunica intima), which showed visible positive staining for EMCN when imaged at ×400 and ×1000 (*Figure 5C-a, b, and d*). In addition, Fshr-ZsGreen⁺ cardiomyocytes and smooth muscles under the endothelial layer were positive for EMCN staining (*Figure 5C-a, b, and c*).

In addition to large blood vessels on the heart, we also took a close look at the ascending aorta. We obtained two types of sections with longitudinal and transverse orientations for IFs with the two antibodies as above. In both sections, we found that Fshr-ZsGreen⁺ smooth muscle fibers were strongly stained for α-SMA in the first layer of SM close to the endothelium, where the second layer of smooth muscle was relatively weak for staining with α-SMA (*Figures 5D-a and 6B-e*). Fshr-ZsGreen was also present at the endothelium, which was costained positively for α-SMA (*Figure 6A-a and B-e*). Using an anti-EMCN antibody, we noticed that positive staining was in the upper part of Fshr-ZsGreen⁺ endothelial cells facing the lumen of the examined blood vessels (*Figure 5A-c and B-f*). In addition to Fshr-ZsGreen expression in the endothelium of the tunica intima, it was also observed in the areas of

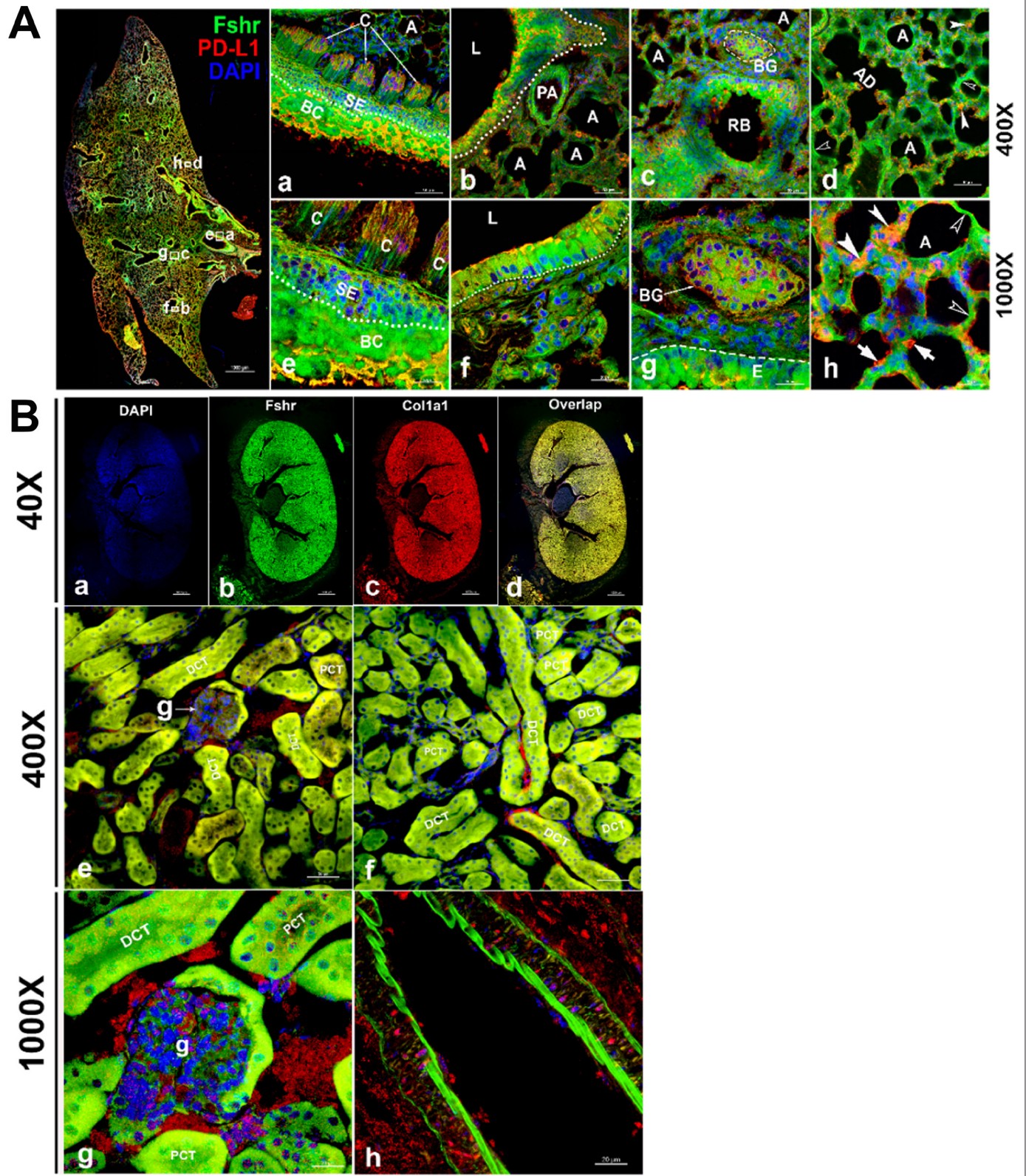

**Figure 6.** Fshr-ZsGreen expression in the lung and kidney. Detection of Fshr-ZsGreen expression and immunofluorescence (IF) staining with PD-L1 was performed in lung sections at different magnifications. The whole image of the frozen sectioned lung is shown in the left panel at ×40 magnification (A - left panel). Representative areas are shown in the right panels at magnifications of ×400 (**A-a to d**) and ×1000 (**A-e to h**). They are the ciliated columnar cells of primary bronchi (**a and c**), the bronchioles with alveoli (**b and f**), respiratory bronchiole with bronchial gland (**c and g**) and alveoli (**d**

*Figure 6 continued on next page*

*Figure 6 continued*

**and h**). Arrows: empty white arrowheads-type I pneumocytes, white arrowheads-type II pneumocytes and white arrows-macrophages. Abbreviations: C, ciliated epithelium; PA, pulmonary arteriole; L, lumen; RB, respiratory bronchiole; BG, bronchial gland; A, alveoli; AD, alveolar duct. Magnifications: ×40 for the left image of the whole section, ×400 for (**a–d**), and ×1000 for (**e–h**). Scale bars: 1000 µm for the left panel, 50 µm for (**a–d**), and 20 µm for (**e–h**). Fshr-ZsGreen expression and staining with Col1a1 were examined in the sectioned kidney under confocal fluorescence microscopy at three magnifications (**B**). The images in the top panel are images of the whole section at ×40 magnification (**a–d**). The images in the middle panel show the colocalization of Fshr expression with positive staining for Col1a1 in the glomerulus and renal tubes (proximal and distal convoluted tubes) at ×400 magnification (**e and f**), while the bottom panel shows images of the glomerulus (**g**) and arteriole (**h**) at ×1000 magnification. Magnification: ×1000. Scale bars: 100 µm for (**a–d**), 50 µm for (**e and f**), and 20 µm for (**g and h**). Abbreviations: A, arteriole; PCT, proximal convoluted tubule; DCT, distal convoluted tubule; g, glomerulus.

subendothelial connective tissue and tunica media (smooth muscle), but these areas were not stained for EMCN (*Figure 5A-c and B-f*).

## Lung and kidney

To identify whether *Fshr* is expressed in epithelial cells, we first detected Fshr-ZsGreen in the lung. Unexpectedly, we observed high Fshr-ZsGreen expression in the lung (*Figure 6A*). Fshr-ZsGreen was brightly expressed in the columnar epithelium of the respiratory conducting zone/tract, including the trachea, bronchus, bronchi, and bronchiole, at low magnification (*Figure 6A*, left panel). At higher magnifications, it was clearly shown that Fshr-ZsGreen was expressed not only in the columnar epithelium but also in the bronchial gland and alveoli (*Figure 6A-a, b, d, e, f, and g*). In the alveoli, Fshr-ZsGreen was observed in the respiratory portion of both type I and II cells, as indicated by empty arrowheads and white arrows, respectively (*Figure 6A-d and h*). We confirmed the identification of respiratory cells by IF with an anti-PD-L1 antibody, which showed the colocalization of Fshr-GFP with PD-L1 (*Guo et al., 2024*) staining (*Figure 6A*).

Then, we aimed to examine Fshr expression in epithelial cells of the kidney. In the frozen section of the kidney, it was astonishing to observe high expression of Fshr-ZsGreen in the proximal and distal convoluted tubules (*Figure 6B-e and f*), whereas relatively weak expression was observed in the glomerular capillaries at different magnifications (*Figure 6B-e and f* at ×400 and *Figure 6B-g* at ×1000). Again, we observed Fshr-ZsGreen expression in the endothelial layer of the arteriole in the kidney tissue (*Figure 6B-h*). We also observed colocalization of Fshr-ZsGreen with positive staining for Col1a (*Kuroda et al., 2014*) in the kidney (*Figure 6B*).

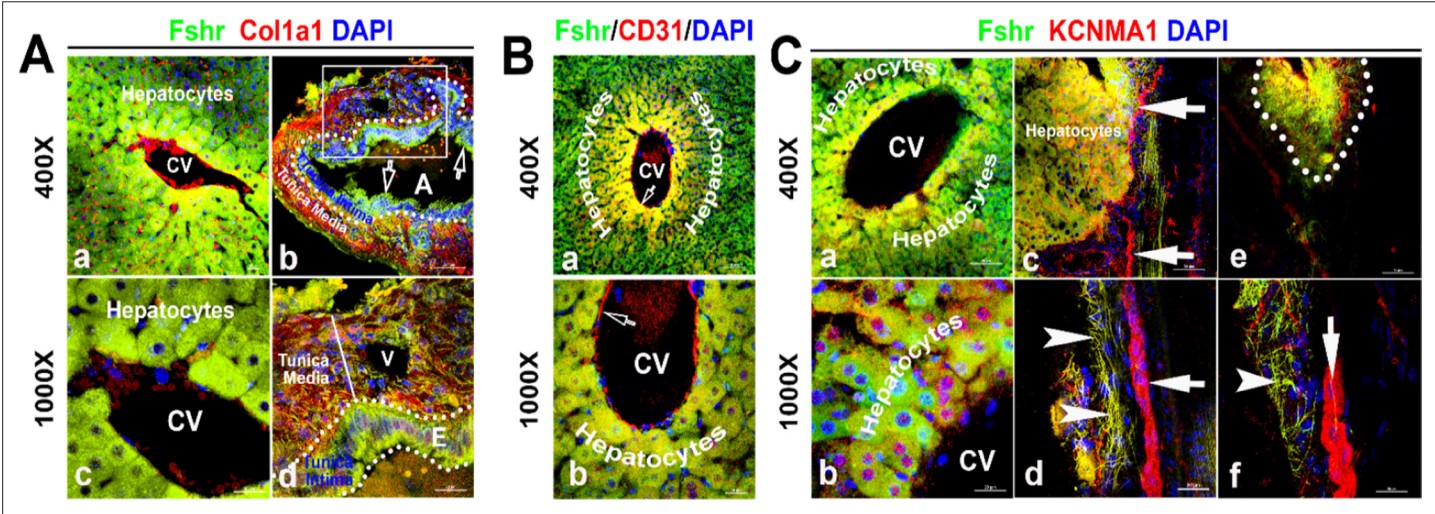

**Figure 7.** Identification of Fshr-ZsGreen expression in the liver. Frozen sectioned liver tissue was stained for Col1a1 (**A**), CD31 (**B**), or KCNMA1 (**C**), followed by florescent imaging for Fshr-ZsGreen and each of these stained molecules simultaneously. In the section stained for Col1a1 (**A**), two representative areas are shown at magnifications of ×400 and ×1000 as indicated: (1) hepatic cells with a central vein (CV) (**a and c**) and (2) hepatic artery (**b and d**). In the section stained for CD31 (**B**), a representative area of hepatic cells with a CV is shown at lower (**a**) and higher (**b**) magnifications. In the section stained for KCNMA1, three representative areas are presented at two magnifications as follows: (1) hepatic cells with a CV (**a and b**); (2) hepatic cells with peripheral nerve fibers (**c and d**); and (3) small nerve fibers located around a vein (**e and f**).

## Other key tissues and organs (liver, pancreas, thyroid, skin and skeletal muscle, spleen, bone marrow, and brain)

With the power of Fshr-ZsGreen, we characterized Fshr-ZsGreen expression in the liver. We observed Fshr-ZsGreen expression in the hepatocytes and arteries inside the hepatocytes, which were positively stained for Col1a1 or CD31 (*Wang et al., 2021*; *Ismail et al., 2003*), respectively (*Figure 7A and B*). Although it is weakly expressed inside large nerve fibers, Fshr-ZsGreen is strongly expressed in small neural fibers and shows a costaining pattern with KCBMA1 (*Simms and Zamponi, 2014*), a marker used for the detection of peripheral nerve fibers (*Figure 7C*, indicated by white arrowhead, whereas large nerve fibers are indicated by white arrows).

We then examined Fshr-ZsGreen expression in the pancreas and found that it was expressed not only in acinar cells but also in islets of Langerhans at low and high magnifications (*Figure 8A-a and b*). We then confirmed the identification of α- and β-cells by IFs with antibodies against NG3, insulin, or glucagon. The imaging results clearly demonstrated that Fshr-ZsGreen was expressed in both α- and β-cells as well as acinar cells at ×400 and ×1000 magnifications, respectively (*Figure 8A–C*).

Regarding Fshr-ZsGreen expression in the thyroid, we found Fshr-ZsGreen in both follicular cells and parafollicular cells (C-cells) at low and high magnifications (×400 and ×1000) (*Figure 9A-a, b, c, d, and e*). We further used an anti-Tshr receptor antibody to confirm Fshr-ZsGreen-positive cells, which showed that follicular cells were stained positively for Tshr in the nuclei, as indicated by dotted circles and white arrowheads, and C-cells were indicated by white arrows (*Figure 9-b to e*).

Next, to examine Fshr-ZsGreen expression in the skin, we used two types of skin sections: thick skin from the tail and thin skin from the abdomen (*Figure 10A-a and b*). Although a dermis layer was not included in the image taken for a thick sample, we observed that Fshr-ZsGreen was expressed in hair follicle (HF) and sweat gland cells and fibroblasts in the dermis and hypodermis (*Figure 10A-a*). Similarly, Fshr-ZsGreen was present in HFs and fibroblasts in the dermis and keratinocytes in the epidermis of the thin skin (*Figure 10A-b*). As we detected Fshr-ZsGreen expression in HFs, we wondered whether stem cells in HFs express Fshr-ZsGreen. To address this question, we carried out IF with an antibody against CD34, a stem cell marker. Not surprisingly, we found that CD34 staining was colocalized with cells with Fshr-ZsGreen in Bulge as quiescent stem cells and in the dermal papilla (DP) and epidermis as active stem/progenitor cells, as shown in *Figure 10B-a and b*. Therefore, these results indicate Fshr-ZsGreen expression in stem/progenitor cells in the skin (*Figure 10B-a and d*).

We then detected *Fshr* expression in skeletal muscle (gastrocnemius), followed by IF staining using three antibodies against α-SMA, PAX7, and TH for the identification of satellite cells and peripheral nerve fibrils. We observed Fshr-ZsGreen across the muscle sections at a lower magnification of ×40 (*Figure 11a*). At higher magnifications, we found that Fshr-ZsGreen was present in the muscle fibers, in which one type had higher Fshr-ZsGreen expression (indicated by white arrowheads) and another type had lower Fshr-ZsGreen expression (*Figure 11-b to h*). Fshr-ZsGreen was also highly expressed in satellite cells stained positively for either αSMA (*Figure 11-b to e*) or PAX7 (*Figure 11-f and g*) in both longitudinal and transverse sections, as indicated by white arrows. In addition, we detected Fshr-ZsGreen expression in the peripheral neural fibers identified by positive staining for TH, which were around vascular structures in the muscle tissue, as indicated by red arrowheads (*Figure 11-h and i*).

To detect *Fshr* expression in immune cells, we examined Fshr-ZsGreen expression in the spleen and bone marrow using antibodies against CD11b and CD3, as integrin αM (CD11b) is expressed in myeloid-lineage cells such as monocytes/macrophages, neutrophils, eosinophils, and basophils and in lymphoid cells such as NK cells and B-1 cells (*Springer, 1994*), while CD3 marks T-cells (*Chatenoud and Bluestone, 2007*). We imaged frozen sections of the spleen under magnifications of ×40, ×400, and ×1000 and observed that Fshr-ZsGreen was highly expressed in trabeculae and cells in both red and white pulps (RP and WP). The Fshr-ZsGreen-positive cells were further identified by IFs with antibodies against CD11B or CD3, indicating that Fshr is expressed in immune cells, such as monocytes/macrophages, neutrophils, eosinophils, basophils, NK cells, B-cells, and T-cells (*Figure 12A and B*). In addition, we further confirmed Fshr-ZsGreen expression in macrophages of bone marrow, which were identified with anti-CD4/80 antibody, as shown in two representative images of BM-one from an area located in the center of bone marrow (*Figure 12C-a*) and another close to cortical bone (*Figure 12C-b*) at ×1000 magnification (empty white arrows indicate the colocalization of Fshr-ZsGreen and CD11B, CD3, or CD4/80 in the RPs, while white arrows point to their colocalization in the WPs).

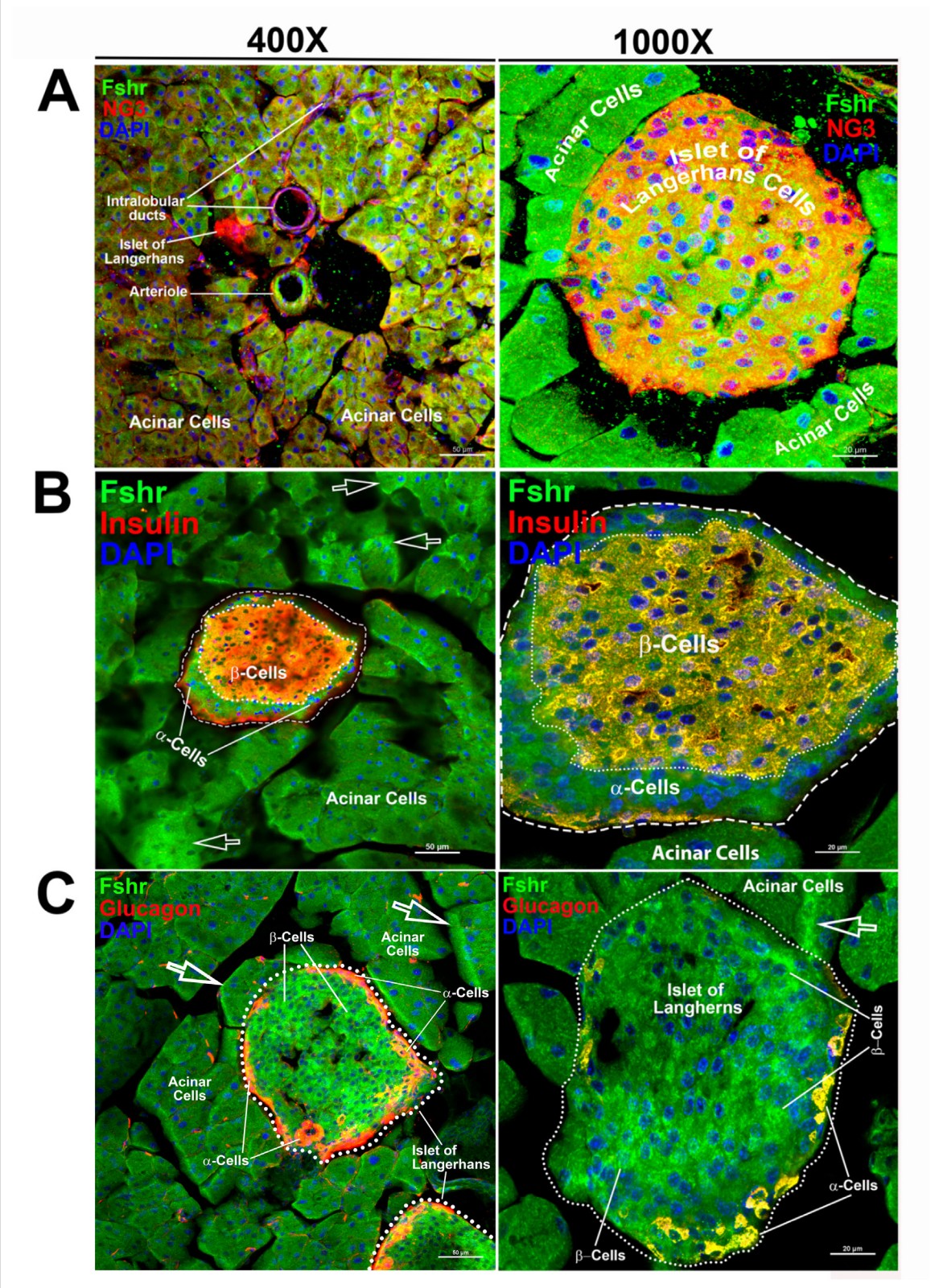

**Figure 8.** Visualization of Fshr-ZsGreen expression in the pancreas. Visualized Fshr-ZsGreen expression was obtained in frozen sections of the pancreas under fluorescence microscopy after immunostaining with antibodies against NG3, insulin, or glucagon at two magnifications (×400 and ×1000). Images of Fshr-ZsGreen expression with NG3, insulin, and glucagon staining are shown in **A**, **B**, and **C**, respectively. Arrows: white arrows indicate acinar cells with stronger Fshr-ZsG expression.

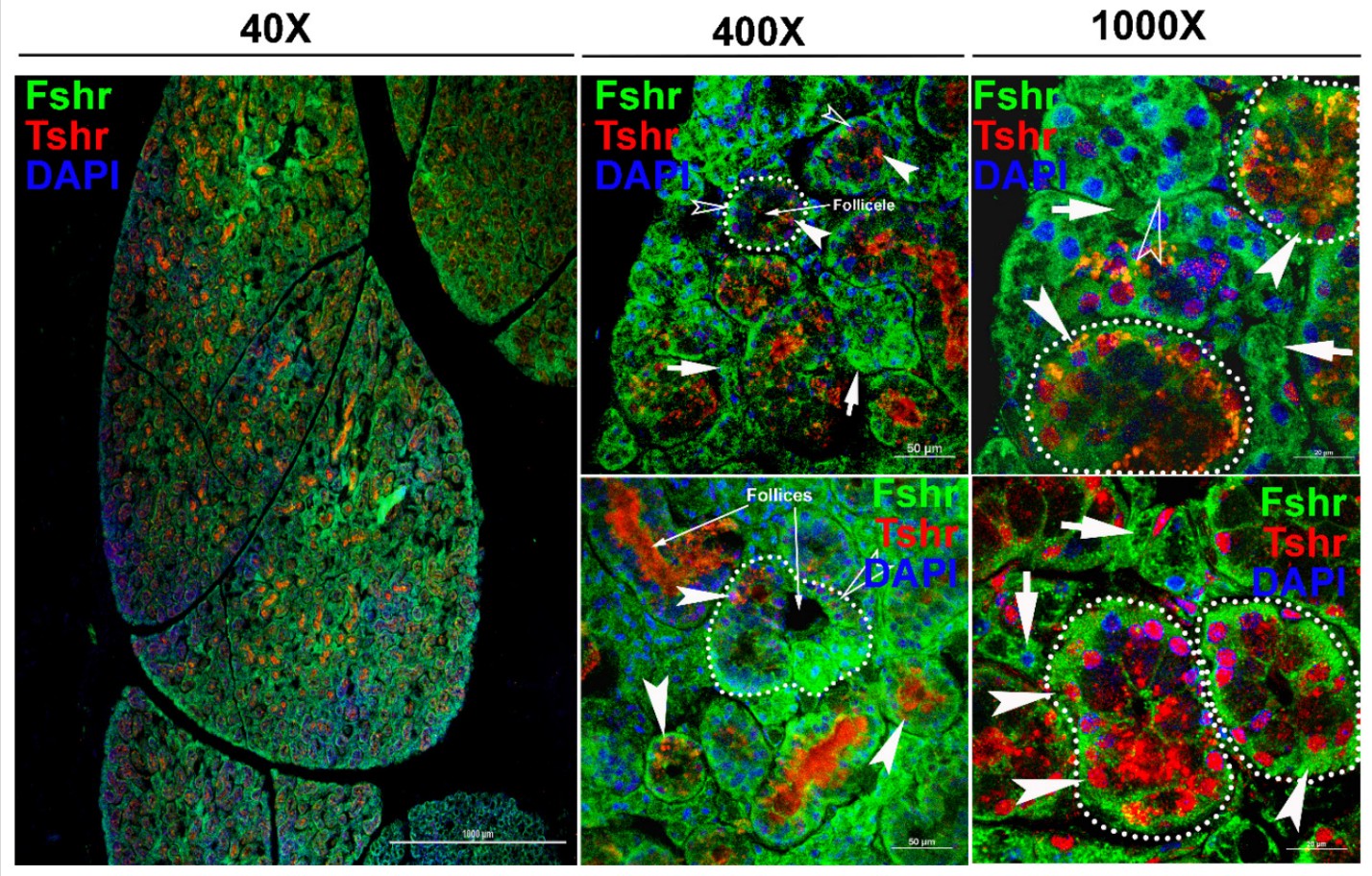

**Figure 9.** Fshr-ZsGreen expression in the thyroid. Fshr-ZsGreen expression was detected by its colocalization with immunostaining of TSH in frozen thyroid sections. The left panel is an entire image of the section, and two representative areas are in the right panels at higher magnifications, demonstrating Fshr-ZsGreen expression in follicular cells and parafollicular cells (C-cells) at the edge (**a and b**) and the center of the thyroid (**c and d**). Arrows: white arrowheads indicate follicular cells, and white arrows indicate parafollicular cells. Magnification: ×40 for the whole image.

Finally, we examined Fshr-ZsGreen expression and its colocalization with markers for either astrocytes, microglia, or neurons in the brain. As expected, we observed Fshr expression across the brain sections and the three representatives from the olfactory bulbs, pallidum, and hippocampus (CA3) are shown in *Figure 12*, demonstrating that Fshr is indeed expressed in astrocytes (*Figure 13-a to c*), microglia (*Figure 12-d to f*), and neurons (*Figure 13-g to j*), as well as neuronal fibers (synapses or projections). Other cell types are needed to be further defined.

### Confirmation of Fshr-ZsGreen expression with an Fshr-specific antibody and ddRT-PCR

Finally, to confirm the accuracy of the above results obtained from Fshr-ZsGreen mice, we performed fluorescence immunostaining with a specific antibody against mouse Fshr and accurate ddRT-PCR with mouse Fshr-specific primers to confirm the above data. An isotype-matched rabbit IgG was used as a negative control for IFs with anti-Fshr antibody using sections from Fshr-ZsGreen mice. Imaging was performed under the same conditions to record each corresponding tissue/organ stained with anti-Fshr antibody. The images of negative controls are shown in *Appendix 1—figure 2*, showing specific binding of the secondary antibody to anti-Fshr antibody without any nonspecific binding of the secondary antibody to the examined sections.

As shown in *Figure 14A*, in all the examined tissues, including bone, BAT, thyroid, cardiac muscle, kidney, liver, lung, aorta, ovary, and testis, we observed colocalization of Fshr-ZsGreen with positive

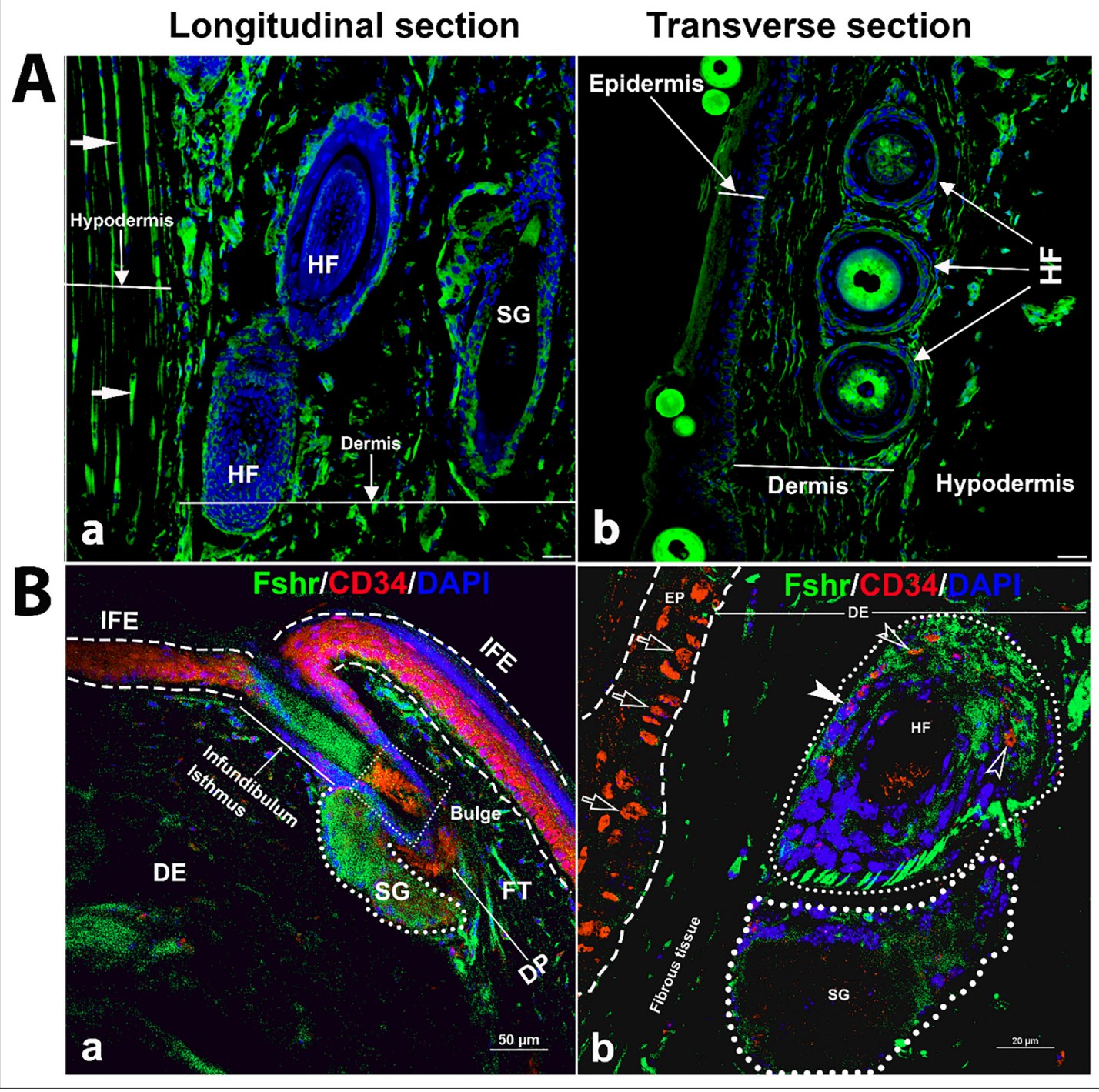

**Figure 10.** Detection of Fshr-ZsGreen expression in the skin. Fluorescence images of Fshr-ZsGreen were taken in two types of skin (**A**): thick skin (**A-a**) and thin skin (**A-b**). Then, immunofluorescence (IF) staining was performed using an antibody against CD34 to identify stem cells with Fshr-ZsGreen expression in the hair follicles (**B**) at lower (**B-a**) and higher (**B-b**) magnifications (×400 and ×1000, respectively). Abbreviations: HF, hair follicle; SG, sweat gland; FT, fat tissue; DP, dermal papillae. Magnifications: ×400 for (**A** and **B-a**), ×1000 for (**B-b**). Scale bars: 50 μm for (**A** and **B-a**) and 20 μm for (**B-b**).

staining for Fshr, further confirming the specific expression of Fshr-ZsGreen in the examined tissues/organs.

We also obtained total RNA from the following tissues: lung, kidney, thoracic vertebra, calvaria, femur, jejunum, liver, teeth, tibia, skeletal muscle, tails, cartilage, skin, spleen, stomach, heart, bladder, tongue, BAT, WAT, thyroid, brain, pancreas, and duodenum. After reversing mRNA from total RNA,

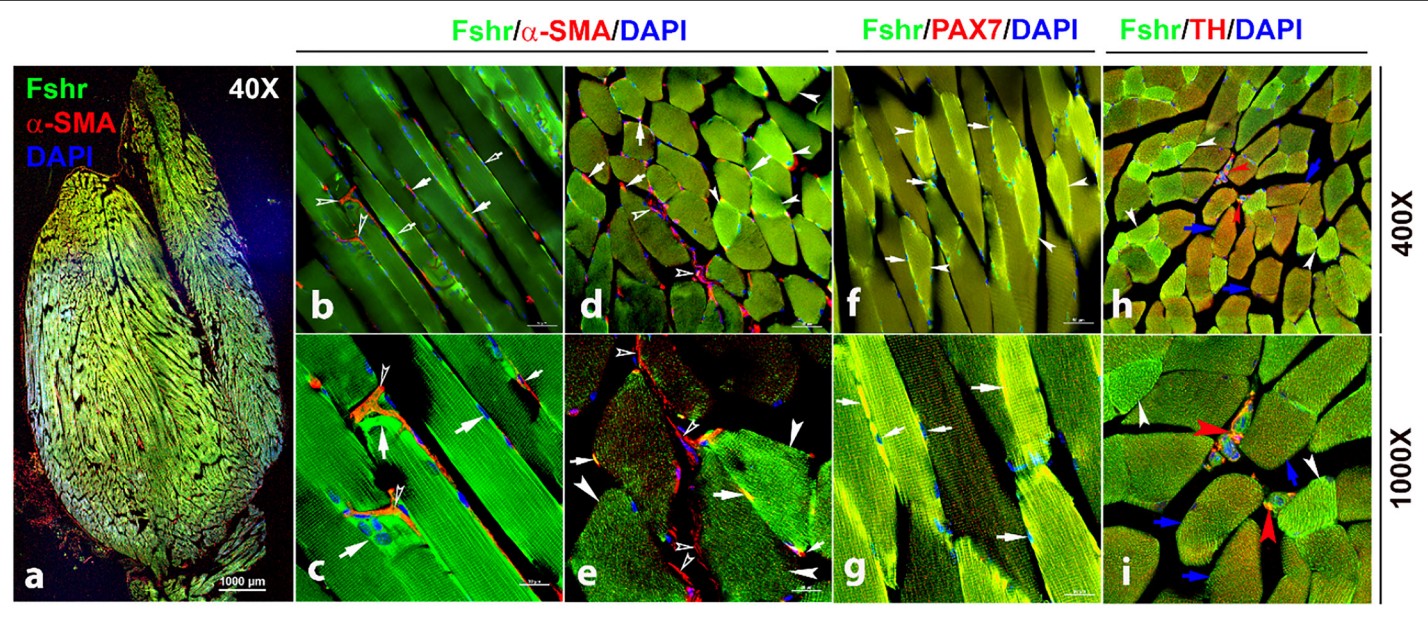

**Figure 11.** Fshr-ZsGreen expression in skeletal muscle. The whole image of frozen sectioned skeletal muscle (gastrocnemius) was taken at a lower magnification (×40) after the section was stained with anti-α-SMA antibody, as shown in the left panel (**a**). Then, longitudinal sections (**b and c**) and cross-sections (**d and e**) were imaged at higher magnifications (×400 and ×1000), respectively. In addition, frozen sections stained with antibodies against PAX7 (**f and g**) and TH (**h and i**) were imaged at two magnifications (×400 and ×1000) as indicated. Arrows: (1) In (**b–e**), empty white arrowheads indicate both Fshr-ZsGreen- and α-SMA-positive satellite cells; empty white arrows point to only Fshr-ZsGreen-positive satellite cells without α-SMA staining; white arrowheads indicate muscle fibrils with strong Fshr-ZsGreen expression but no α-SMA staining. (2) In (**f and g**), white arrowheads indicate muscle fibrils with strong Fshr-ZsGreen expression, blue arrows point to muscle fibrils with both Fshr-ZsGreen and TH expression, and red arrowheads indicate peripheral nerves with both Fshr-ZsGreen and TH expression. Magnifications: ×40 for (a), ×400 for (**b, d, f,** and **h**), and ×1000 for (**c, e, g,** and **i**). Scale bars: 1000 μm for (**a**), 50 μm for (**b, d, f,** and **h**), and 20 μm for (**c, e, g,** and **i**).

we performed ddRT-PCR with mouse-specific primers to check *Fshr* expression at the transcriptional level as described in the Methods section. The results demonstrated Fshr expression in these tissues, and the expression profile was categorized into three groups: (1) high expression in the lung and kidney tissues; (2) moderate expression in the thoracic (T) vertebrae, calvaria, femur, jejunum, liver, teeth, tibia, and muscle; and (3) low expression in the tail, cartilage, skin, spleen, stomach, heart, bladder, tongue, BAT, WAT, brain, pancreas, and duodenum (*Figure 14B*).

Finally, we also searched *Fshr* expression at single-cell level in four single-cell databases including DISCO (a database of <u>D</u>eeply <u>I</u>ntegrated <u>S</u>ingle-<u>C</u>ell <u>O</u>mics data, https://www.immunesinglecell.org/), BioGPS (a free extensible and customizable gene annotation portal, a complete resource for gene expression and protein function, http://biogps.org/exrna/#goto=welcome), Single Cell Portal (SCP, an interactive home for single-cell genomics data, https://singlecell.broadinstitute.org/), and Genotype-Tissue Expression (GTEx portal, https://gtexportal.org). The selected results from these databases are presented in *Appendix 1—figure 3*, which further support our findings of the widespread express pattern of Fshr described as above. In particular, Fshr expression was detected in Leydig cells by scRNA-seq as seen in DISCO (https://immunesinglecell.org/genepage/FSHR) and BioGPS (http://biogps.org/#goto=genereport&id=2492) for human cells.

Taken together, these data further confirmed Fshr-ZsGreen expression patterns from the Fshr-ZsGreen reporter line, convincingly demonstrating that Fshr is not limited to previously reported cells, tissues, or organs, such as the reproductive system, osteoclasts, adipose, endothelium in tumors, and neurons in the brain, but rather has a wide expression in the cells, tissues, and organs in the body, particularly in the lung, kidney, and heart, as well as Leydig cells in the testis, which has not previously been recognized.

## Discussion

Although numerous efforts have been made to characterize *Fshr* expression in tissues/cells, defining the locations of *Fshr* expression remains an imperative challenge in Fsh-Fshr biology, primarily because

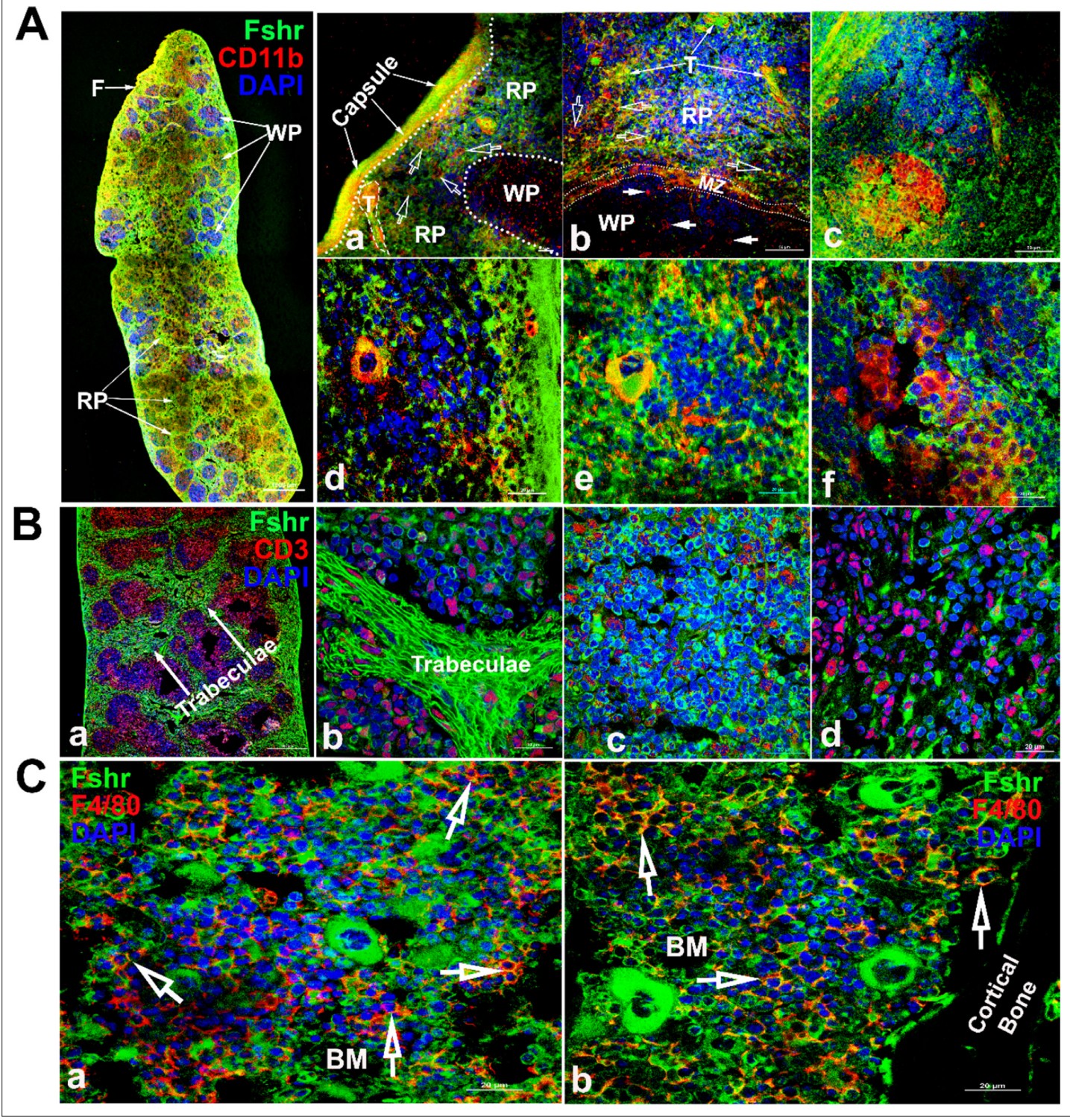

**Figure 12.** Examination of Fshr-ZsGreen expression in the spleen and bone marrow. To identify Fshr-ZsGreen expression in immune cells of the spleen, immunofluorescence (IF) staining was performed in frozen sections of the spleen. Two antibodies against either CD11B or CD3 were used to identify myeloid-lineage cells or T-cells. Fshr-ZsGreen expression in myeloid-lineage cells with staining of CD11B is shown in (A), in which the left panel is the whole image of the spleen at a low magnification and its three representative areas are presented at higher magnifications (×400 and ×1000): (1) an area located at the edge showing strong Fshr-ZsGreen and CD11B expression (**a and b**); (2) an area of red pulp (**c** and **d**); (3) an area of white pulp (**e and f**). Fshr-ZsGreen expression in T cells with staining of CD3 is shown in (**B**), in which a large piece of spleen is shown in a and three representative areas are presented at a magnification of 100x (**b**-trabeculae; **c**-a WP area and **d**-a RP area). Images for bone marrow sections with staining of F4/80 are shown

*Figure 12 continued on next page*

*Figure 12 continued*

in (**C**) (**a**-the center of bone marrow and **b**-an area close to the cortex). Abbreviations: RP, red pulp; WP, white pulp; BM, bone marrow; F, fibroelastic capsule; T, trabeculae; and MZ, marginal zone. Scale bars are indicated in each image.

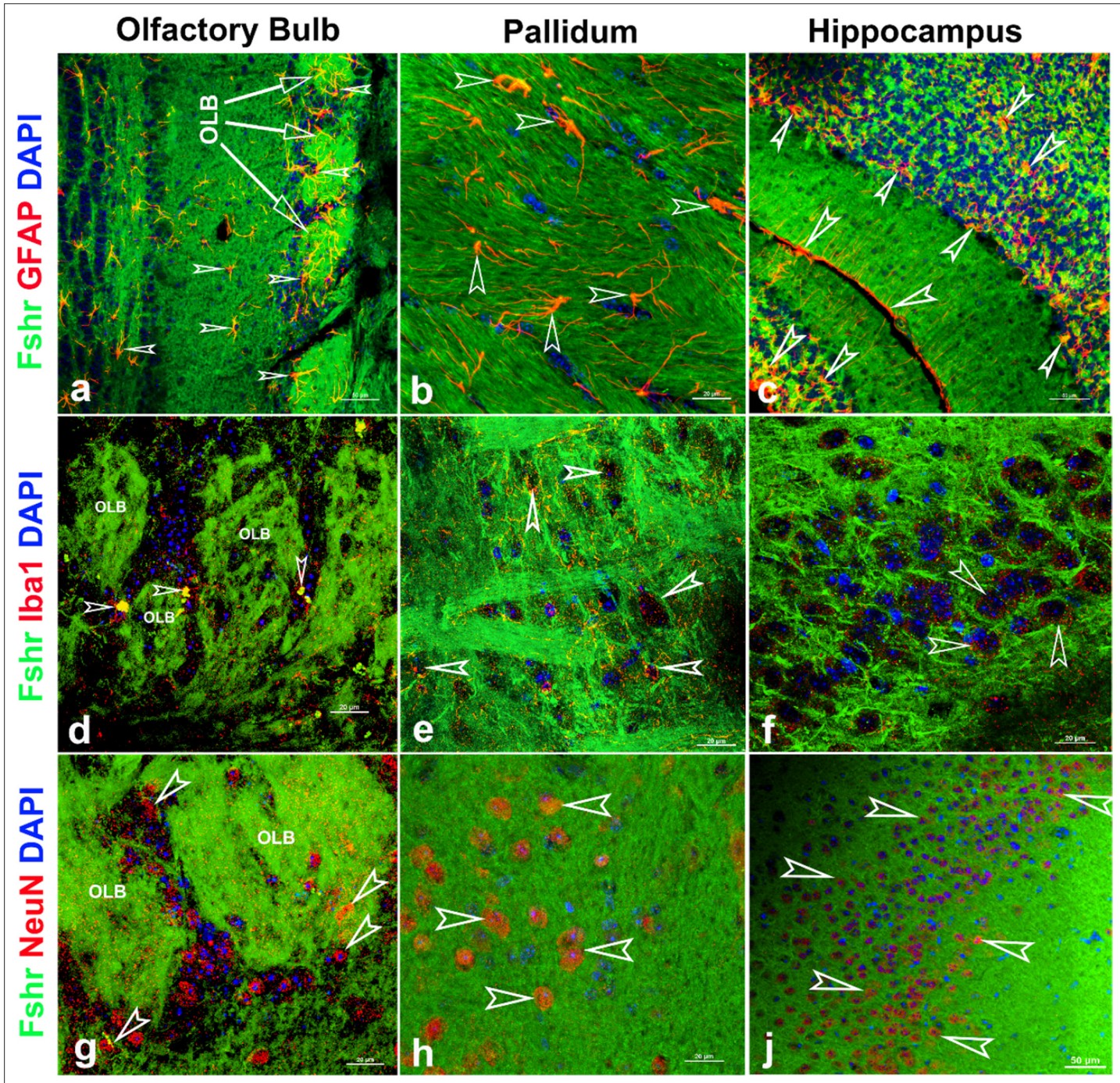

**Figure 13.** Fshr-ZsGreen expression in the representative areas of the brain. Fshr expression in three representative areas of the brain - the olfactory bulbs (**a**, **d**, and **g**), pallidum (**b**, **e**, and **h**), and hippocampus (**c**, **f**, and **j**). Each section was immunofluorescently stained with antibodies against GFAP, Iba1, or NeuN that recognized markers for astrocytes (**a–c**), microglia (**d–f**), or neuron (**g–j**), respectively. Their colocalizations are indicated by white arrowheads. Abbreviation: OLB, olfactory bulb. Magnifications: ×400 for (**a, c, and f**) and ×1000 for (**b, d, e, f, g, and h**). Scale bars are indicated in each image.

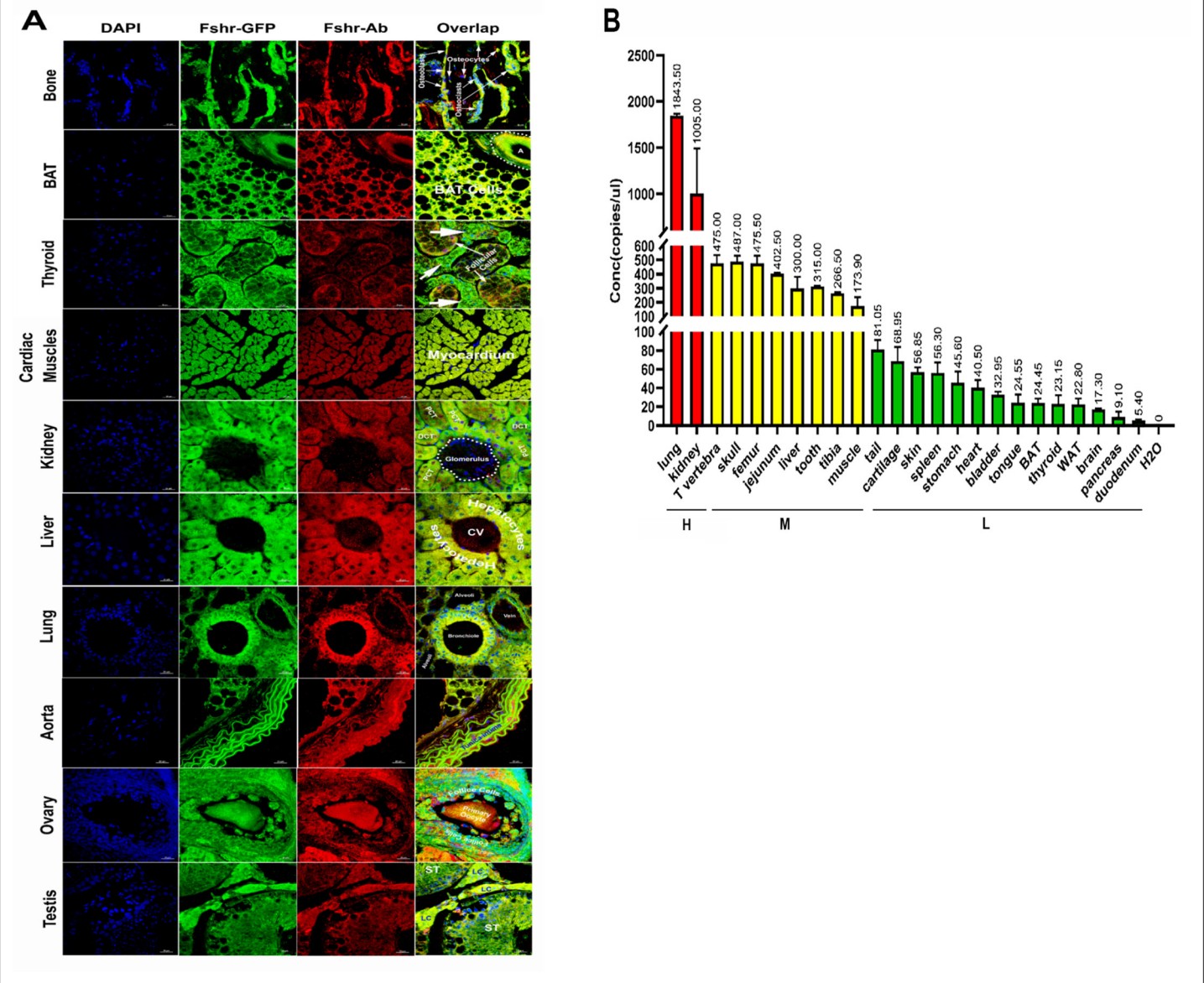

**Figure 14.** Confirmation of Fshr-ZsGreen expression by immunofluorescence (IF) staining with a specific antibody against mouse Fshr and droplet digital RT-PCR (ddRT-PCR). The Fshr-ZsGreen expression described above was further confirmed by IF staining using an antibody against mouse Fshr and ddRT-PCR. The frozen tissue sections used for this confirmation include bone, BAT, thyroid, cardiac muscles, kidney, liver, lung, aorta, ovary, and testis (**A**), demonstrating Fshr-ZsGreen colocalization with Fshr-positive staining in these tissues/organs. Fshr expression at the mRNA level in different tissues/organs was examined by ddRT-PCR (**B**) (the representative of two experiments). The results indicate that Fshr is expressed in all examined tissues/organs (n=3). Based on their expression levels, they can be categorized into three groups: (1) high (H), including lung and kidney, ranging from 1005 to 1843 copies/µL; (2) middle (M), including thoracic (T) vertebra, skull, femur, jejunum, liver, tooth, tibia, and muscle, ranging from 173 to 475 copies/µL; and (3) low (L), including duodenum, pancreas, brain, WAT, thyroid, BAT, tongue, bladder, heart, stomach, spleen, skin, cartilage, and tail, ranging from 5.4 to 81 copies/µL.

of concerns about the specificity of available antibodies against Fshr (*Chrusciel et al., 2019*; *Kumar, 2014*; *Tedjawirja et al., 2023*). In this case, we developed CRISPR/Cas9-mediated *Fshr*-ZsGreen knockin reporter mice to address this issue.

To maintain the integrity of the splicing acceptor, donor, and gene promoter regions, we designed and inserted the ZsGreen (ZsG) reporter into the C-terminus of Fshr by sequence-specific gRNA-guided CRISPR/Cas9-mediated precise genome modification with a long ssDNA template to create the *Fshr*-P2A-ZsG reporter mice (*Bai et al., 2020*). Because of its high turnover of Fshr mRNA and a difficulty in detection of Fshr expression by RT-PCR and northern blotting, in this reporter line, we

utilized ZsG as a GFP reporter, as ZsGreen, also called ZsGreen1, is an exceptionally bright green fluorescent protein derived from a *Zoanthus* sp. reef coral (*Matz et al., 1999*) that has been modified for high solubility, bright emission, and rapid chromophore maturation. ZsGreen is the brightest commercially available green fluorescent protein - up to 4× brighter than EGFP with the half-life of 26 hr - and is ideally suited for whole-cell labeling and promoter-reporter studies to indicate the promoter activity of a gene of interest, which has been used in numerous GFP reporter mice (*Zhu et al., 2012b*; *Pippin et al., 2013*; *Pippin et al., 2014*; *Büller et al., 2015*; *Dewas et al., 2015*; *Nazareth et al., 2015*; *Xiao et al., 2015*; *Kaverina et al., 2016*; *Saito et al., 2016*; *Fougère et al., 2021*). In addition, we also employed a short and conserved picornavirus-derived 'self-cleaving' 2A peptide to allow bicistronic expression of Fshr and ZsG, as the 19-amino acid P2A has the highest cleavage efficiency (*Wang et al., 2015*; *Kim et al., 2011*). The cleavage is triggered by ribosomal skipping of the peptide bond between the proline (P) and glycine (G) in C-terminal of P2A peptide, resulting in the peptide located upstream of the 2A peptide to have extra amino acids on its C-terminal end, while the peptide located downstream the 2A peptide will have an extra proline on its N-terminal end. Therefore, P2A can nearly equalize the expression of genes upstream and downstream. As a result, Fshr is normally expressed without interruptions and ZsG expression indicates Fshr promoter activity in the defined cells, demonstrated by the comparison of Fshr expression in the testes and ovaries between Fshr-ZsGreen and B6 mice (*Figure 2G*).

The P2A-ZsGreen construct was precisely inserted into the site between the last exon (exon 10) and the stop codon of the Fshr locus, which was confirmed by integration detection PCR, sequencing of PCR fragments, and Southern blotting with Fshr locus-specific enzyme restriction digestions. Successful insertion allows the endogenous promoter of Fshr to drive ZsGreen reporter expression. Because of the site-specific insertion of the P2A ZsG vector to produce Fshr-ZsG reporter, we only used one founder line for characterizing Fshr-ZsG expression, rather than multiple founders when random insertions of a transgene were used previously for generating transgenic mice.

This approach greatly enhanced our understanding of this critical cellular pathway in several ways. First, employing the native regulatory element responsible for governing the expression of the target gene ensures that the expression pattern of the GFP reporter closely mirrors that of the gene of interest, Fshr, within its natural context. Consequently, this approach provides a more faithful representation of gene expression. Second, endogenous promoters often exhibit specific spatial and temporal expression patterns, driving gene expression in specific cell types or developmental stages. This capability facilitates the capture of the dynamic nature of gene regulation and enables the study of gene expression under diverse physiological conditions. Third, utilizing endogenous promoters minimizes potential perturbations to the native gene regulation machinery. It avoids the need to introduce exogenous elements or artificial constructs, reducing the likelihood of altering the gene's expression behavior or interfering with its regulatory interactions. Unlike biochemical assays or immunostaining, using a tagged protein under endogenous regulation avoids fixation artifacts and allows detection of the target's activity in live cells. Therefore, these distinct advantages of enhanced physiological relevance, precise spatiotemporal control, preservation of regulatory elements, and minimal perturbation provide us with a powerful tool to understand Fshr expression in more accurate and context-specific ways compared to other methods, such as using antibodies, northern blotting, RT-PCR, and in situ hybridization.

In this study, we systemically investigated Fshr expression at the single-cell level in this reporter line with confocal fluorescence microscopy and further confirmed the location of Fshr-ZsGreen expression by IF staining, in situ hybridization, and ddRT-PCR. The results from this work demonstrate that as a receptor for Fsh, Fshr is widely expressed in virtually every cell at variable levels in the examined tissues/organs of mice.

As expected in the examined testis, we noticed Fshr expression in Sertoli cells in the testis and granular cells in the ovary. Surprisingly, we observed that *Fshr* was also more strongly expressed in Leydig cells than in Sertoli cells. This expression was also detected in spermatocytes, spermatids, spermatozoa, and spermatogonia. Although this finding is different from present thoughts that *Fshr* is only present in Sertoli cells but not in other cell types of the testis, our finding is in line with previous works in fishes (including African catfish, zebrafish, teleosts, and Japanese eels) (*García-López et al., 2009*; *Levavi-Sivan et al., 2010*; *García-López et al., 2010*; *Burow et al., 2020*; *Chauvigné et al., 2012*; *Ohta et al., 2007*, rats *An et al., 2022*), dogs (*Kasimanickam and Kasimanickam, 2015*), and

humans (*Bonci et al., 2018*) performed by immunohistochemistry (IHC) and in situ hybridization. In humans, *Fshr* was also highly expressed in Leydig cells, although it was taken as non-specificity of the anti-Fshr antibodies. However, as shown in this study, this is not the case, because Fshr is widely expressed, as demonstrated in our work.

Furthermore, our findings are supported by a report of the failure of normal Leydig cell development resulting from the deficiency of *Fshr* but not Fsh-beta (*Baker et al., 2003*) and scRNA-seq studies in Hu sheep (originated from Mongolian sheep) (*Su et al., 2023*) and in human as shown in *Appendix 1—figure 3* (DISCO, https://immunesinglecell.org/genepage/FSH) and BioGPS for human cells (http://biogps.org/#goto=genereport&id=2492). Fshr expression in Leydig cells strongly indicates that Fsh-Fshr plays a role in the production of steroids in males. Unexpectedly, Leydig cell line TM3 expresses much lower Fshr, compared to the Leydig cells in vivo (*Figure 2B–G*). It may provide an evidence on a reason that these cells do not respond well to FSH treatments (*Mather, 1980*), and indicates that this cell line may not be a typical Leydig cell population and new Leydig cell lines should be established in the future.

In the ovary, *Fshr* is highly expressed in follicles at different stages, from primordial cells, primary follicles, and secondary follicles to mature/Graafian follicles and the corpus luteum. In addition to granulosa cells, Fshr expression was observed in oocytes of follicles. Taken together, these data indicate that Fshr plays an intragonadal role in the ovary and testis beyond the granulosa and Sertoli cells.

In the skeletal system, we observed *Fshr* expression in vivo not only in osteoclasts, as reported previously (*Sun et al., 2006*; *Zhu et al., 2012a*; *Zhu et al., 2012c*), but also, interestingly, in osteoblast lineage cells, such as osteoblasts, bone lining cells, osteocytes, and progenitor cells of the periosteum, as well as in chondrocytes. In previous reports, *Fshr* expression in osteoclasts was detected only by RT-PCR, western blot, and immunostaining in cultures of primary murine precursors (*Sun et al., 2006*; *Zhu et al., 2012a*; *Zhu et al., 2012c*). Using the Fshr-ZsGreen reporter line, we visualized *Fshr* expression in multinucleated osteoclasts in frozen sections, clearly demonstrating *Fshr* expression in osteoclasts. Intriguingly, this powerful tool enabled us to examine *Fshr* expression in other cell types in bone. Surprisingly, we observed Fshr expression in cells of the osteoblast lineage, from osteoprogenitor cells and osteoblasts to osteocytes and bone lining cells. This finding indicates that Fsh may regulate not only osteoclast-mediated bone resorption but also osteoblasts for bone formation. To functionally prove the presence of Fshr in osteoblasts/osteocytes, we also deleted Fshr in osteocytes in an inducible model. The Fshr cKO induced in osteocytes significantly reduce Fshr expression and triggered an increase in the cortical thickness (*Figure 3D*) and a much more profound increase in bone mass and a decrease in fat mass than blockade by Fsh antibodies (unpublished data), illuminating Fshr expression in the osteoblast lineage.

In addition to its expression in the reproductive system and skeletal system, we also strikingly identified other cell types that highly express *Fshr*: endothelial cells in blood vessels and epithelial cells in the lung and kidney. In every examined tissue/organ, we found that endothelial cells stained positively for CD34 lining on the arterioles had a higher expression of Fshr-ZsGreen than other cell types. This bright Fshr-ZsGreen is more obviously seen in large arteries, such as the ascending aorta and others located in the heart. Similarly, Fshr was detected in vessels in solid malignant tumors by IHC or RT-PCR (*Abudureyimu et al., 2018*). However, it was not seen in normal tissues or organs by these methods, possibly due to its rapid turnover, fast degradation, or selected antibodies. Strikingly, we detected the highest Fshr-ZsGreen expression in bronchial and bronchiole ciliated epithelial cells by both fluorescence microscopy and ddRT-PCR. It was also more highly expressed in other cell types in the lung, such as type I pneumocytes (alveolar lining cells), type II pneumocytes (great alveolar or septal cells), and gland cells. However, the role of this unexpectedly high Fshr expression in the lung remains unknown. Similarly, we also found the second highest expression of Fshr-ZsGreen in renal epithelial cells in proximal and distal convoluted tubules but weak expression in renal corpuscles. In addition, our finding of Fshr expression in adipose tissues is consistent with the observations from previous works (*Liu et al., 2017*; *Liu et al., 2015*). Recently, Fshr expression was reported to be present in β-cells of the pancreas to regulate glucose-stimulated insulin secretion, further supporting our findings of the Fshr expression pattern (*Cheng et al., 2024*).

In summary, we established and validated an Fshr-ZsGreen protein reporter in vivo that faithfully recapitulates endogenous *Fshr* expression at single-cell resolution. Our compelling findings reveal that in addition to gonadal tissues, Fshr is also highly expressed in extragonadal systems, such as the

lung, kidney, heart, and pancreas. This will provide insight to better understand the biology of the Fsh-Fshr axis and its roles in the physiology and pathology of these tissues/organs. In addition to the above described, we detected Fshr expression in cells of the teeth and brain; those findings are not presented here because of space limitations and will be published elsewhere, except that three representative areas of the brain are shown in *Figure 14*.

Although the Fshr-ZsGreen reporter line is a powerful tool for detecting the location of Fshr expression, it is limited in the definition of individual isoforms of Fshr transcripts and the detection of their turnover, which should be addressed by specific antibodies and determined by the rates of transcription and RNA degradation, including Xrn1 for 5'-to-3' degradation, exosomes for 3'-to-5' degradation, and nonsense-mediated decay (*Wada and Becskei, 2017*; *Kaverina et al., 2016*).

## Methods
### Generation of the CRISPR/Cas9-mediated Fshr-ZsGreen knockin reporter line

Fshr-P2A-ZsGreen knockin reporter mice were generated by a CRISPR/Cas9-based approach. Briefly, one sgRNA was designed by the CRISPR design tool (http://www.sanger.ac.uk/) to target the region of the stop codon in the transcript NM_013523.3 exon 10 of mouse *Fshr* (*Huhtaniemi et al., 1992*), and then was screened for on-target activity using a Universal CRISPR Activity Assay (UCATM, Biocytogen Pharmaceuticals [Beijing] Co., Ltd). The targeting vector containing P2A-ZsGreen and two homology arms of left (1378 bp) and right (1493 bp) each was used as a template to repair the DSBs generated by Cas9/sgRNA. P2A-ZsGreen was precisely inserted before the stop codon of the Fshr locus. The *T7* promoter sequence was added to the Cas9 or sgRNA template by PCR amplification in vitro. Cas9 mRNA, sgRNA, and the targeting vector were co-injected into the cytoplasm of one-cell stage fertilized C57BL/6J eggs. The injected zygotes were transferred into oviducts of Kunming pseudopregnant females to generate F0 mice. F0 mice with the expected genotype as confirmed by tail genomic DNA PCR, DNA sequencing, and Southern blotting were mated with C57BL/6J mice to establish germline transmitted F1 heterozygous mice. F1 heterozygous mice were further genotyped by tail genomic PCR, DNA sequencing, and Southern blotting. Primer sequences for genotyping F0 and F1 are described in Appendix 2.

The produced Fshr-ZsGreen (Fshr-ZsG) knockin mice were maintained as heterozygotes, and homozygotes were used for experiments. For genotyping, genomic DNA was extracted from tail tips and assayed using polymerase chain reaction (PCR) primer sets for the Fshr-ZsGreen allele. The primer sequences for genotyping are described in Appendix 2. All mice were maintained on a 12 hr light/dark cycle with food and water ad libitum. The care and treatment of animals in all procedures strictly followed the NIH Guide for the Care and Use of Laboratory Animals. The animal protocols used in this study were approved by the Shanxi Medical University IACUC committee.

### Tissue harvest and preparation

Mice were terminated by $CO_2$, and organs were harvested and fixed with 4% paraformaldehyde for 12–24 hr at 4°C. For bone samples, decalcification was performed with daily changes of EDTA solution (0.5 M, pH 8) for 7 days. After fixation and decalcification processes, samples were transferred to 15% and 30% sucrose overnight, respectively. The tissues were then embedded with optimal cutting temperature compound and stored at –80°C. The embedded samples were sectioned into 5- to 25-µm-thick sections using a Leica cryostat CM1950.

### IF staining

IF staining was performed as described previously (*Chai et al., 2023*). Briefly, air-dried 7- to 25-µm-thick frozen sections were washed with 1× PBS for 10 min three to five times, followed by blocking with 10% BSA for 20 min at room temperature, and the samples were permeabilized with 0.5% Triton X-100 in 1× PBS for 10 min. Then, diluted primary antibodies were added to the slides. After overnight incubation with antibodies at 4°C, the slides were washed with 1× PBS for 15 min three to five times and then incubated with secondary antibody conjugated with fluorescence for 30 min at room temperature. The slides were washed again thoroughly with 1× PBS for 10 min three times. The slides were then stained with DAPI, rinsed with 1× PBS three times for 5 min each, and mounted

with 50% glycerol for confocal imaging. Rabbit IgG was used as the negative control. Primary and secondary antibodies were purchased from Servicebio (Wuhan Servicebio Technology Co., Wuhan, China), except when indicated otherwise. The primary antibodies are listed in Appendix 3. In addition, sections of different tissues/organs derived from C57BL/6J (B6) mice were used as negative controls for Fshr-ZsGreen expression. A donkey anti-rabbit IgG conjugated with Cy3 was employed at a 1:400 dilution as the secondary antibody (Servicebio, Wuhan, China).

## RNA-smFISH

RNA in situ hybridization, the gold standard method for visualizing RNA expression and localization in cells, tissue sections, and whole organs, was performed on tissue sections as described previously (*Young et al., 2020*). Briefly, frozen tissue sections of the testis from Fshr-ZsGreen mice were washed with 1× PBS three times. The sections were then treated with proteinase K (20 µg/mL) for 5 min at 37°C to permeabilize the cells and allow probe penetration, followed by three washes with 1× PBS. Specific oligonucleotide sense and antisense DNA probes for Fshr were designed and synthesized by the manufacturer (GeneCreate Biological Engineering Co., Ltd., Wuhan, China). The probes were labeled with a fluorescent dye (e.g. Cy3) for visualization. RNA-smFISH was performed on tissue sections using a commercially available kit (e.g. SureFISH, Agilent Technologies) at 37°C for 2 hr. After hybridization, the sections were washed with SSC solution (2× SSC, 37°C for 10 min, 1× SSC, 37°C for 5 min twice, 0.5× SSC for 10 min) to remove unbound probes, and counterstained with a nuclear stain (e.g. DAPI) to visualize the tissue architecture. The sections were imaged using a fluorescence microscope equipped with appropriate filter sets. A sense probe was used as a negative control to ensure specificity and sensitivity. The sequences of sense and antisense probes are described in Appendix 3.

## Imaging

Imaging of the slides was carried out as described before (*Chai et al., 2023*). Briefly, the fluorescence images of the frozen sections were obtained using a Nikon A1 HD25 confocal microscope with a DUVB detector and plan Apo $\lambda$ 4×, plan Apo VC 20× DIC N2, plan Apo $\lambda$ 40×, and plan Apo $\lambda$ 100×C oil objectives, illuminated with a wavelength of 405, 488, or 561 nm to excite DAPI, GFP, or Cy3, respectively; detection was performed with a 425–475, 500–550, or 570–620 nm bandpass filter. To assess the number of cells in each field of view, tissue-cleared images were converted from 3D to MAIP (maximum projection of the Z-stack across the whole section). Data were acquired with NIS-Elements AR 5.20.00 64-bit software.

## ddRT-PCR

Tissues were harvested from 10-week-old B6 mice. Samples were dissected free of connective tissue and homogenized with a fast multi-sample tissue cryogenic grinder (LC-FG-96, Lichen Instrument Technology Co., Ltd., Shanghai, China), and total RNA was extracted using NucleoZOL (NucleoZOL; Macherey-Nagel GmbH & Co., KG, Dylan, Germany). mRNA was reverse-transcribed using an M5 Super qPCR RT kit with gDNA remover (MF012-T, Mei5bio, Beijing). Droplet digital PCR was performed as described previously (*Chai et al., 2023*). The primers used for ddRT-PCR were as follows: mFshr Fwd-5'-ccgcagggacttcttcgtcc-3'; mFshr Rev-5'-ttggtgactctgggagccga-3'.

## Cell cultures of Leydig cell line TM3

TM3 cells were purchased from Pricella (Wuhan Pricella Biotechnology Co., Ltd., Wuhan, China) and subcultured with DMEM/F12+5% HS+2.5% FBS+1% P/S with the company's instruction in T25 flasks for extraction of total RNA and on coverslips for IF staining of Fshr expression.

## Acknowledgements

This study was supported by a fund from Shanxi Medical University to PL.

## Additional information

### Competing interests

Peng Liu: Reviewing editor, *eLife*. The other authors declare that no competing interests exist.

## Funding

| Funder | Grant reference number | Author |
|---|---|---|
| Shanxi Medical University | | Peng Liu |

The funders had no role in study design, data collection and interpretation, or the decision to submit the work for publication.

## Author contributions

Hongqian Chen, Data curation, Formal analysis, Validation, Investigation, Visualization; Hui-Qing Fang, Data curation, Formal analysis, Investigation; Jin-Tao Liu, Shi-Yu Chang, Formal analysis; Li-Ben Cheng, Ming-Xin Sun, Jian-Rui Feng, Ze-Min Liu, Yong-Hong Zhang, Resources; Clifford J Rosen, Writing - original draft; Peng Liu, Conceptualization, Resources, Data curation, Formal analysis, Supervision, Funding acquisition, Validation, Investigation, Visualization, Methodology, Writing - original draft, Project administration, Writing - review and editing

## Author ORCIDs

Ze-Min Liu ![ORCID] https://orcid.org/0000-0003-4810-9200
Clifford J Rosen ![ORCID] https://orcid.org/0000-0003-3436-8199
Peng Liu ![ORCID] https://orcid.org/0000-0001-5607-2482

## Ethics

The care and treatment of animals in all procedures strictly followed the NIH Guide for the Care and Use of Laboratory Animals. The animal protocols used in this study were approved by the Shanxi Medical University IACUC committee.

Reviewer #1 (Public Review): https://doi.org/10.7554/eLife.93413.3.sa1
Author response https://doi.org/10.7554/eLife.93413.3.sa2

# Additional files

## Supplementary files
MDAR checklist

## Data availability

Raw dataset presented in this study were deposited into a repository Dryad, including gel images of genotyping PCR and Southern blots; confocal images of Fshr-ZsGreen expression and immunofluorescence staining with various antibodies for tissue/cell type markers of frozen sections of organs harvested from Fshr-ZsGreen reporter and B6 mice, and smFISH; and dd.rtPCR for Fshr expression of mouse organs and TM3 cells.

The following dataset was generated:

| Author(s) | Year | Dataset title | Dataset URL | Database and Identifier |
|---|---|---|---|---|
| Liu P | 2024 | Atlas of Fshr Expression from Novel Reporter Mice | https://doi.org/10.5061/dryad.1jwstqk4t | Dryad Digital Repository, 10.5061/dryad.1jwstqk4t |

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

## Appendix 1

**-No non-specific Fshr-ZsGreen expression in the examined tissues/organs from B6 mice**

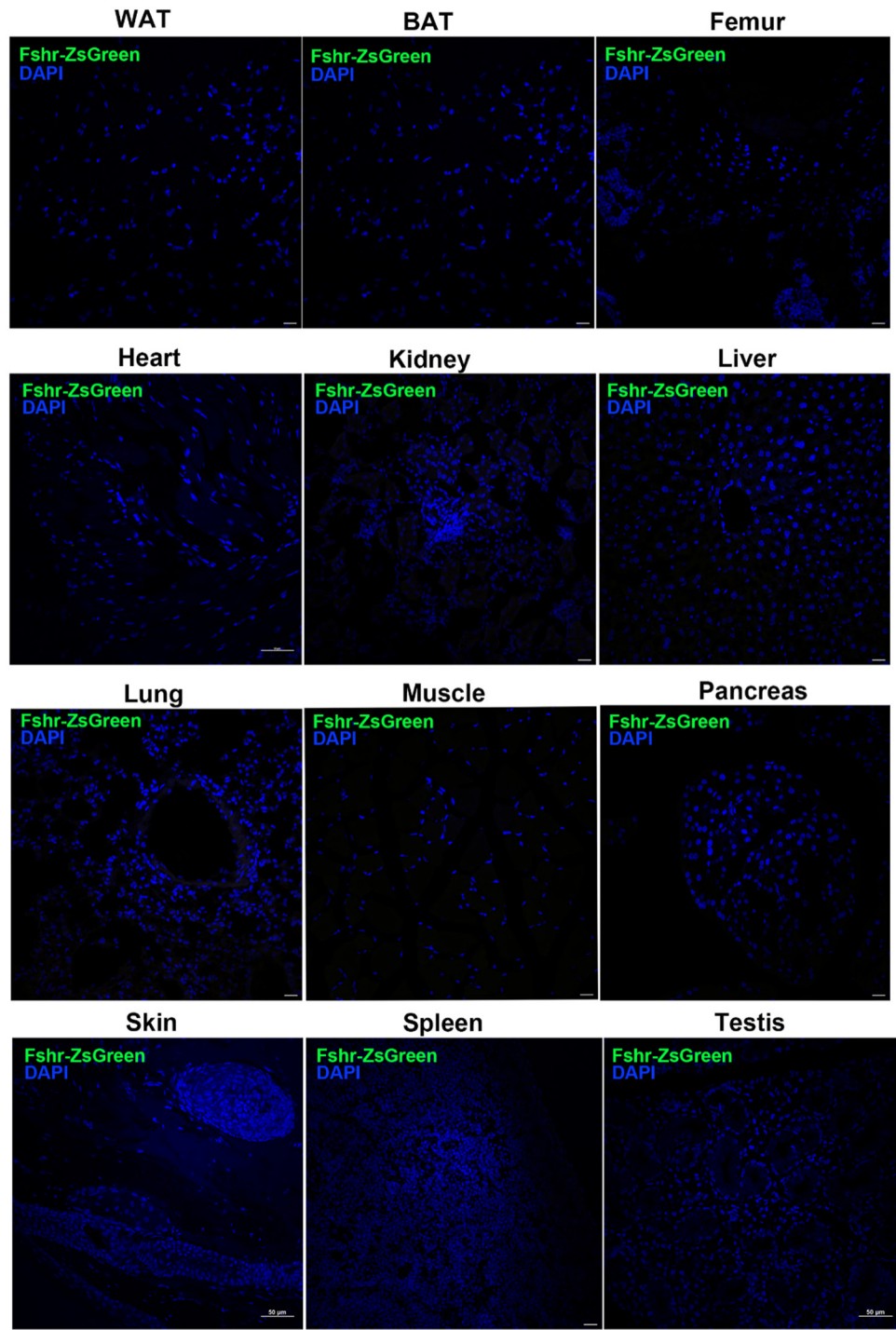

**Magnification: 400X. Scale Bar: 50μM.**

**Appendix 1—figure 1.** No non-specific Fshr-ZsGreen expression in the examined tissues/organs from B6 mice.

## Negative controls for IFs with anti-Fshr antibody

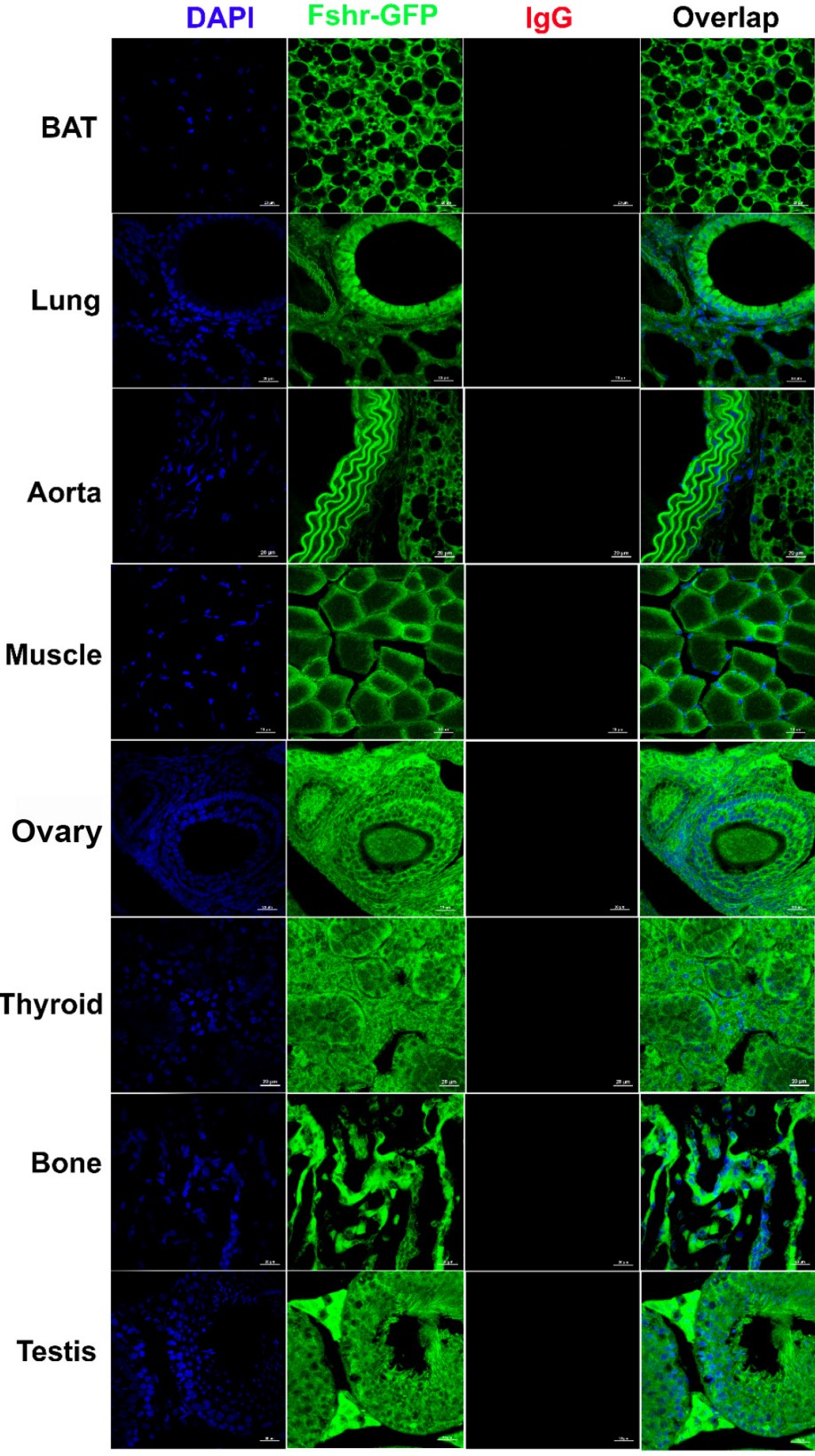

**Magnification: 400X. Scale Bar: 50μM.**

**Appendix 1—figure 2.** Negative controls for IFs with anti-Fshr antibody.

## Liver

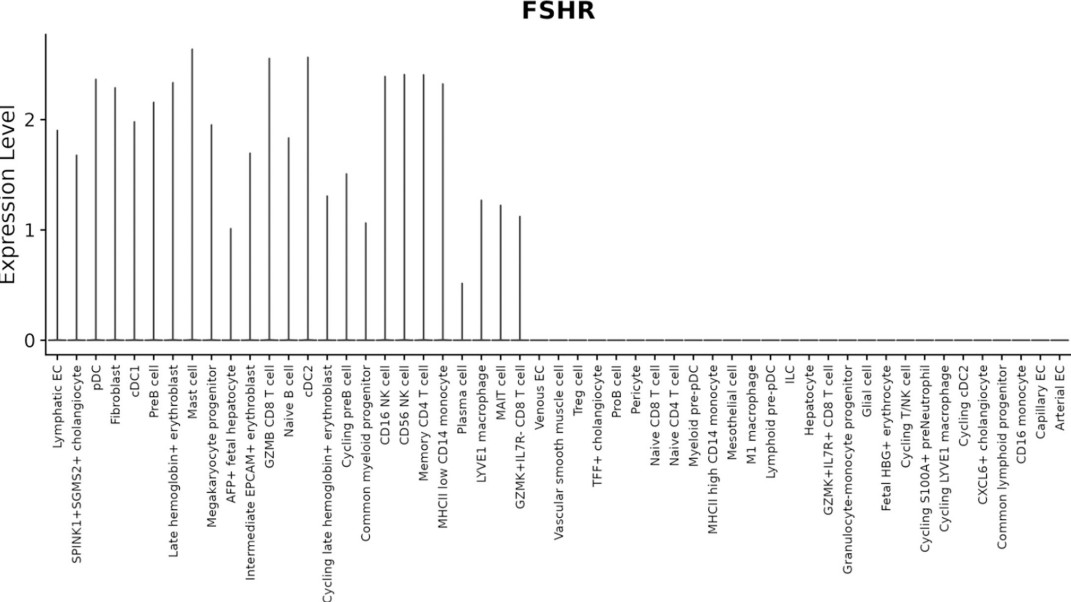

## Lung

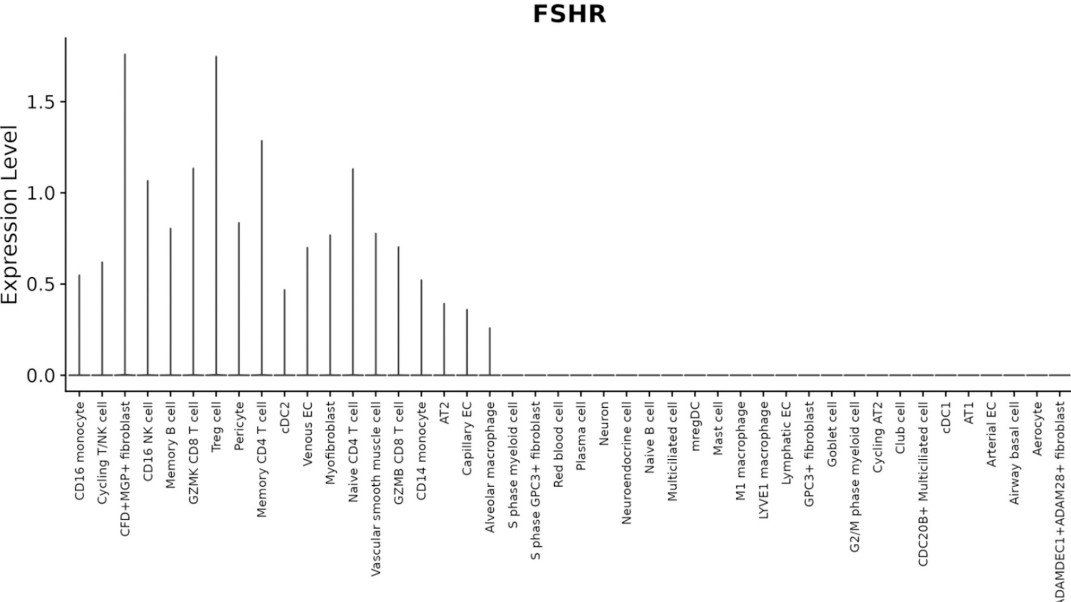

**Appendix 1—figure 3.** FSHR expression in the liver and lung detected by scRNA-seq from DISCO (https://immunesinglecell.org/genepage/FSHR).

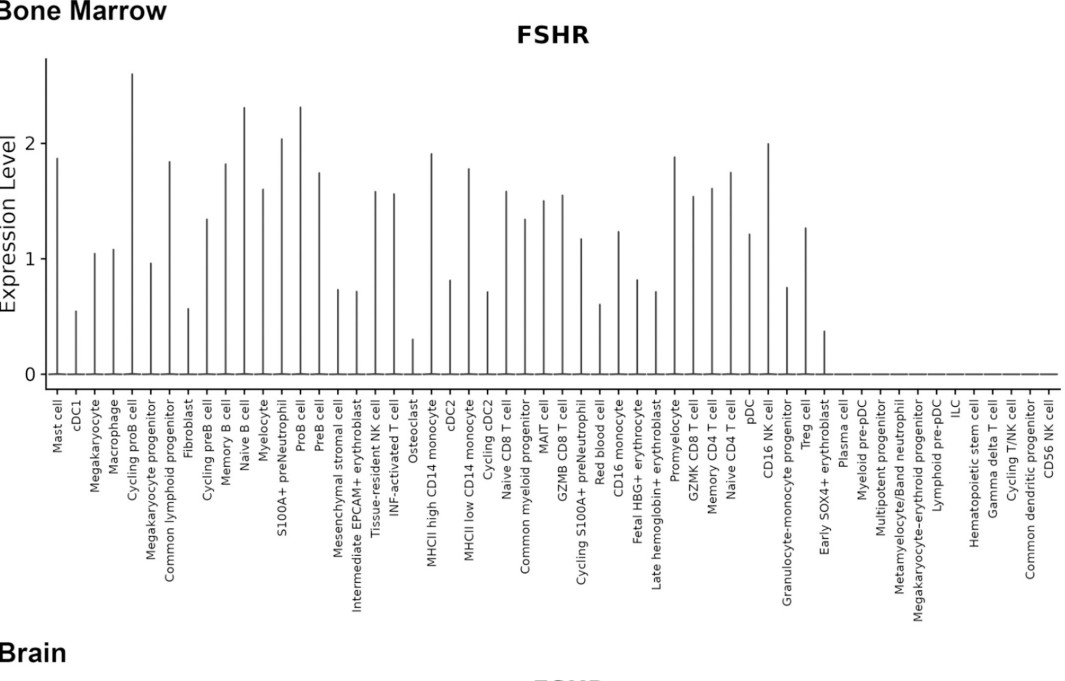

**Appendix 1—figure 4.** FSHR expression in bone marrow and heasrt detected by scRNA-seq detected by scRNA-seq from DISCO (https://immunesinglecell.org/genepage/FSHR).

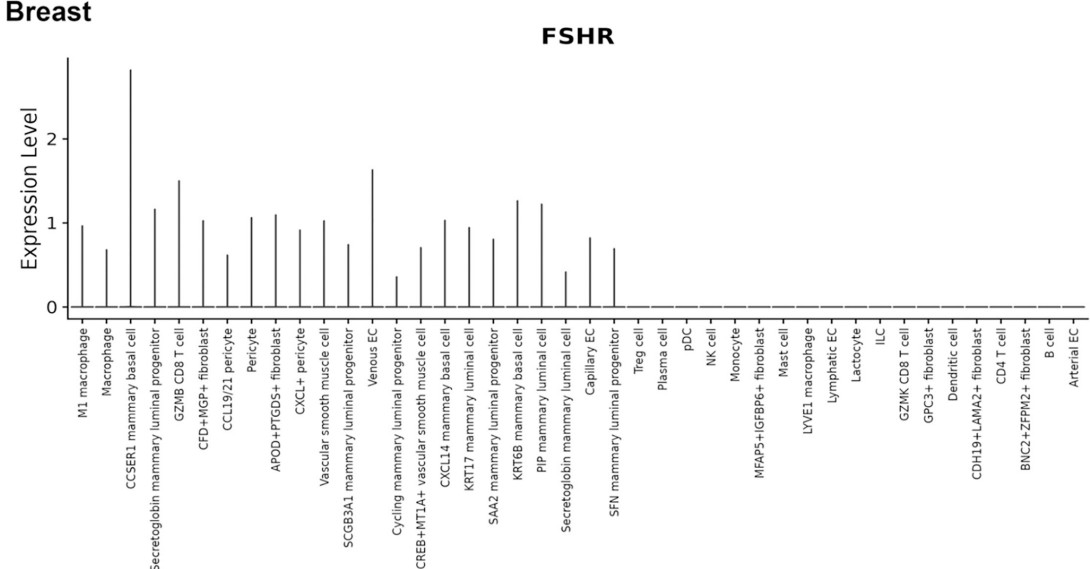

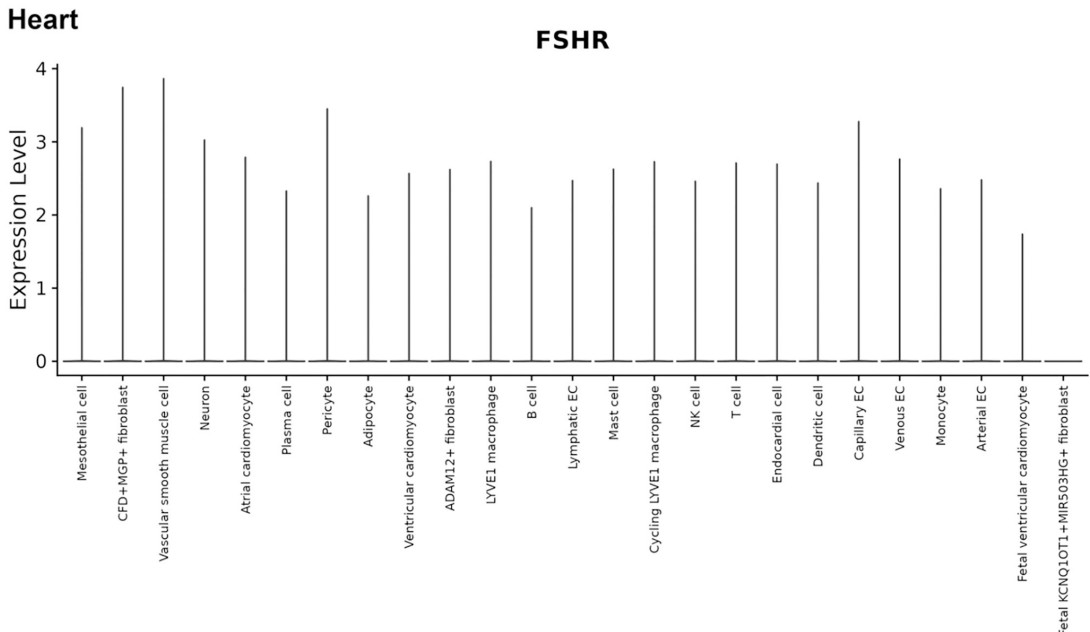

**Appendix 1—figure 5.** FSHR expression in the breast and heart detected by scRNA-seq detected by scRNA-seq from DISCO (https://immunesinglecell.org/genepage/FSHR).

## Skeletal muscle

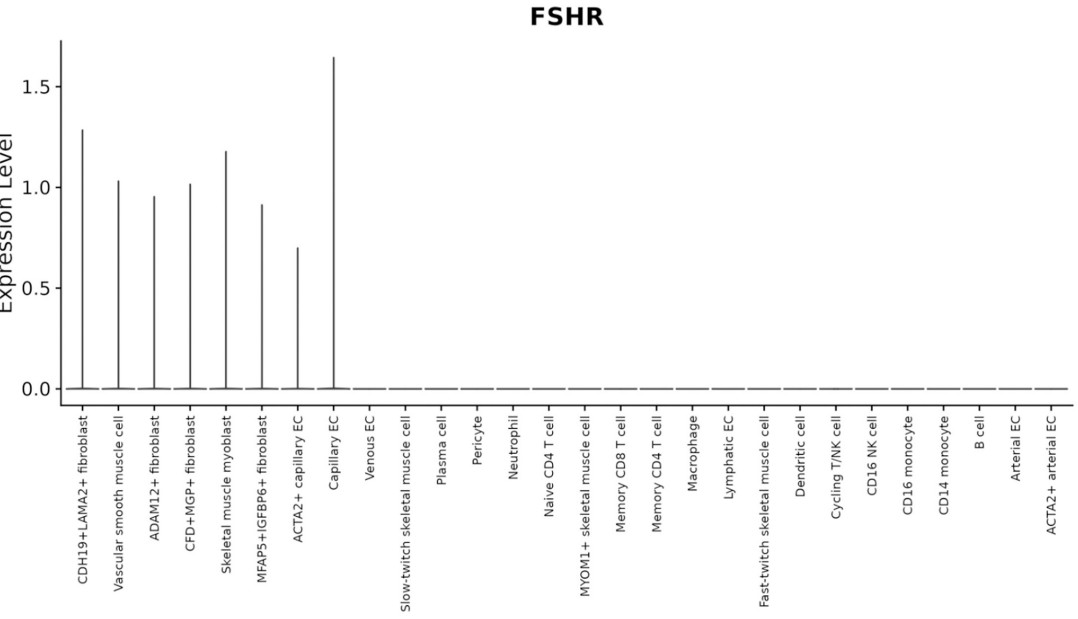

## Thymus

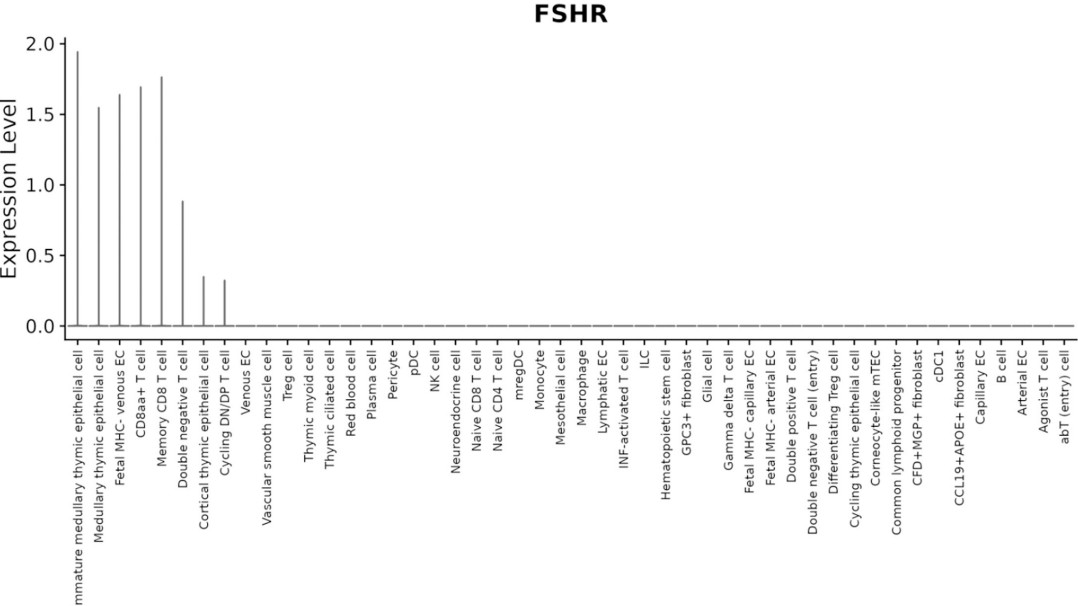

**Appendix 1—figure 6.** FSHR expression in skeletal muscle and thymus detected by scRNA-seq from DISCO (https://immunesinglecell.org/genepage/FSHR).

## Testis

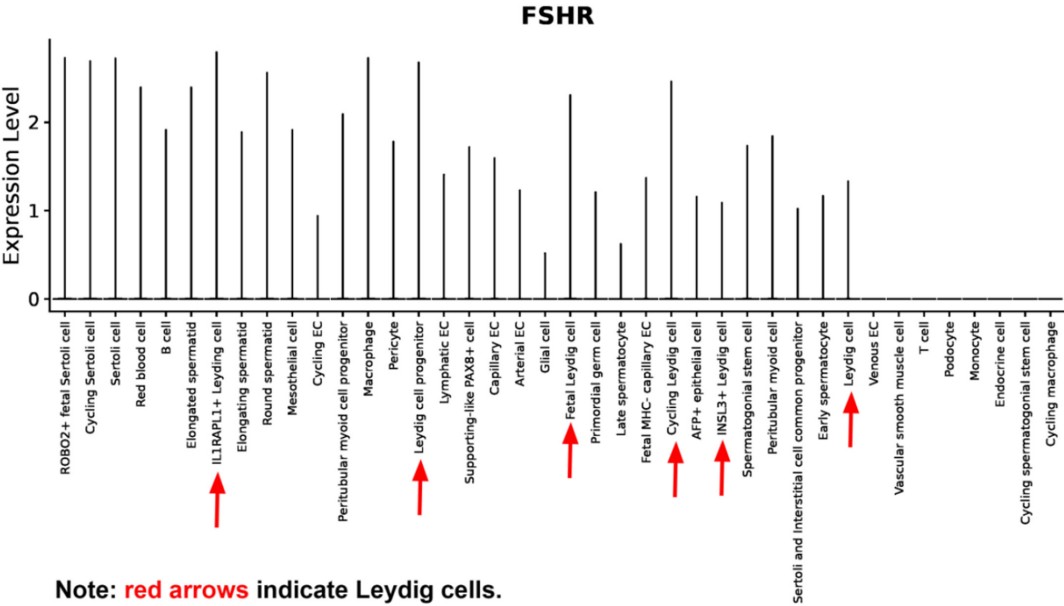

Note: **red arrows** indicate Leydig cells.

## Ovary

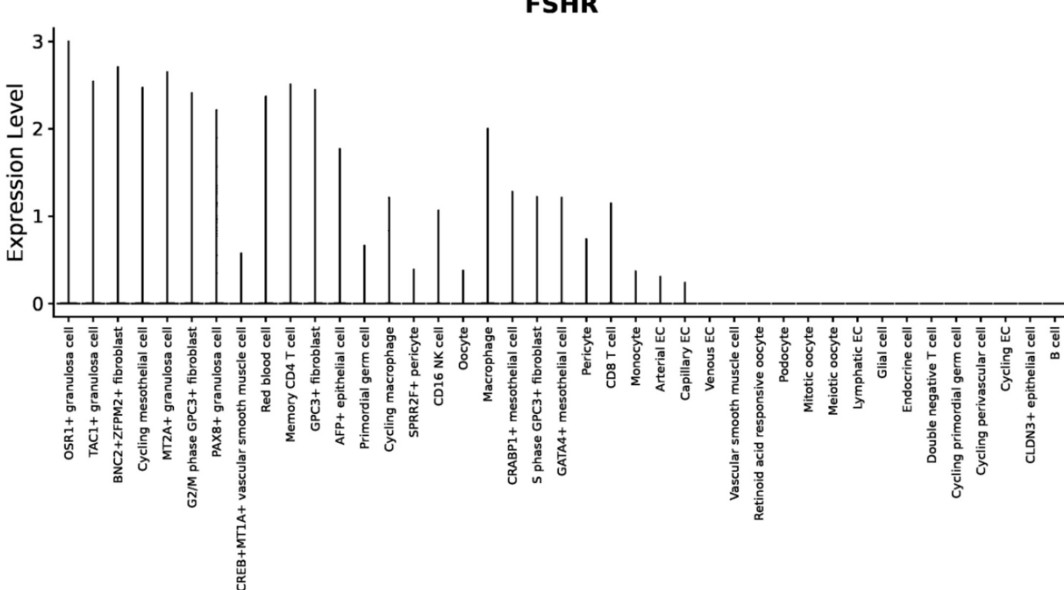

**Appendix 1—figure 7.** FSHR expression in the testis and ovary detected by scRNA-seq from DISCO (https://immunesinglecell.org/genepage/FSHR).

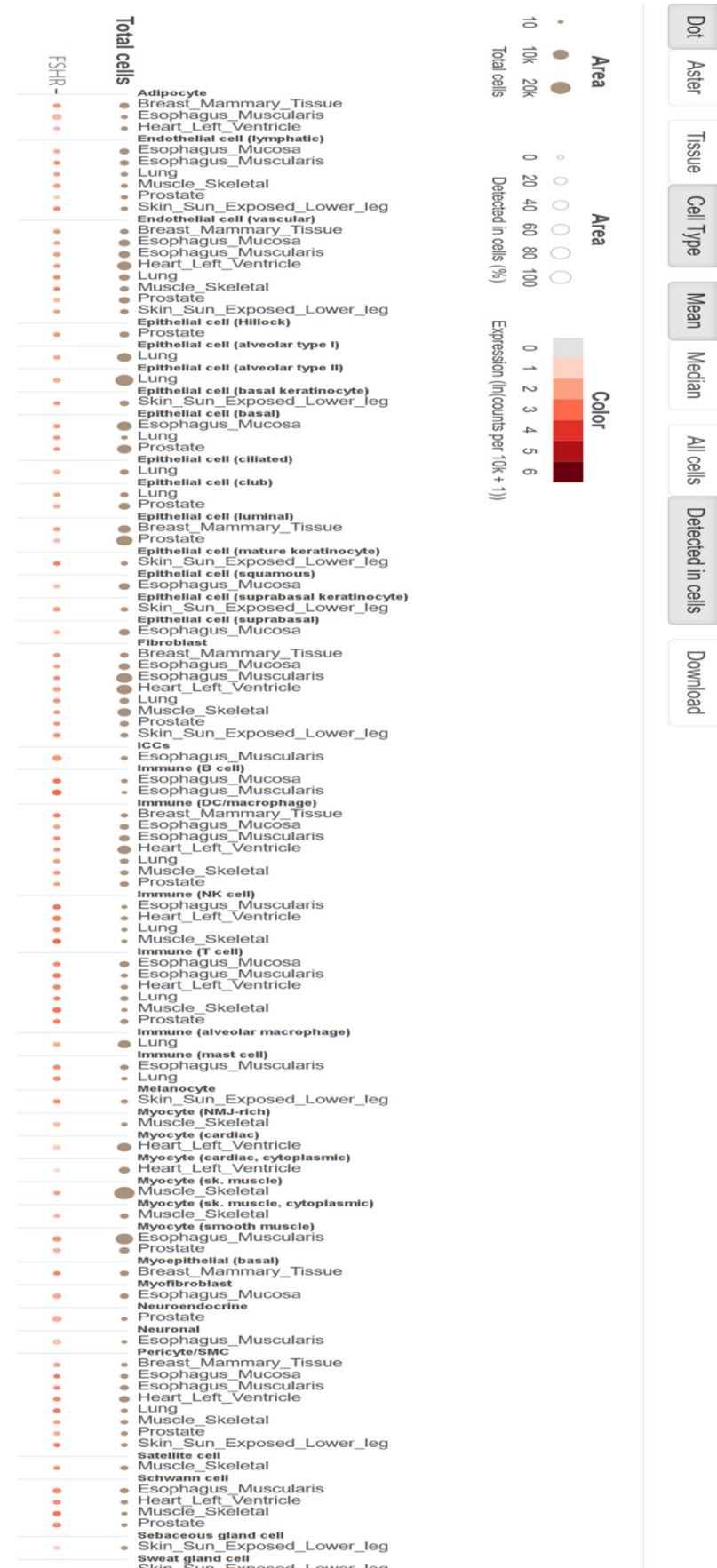

**Appendix 1—figure 8.** FSHR exxpression detected by scRNA-seq from GTEx (https://www.gtexportal.org/home/gene/FSHR#singleCell).

## Detection of FSHR Expression by scRNA-seq in Human Cells

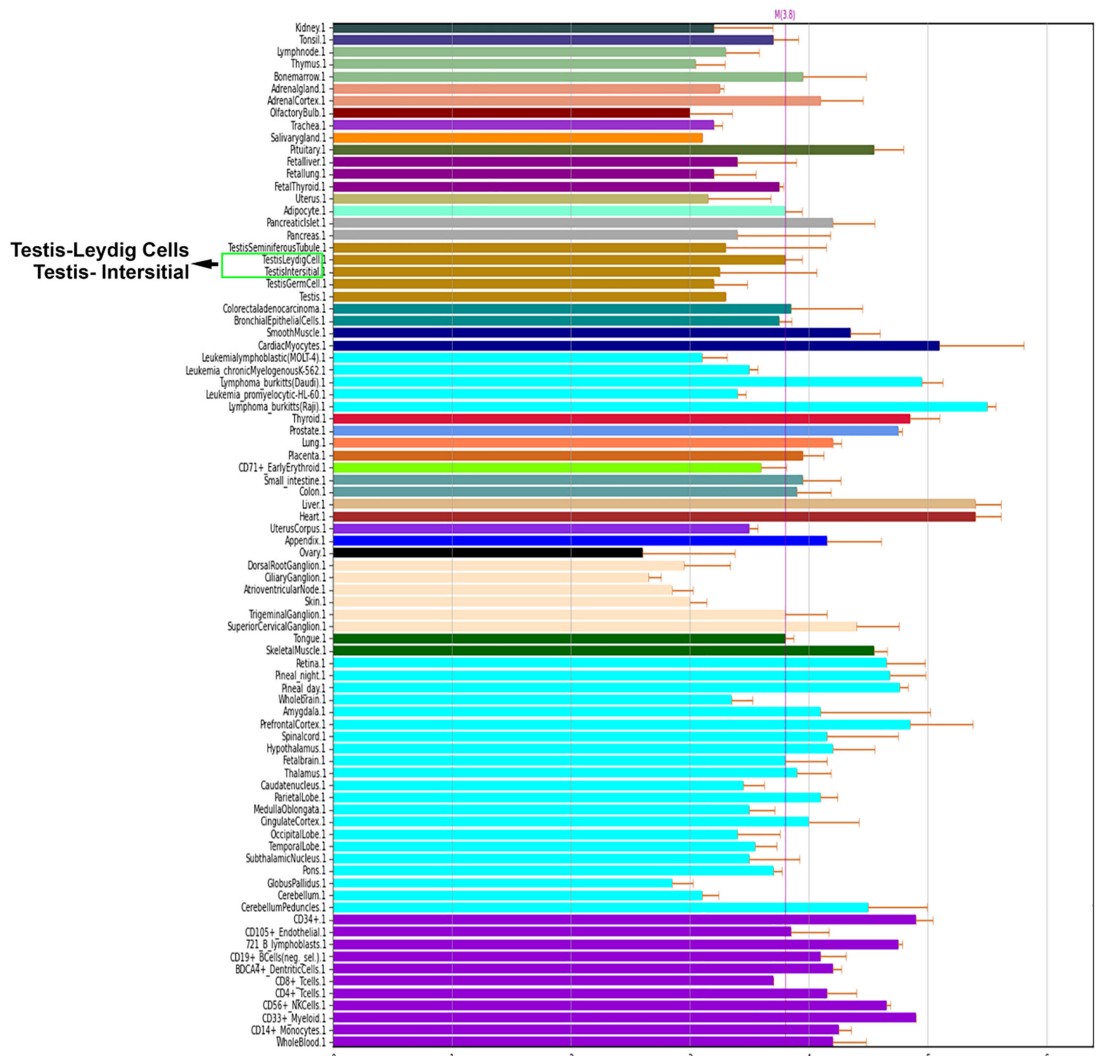

**Appendix 1—figure 9.** FSHR expression in human cells detected by scRNA-seq from BioGPS (http://biogps.org/#goto=searchresult).

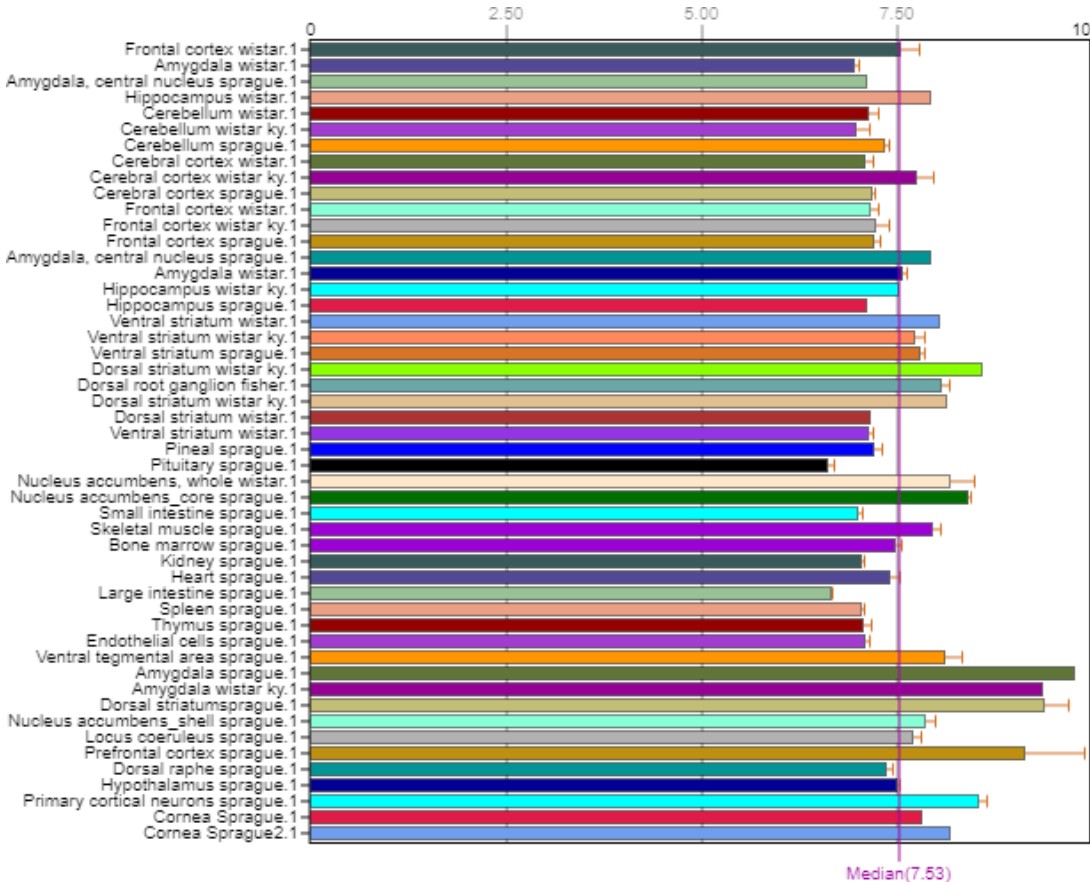

**Appendix 1—figure 10.** FSHR expression in rat cells detected by scRNA-seq from BioGPS (http://biogps.org/#goto=searchresult).

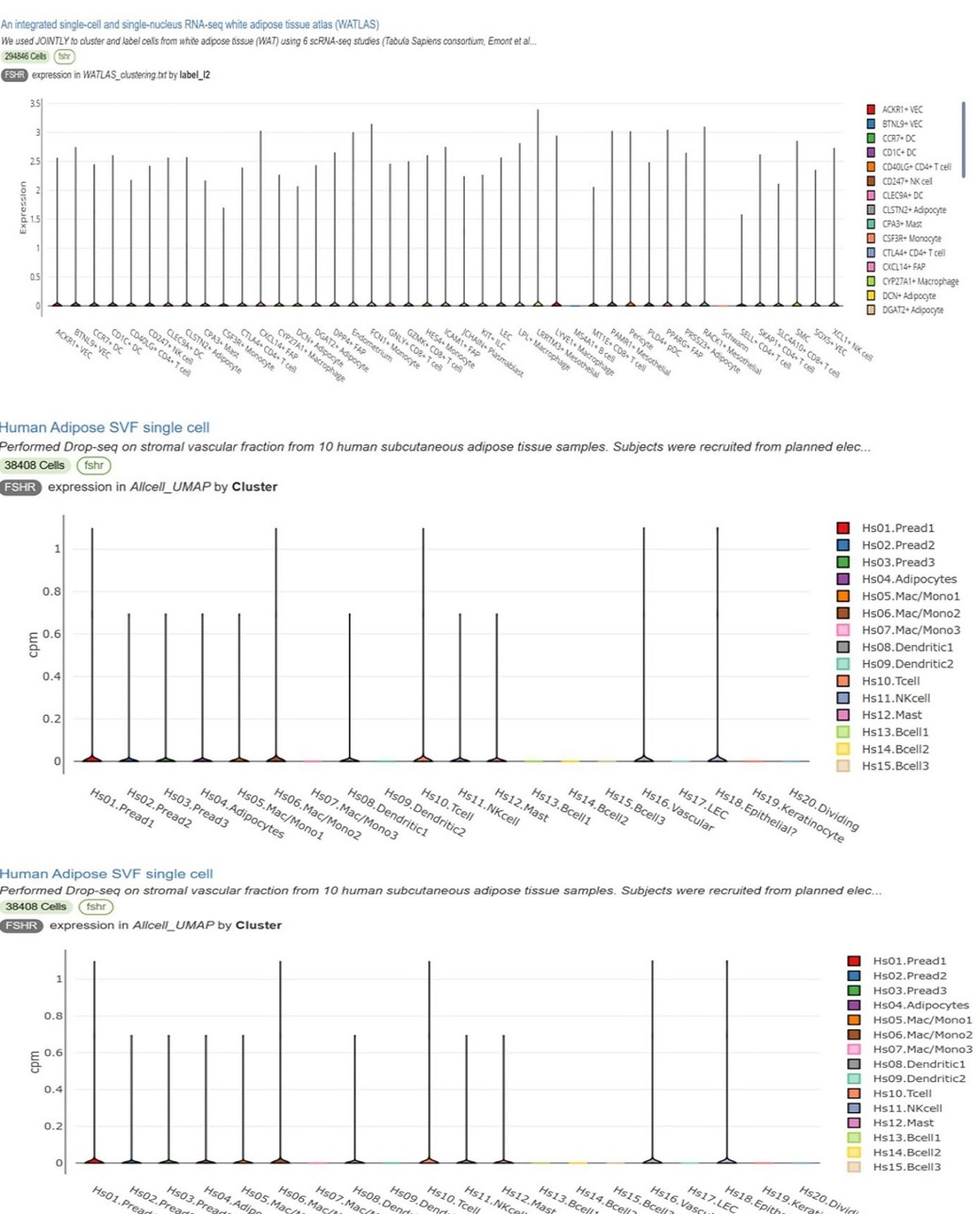

**Appendix 1—figure 11.** FSHR expression in adipose tissues detected by scRNA-seq from Single Cell Portal (https://singlecell.broadinstitute.org/single_cell).

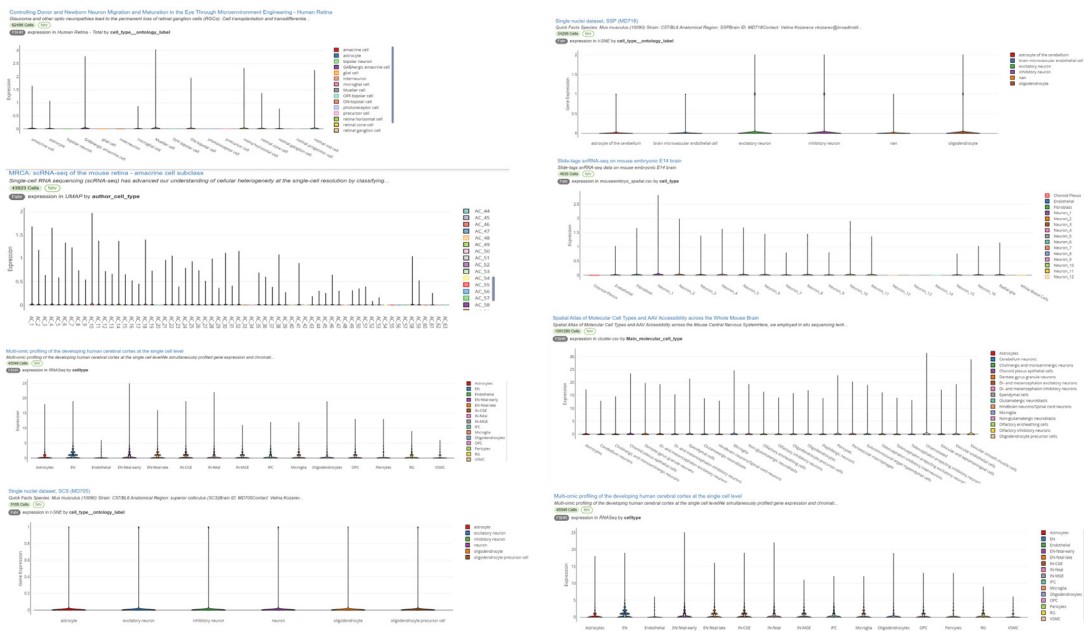

**Appendix 1—figure 12.** FSHR expression in the brain detected by scRNA-seq from Single Cell Portal.

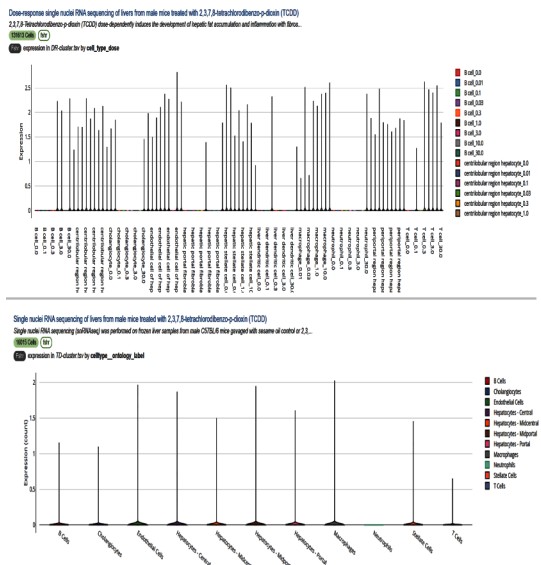

**Appendix 1—figure 13.** FSHR expression in the liver detected by scRNA-seq from Single Cell Portal (https://singlecell.broadinstitute.org/single_cell).

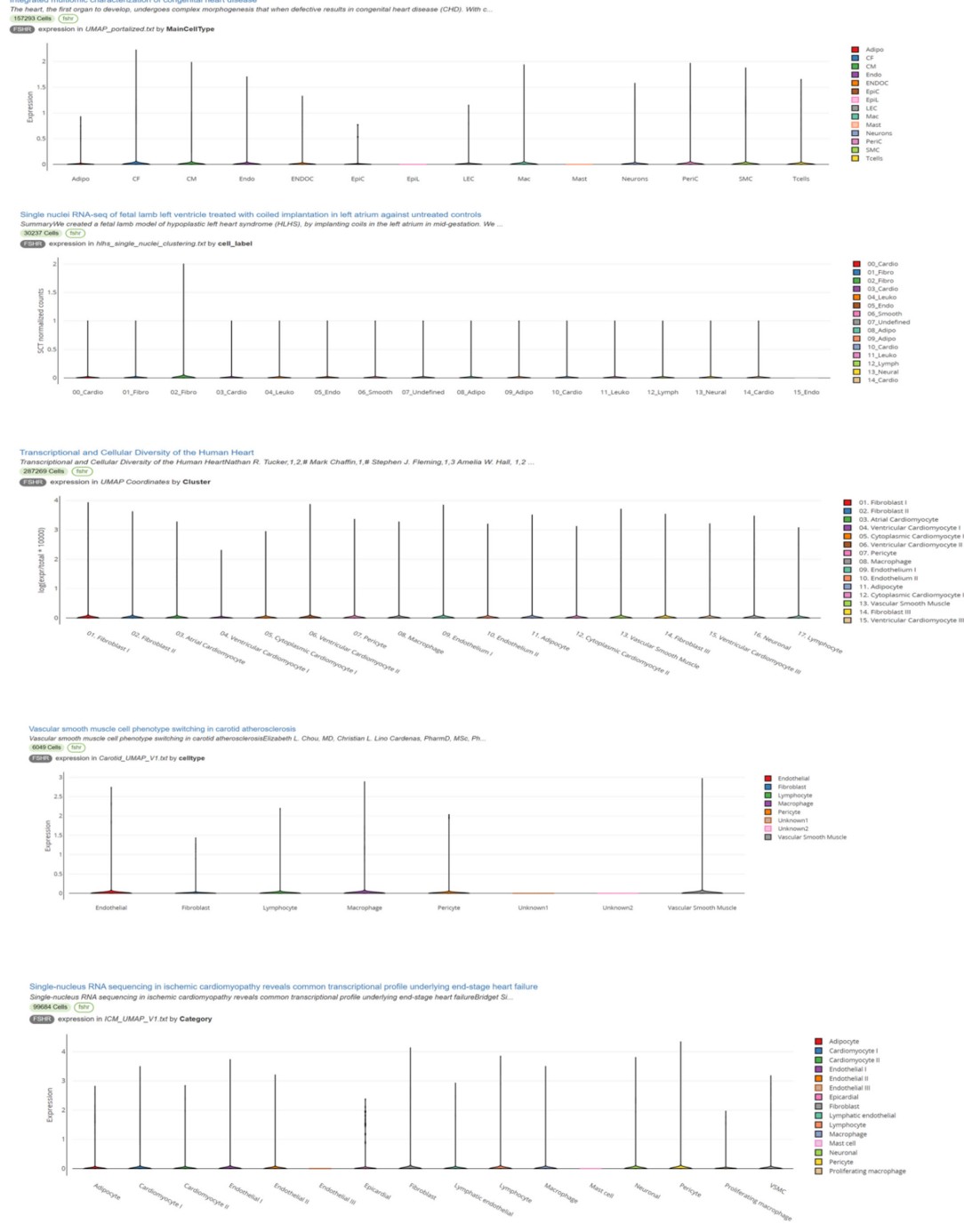

**Appendix 1—figure 14.** FSHR expression in the heart and vascular system detected by scRNA-seq from Single Cell Portal (https://singlecell.broadinstitute.org/single_cell).

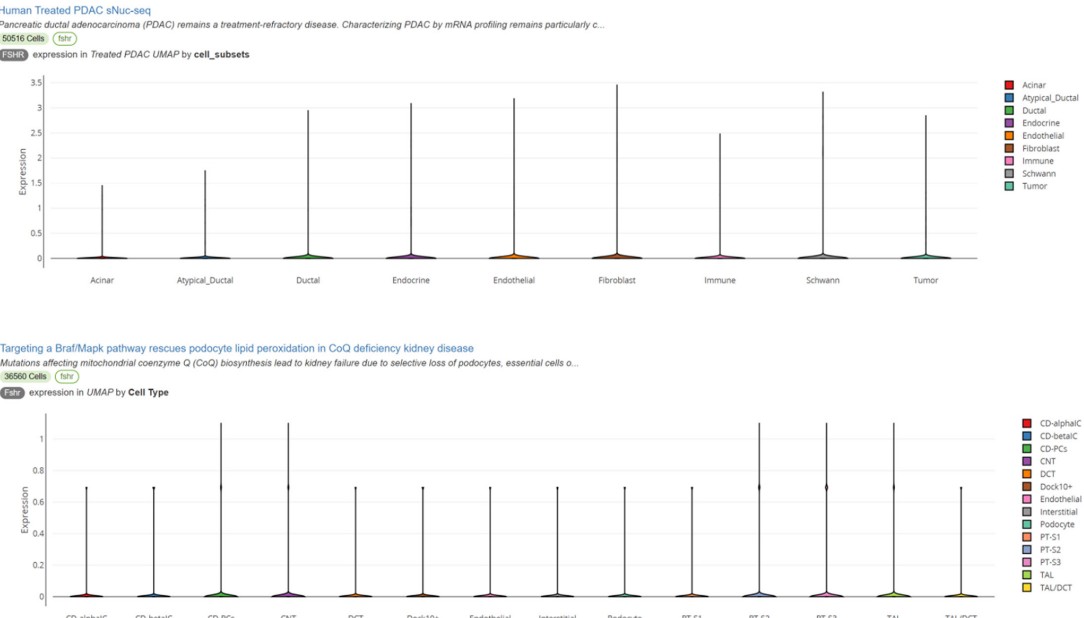

**Appendix 1—figure 15.** FSHR expression in the kidney detected by scRNA-seq from Single Cell Portal (https://singlecell.broadinstitute.org/single_cell).

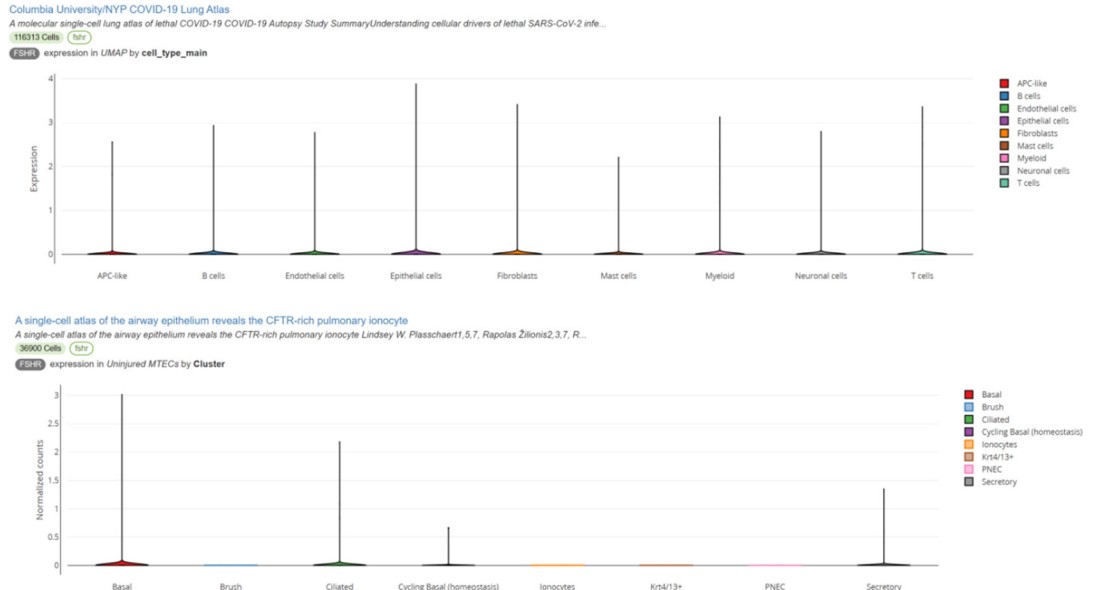

**Appendix 1—figure 16.** 16 FSHR expression in the lung detected by scRNA-seq from Single Cell Portal.

## Ovary

40dpf ovary ALL CELLS

*Single-cell transcriptome reveals insights into the development and function of the zebrafish ovary Yulong Liu, Michelle E. Kossack#, Matthew...*

25089 Cells    fshr

fshr expression in *UMAP* by **active.ident**

## Pancreas

Human Treated PDAC sNuc-seq

*Pancreatic ductal adenocarcinoma (PDAC) remains a treatment-refractory disease. Characterizing PDAC by mRNA profiling remains particularly c...*

50516 Cells    fshr

FSHR expression in *Treated PDAC UMAP* by **cell_subsets**

**Appendix 1—figure 17.** FSHR expression in the ovary and pancreas detected by scRNA-seq from Single Cell Portal.

## Appendix 2

### The first set of Integration detection PCR primer designs and sequences

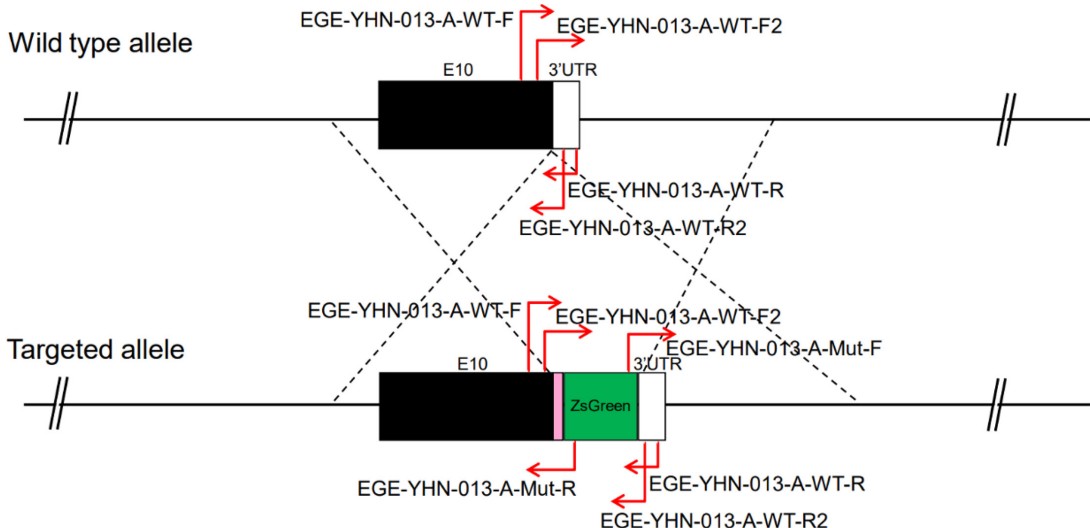

**Appendix 2—figure 1.** First set of Integration detection PCR primer designs.

**Appendix 2—table 1.** Primer sequences.

| Primer | Sequence (5'–3') | Tm(°C) | Product size (bp) |
|---|---|---|---|
| EGE-YHN-013-A-WT-F2 | ACTTCCACTCCAGAAAGAATCCCT | 58 | WT:166Mut:937 |
| EGE-YHN-013-A-WT-R2 | CTCTTCCATAGCCCTCTCTCCAG | 58 | |
| EGE-YHN-013-A-Mut-F | CGACACCGTGTACAAGGCCAAGTC | 64 | Mut:316 |
| EGE-YHN-013-A-WT-R | TAATGGTCCCTGACCTATCTGCCAT | 61 | |
| EGE-YHN-013-A-WT-F | ATCACTGTGTCCAAGGCCAAGATCC | 63 | Mut:542 |
| EGE-YHN-013-A-Mut-R | TACATGAAGGCGGCGGACAAGATG | 63 | |

Enzyme: 2x Taq Plus Master Mix II (Dye Plus)

Program: 2x Taq Plus Master Mix II (Dye Plus) progress

| | | |
|---|---|---|
| 95 °C | 3 min | |
| 95 °C | 15 sec | |
| 62 °C | 20 sec | 32 cycles |
| 72 °C | 1 kb / min | |
| 72 °C | 10 min | |
| 4 °C | hold | |

**Appendix 2—figure 2.** The schematic for the first set of genotyping PCR primer design and conditions.

## The second set of junction PCR primer designs and sequences for genotyping F0 and F1

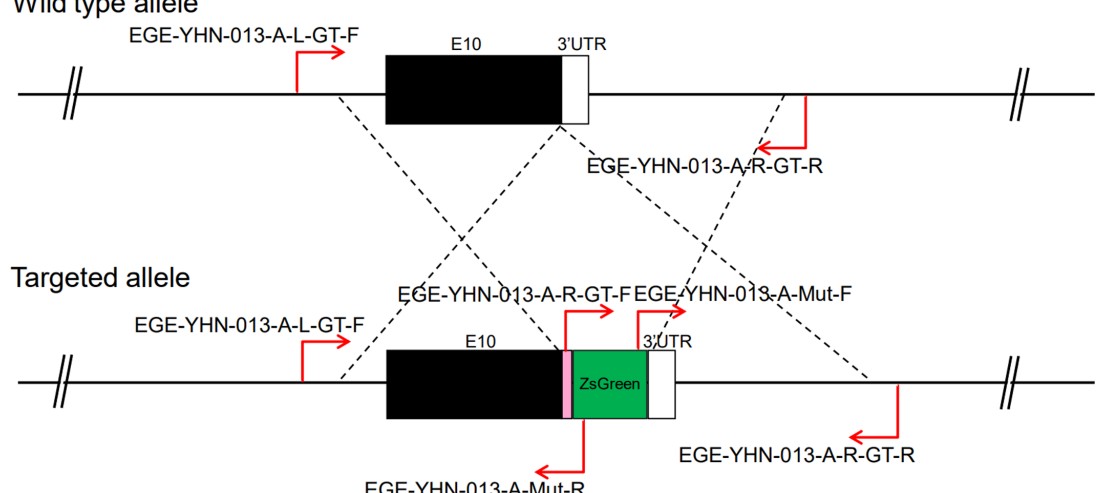

**Appendix 2—figure 3.** Second set of junction PCR primer designs.

**Appendix 2—table 2.** Sequences for genotyping F0 and F1.

| Primer | Sequence (5'–3') | Tm(°C) | Product size (bp) |
|---|---|---|---|
| EGE-YHN-013-A-L-GT-F | ACTCAGGTTGTGGCCAGATGGTTTC | 63 | Mut:2188 |
| EGE-YHN-013-A-Mut-R | TACATGAAGGCGGCGGACAAGATG | 63 | |
| EGE-YHN-013-A-Mut-F | CGACACCGTGTACAAGGCCAAGTC | 64 | Mut:2074 |
| EGE-YHN-013-A-R-GT-R | TGGCCTCACAAAGACAGCACAGATT | 62 | |
| EGE-YHN-013-A-R-GT-F | TGGAGGAGAACCCTGGACCTATGG | 62 | Mut:2636 |
| EGE-YHN-013-A-R-GT-R | TGGCCTCACAAAGACAGCACAGATT | 62 | |

Enzyme: KOD-FX

Program: Touchdown PCR

| | | |
|---|---|---|
| 94 ℃ | 2 min | |
| 98 ℃ | 10 sec | } 15 cycles |
| 67 ℃ | 30 sec (- 0.7℃/cycle) | |
| 68 ℃ | 1 kb / min | |
| 98 ℃ | 10 sec | } 25 cycles |
| 57 ℃ | 30 sec | |
| 68 ℃ | 1 kb / min | |
| 68 ℃ | 10 min | |
| 4 ℃ | hold | |

**Appendix 2—figure 4.** A schematic for the second set of genotyping PCR primer design and condition.

## PCR primer designs and sequences for genotyping heterozygotes and homozygotes

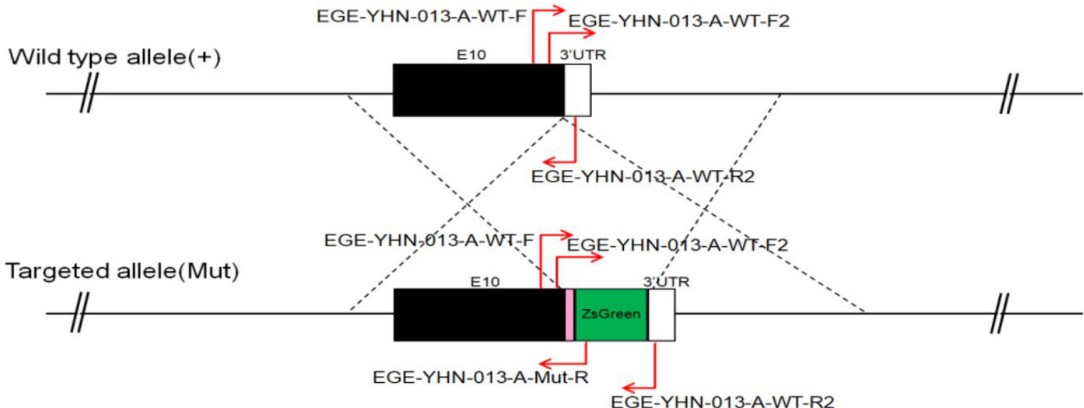

**Appendix 2—figure 5.** PCR primer designs.

**Appendix 2—table 3.** Sequences for genotyping heterozygotes and homozygotes.

| Primer | Sequence (5'–3') | Tm(°C) | Product size (bp) |
|---|---|---|---|
| EGE-YHN-013-A-WT-F2 | ACTTCCACTCCAGAAAGAATCCCT | 58 | WT:166Mut:937 |
| EGE-YHN-013-A-WT-R2 | CTCTTCCATAGCCCTCTCTCCAG | 58 | |
| EGE-YHN-013-A-WT-F | ATCACTGTGTCCAAGGCCAAGATCC | 63 | Mut:542 |
| EGE-YHN-013-A-Mut-R | TACATGAAGGCGGCGGACAAGATG | 63 | |

## Appendix 3

The sequences of sense and antisense probes for FISH in testis:

1. For the sense probe:
   5'-ATCACTGGCTGTGTCATTGCTCTAA-3'
2. For antisense probes: a mixture of the following 5 probes used for FISH.
   5'-TTAGAGCAATGACACAGCCAGTGAT-3'
   5'-CACCTTGCTATCTTGGCAGAGGAAG-3
   '5'-AGCACAAATCTCAGTTCAATGGCGT-3'
   5'-CAAGTGTTTAATGCCTGTGTTGGAT-3'
   5'-GCAGGGAATAGACCTTTGTCCTTGA-3'

**Appendix 3—table 1.** The primary antibodies used for IFs.

| # | Samples | The primary Abs.[rabbit anti-mouse (IgG)] | Sources | Cat. No. | Dilutions |
|---|---------|-------------------------------------------|---------|----------|-----------|
| 1 | WAT | Anti-Ucp1 pAb | Servicebio | GB112174 | 1:400 |
| 2 | | Anti-Ucp1 pAb | Servicebio | GB112174 | 1:400 |
| | BAT | Anti-Peripherin pAb | Servicebio | GB111635 | 1:1100 |
| | | Anti-Tyrosine hydroxylase pAb | Servicebio | GB11181 | 1:800 |
| 3 | Heart | Anti-Endomucin pAb | Servicebio | GB112648 | 1:600 |
| | | Anti--α-SMA Ab | Servicebio | GB111364 | 1:200 |
| 4 | | Anti-CD31 pAb | Servicebio | GB11063-2 | 1:100 |
| | Liver | Anti-Collagen I pAb | Servicebio | GB11022-3 | 1:800 |
| | | Anti-KCNMA1/BK channel Ab | Bioss | bs-4775R | 1:200 |
| 5 | Spleen | Anti -CD3 pAb | Servicebio | GB11014 | 1:400 |
| | | Anti -CD11b pAb | Servicebio | GB11058 | 1:500 |
| 6 | Lung | Anti-PD-L1/CD274 pAb | ABclonal | A1645 | 1:100 |
| 7 | Kidney | Anti-Laminin antibody | Bioss | bs-0821R | 1:300 |
| | | Anti-Col1a1 pAb | Servicebio | GB11022-3 | 1:800 |
| 8 | | Anti-Tyrosine Hydroxylase pAb | Servicebio | GB11181 | 1:800 |
| | Muscle | Anti-α-SMA pAb | Servicebio | GB111364 | 1:500 |
| | | Anti-CD31 pAb | Servicebio | GB11063-2 | 1:100 |
| | | Anti-PAX7 pAb | Servicebio | GB113190 | 1:2500 |
| 9 | Aorta | Anti-Endomucin pAb | Servicebio | GB112648 | 1:600 |
| | | Anti--α-SMA pAb | Servicebio | GB111364 | 1:200 |
| 10 | Testis | Anti-Stra8 pAb | Bioss | bs-1903R | 1:300 |
| 11 | | Anti-Insulin pAb | Servicebio | GB11334 | 1:400 |
| | Pancreas | Anti-Glucagon pAb | Servicebio | GB11097 | 1:800 |
| | | Anti-Neurogenin3 Ab | Bioss | bs-0922R | 1:300 |
| 12 | Brain | Anti-NeuN pAb | Servicebio | GB11138 | 1:500 |
| | | Anti-GFAP pAb | Servicebio | GB11096 | 1:800 |

*Appendix 3—table 1 Continued on next page*

*Appendix 3—table 1 Continued*

| # | Samples | The primary Abs.[rabbit anti-mouse (IgG)] | Sources | Cat. No. | Dilutions |
|---|---------|-------------------------------------------|---------|----------|-----------|
| 13 | | Anti-Osteocalcin pAb | Servicebio | GB11233 | 1:100 |
| | Femur | Anti-Trap pAb | Servicebio | GB11416 | 1:400 |
| | | Anti-Tyrosine hydroxylase pAb | Servicebio | GB11181 | 1:800 |
| 14 | Skin | Anti-CD34 pAb | Servicebio | GB111693 | 1:500 |
| 15 | Thyroid | Anti-Tshr pAb | Servicebio | GB113007 | 1:500 |
| 16 | | Anti-CD34 pAb | Servicebio | GB111693 | 1:500 |
| | Colon | Anti-CD133 pAb | Servicebio | GB113807 | 1:1000 |
| 17 | Duodenum | Anti-Taf4 pAb | Servicebio | GB111725 | 1:900 |
| 18 | | Anti-CD34 pAb | Servicebio | GB111693 | 1:500 |
| | Jejunun | Anti-CD133 pAb | Servicebio | GB113807 | 1:1000 |
| 19 | | Anti-Taf4 pAb | Servicebio | GB111725 | 1:950 |
| | Ovary | Anti-CD34 pAb | Servicebio | GB111693 | 1:500 |
| 20 | Bladder | Anti-SET pAb | Servicebio | GB113290 | 1:5000 |
| 21 | Prostate | Anti-SET pAb | Servicebio | GB113290 | 1:5000 |
| 22 | Testis | Anti-SET pAb | Servicebio | GB113290 | 1:5000 |

