## [Editor Report · eLife assessment]

These **valuable** findings develop a mouse model with trackable fusion Fshr protein, which will be of use to the field. The animal model helps to elucidate the expression and function of the FSH receptor in extra-gonadal tissues. The strength of the evidence is **solid** in most parts, although additional validation of the localization data would strengthen the study considerably.

---

## [Referee Report · Reviewer #1 (Public Review)]

The manuscript by Hong-Qian Chen and collaborators describes the development of a mouse model that co-expresses a fluorescent protein ZsGreen marker in gene fusion with the Fshr gene.

The authors are correct in that there is a lack of reliable antibodies against many of the GPCR family members. The approach is novel and interesting, with a potential to help understand the expression pattern of gonadotropin receptors. There has been a very long debate about the expression of gonadotropin receptors in other tissues other than gonads. While their expression of the Fshr in some of those tissues has been detected by a variety of methods, their physiological, or pathophysiological, function(s) remain elusive.

The authors in this manuscript assume that the expression of ZsGren and the Fshr are equal. While this is correct genetically (transcription->translation) it does not go hand in hand to other posttranslational processes.

One of the shocking observations in this manuscript is the expression of Fshr in Leydig cells. Other observations are in the osteoblasts and endothelial cells as well as epithelial cells in different organs. The expression of ZsGreen in these tissues seems high and one shall start questioning if there are other mechanisms at play here.

First, the turnover of fluorescent proteins is long, longer than 48h, which means that they accumulate at a different speed than the endogenous Fshr. This means that ZsGreen will accumulate in time while the Fshr receptor might be degraded almost immediately. This correlated with mRNA expression (by the authors) but does not with the results of other studies in single-cell sequencing (see below).

Then, the expression of ZsGreen in Leydig cells seems much higher than in Sertoli cells, this is "disturbing" to put it mildly. This is visible in both, the ZsGreen expression and the FISH assay (Fig 2 B-D).

The expression in WAT and BAT is also questionable as the expression of ZsGreen is high everywhere. What makes it difficult to actually believe that the images are truly informative? For example, the stainings of the aorta show the ZsGreen expression where elastin and collagen fibres are - these are not "cells" and therefore are not expressing ZsGreen.

FISH expression (for Fshr) in WT mice is missing.

Also, the tissue sections were stained with the IgG only (neg control) but in practice both the KI and the WT tissues should be stained with the primary and secondary antibodies.

The only control that I could think of to truly get a sense of this would be a tagged receptor (N-terminal) that could then be analysed by immunohistochemistry.

The authors also claim:

To functionally prove the presence of Fshr in osteoblasts/osteocytes, we also deleted Fshr in osteocytes using an inducible model. The conditional knockout of Fshr triggered a much more profound increase in bone mass and decrease in fat mass than blockade by Fsh antibodies (unpublished data)

This would be a good control for all their images. I think it is necessary to make the large claim of extragonadal expression, as well as intragonadal such as Leydig cells.

Claiming that the under-developed Leydig cells in FSHR KO animals is due to a direct effect of the FSHR, and not via a cross-talk between Sertoli and Leydig cells, is too much of a claim. It might be speculated to some degree but as written at the moment is suggests this is "proven".

We also do not know if this Fshr expressed is a spliced form that would also result in the expression of ZsGreen but in a non-functional Fshr, or whether the Fshr is immediately degraded after expression. The insertion of the ZsGreen might have disturbed the epigenetics, transcription or biosynthesis of the mRNA regulation.

The authors should go through single-cell data of WT mice to show the existence of the Fshr transcript(s). For example here:

https://www.nature.com/articles/sdata2018192

Comments after revision:

The response by the authors does not seem sufficient or adequate, by any length, for what one would expect for a work having such a large claim as the expression of the Fshr in multiple cell types and organs. It is not the fact that Fshr might be expressed extragonadally or even by other cells in the gonads, but the surprising images where virtually every cell in the provided tissues, and not only cells but structures, glows green.

It is not possible to know, as a reviewer, whether the excitation intensity and exposure for all images is equal. We believe that they cannot be, as control organs such as fat, testes, ovaries, and vasculature have a natural fluorescence background.

Leydig cells cannot simply express more Fshr than Sertoli cells, that would go against what we have known for >50 years in physiology. While it is scientific to question 'old' data, to make extraordinary claims there is a need for "extraordinary evidence". There is very low expression in Sertoli cells (Fig 2) while Leydig cells and spermatozoa glow vividly.

Moreover, even the tails of spermatozoa glow! This is not cytoplasm and cannot contain a soluble fluorescent protein.

The controls should be shown side-by-side to the experimental images. It would be a lot more credible if the WT and the KI tissues were placed on the same slide, with images taken from them side by side not only for ZsGreen but antibody immunofluorescence staining.

Moreover, I noticed that the entire manuscript is based on a single founder mouse, which is not acceptable as an error - either multiple integrations other than in the correct locus or genetic instability created by the KI integration would result in promiscuous expression. The founder mouse is not well enough characterised as it is only performed by Southern blots and PCR, while additional integrations cannot be detected by such. Other methods should be used such as FISH or even whole genome sequencing. Yet, several lines should be used to ensure no other effects exist.

In Fig 5, the section of aorta shows low staining in the elastin/collagen fibres, while there is clearly in Suppl Data 2. In the same figure, the 2nd lung images show green fluorescence in the mucosa (centre) which should not be as there is no cells there.

The additional single-cell data does not truly support their claims, in the sense that while some of the data might go in line e.g. Leydig cells showing as high expression as "tubules", there are many other cell types that show no expression such as hepatocytes and skeletal muscle, where the authors claim to have high expression of Fshr. Moreover, in the datasets presented organs like "ovary" have almost no Fshr expression, which should question the validity of such.

The authors use an Fshr antibody without enough validation. The Fshr KO animals should be used for this. In fact, one of the very first statements in the manuscript is that antibodies against GPCRs in general, and gonadotropin receptors more specifically, are unreliable. The fact that controls show the same pattern as transgenic animals questions the validity, as no single acceptable antibody against FSHR recognises Leydig cells.

The detection of Fshr in e.g. adipocytes of B6 mice is as questionable as many other claims of gonadotropin receptors in extragonadal tissues, which has been questioned a number of times by many researchers.

One question we should ask is, is there any tissue on these mice that does not 'express' (Fshr)-ZsGreen? Because from what I see every single tissue analysed has 'Fshr". Which might be the problem why it is so difficult to find.

Some images seem to be duplicated such as in Fig 2C where the first row and the 3rd row seem to be the same image.

---

## [Author Response]

The following is the authors’ response to the original reviews.

We would like to thank you very much for reviewing our manuscript and express our sincere appreciation for the valuable and thoughtful comments that led us to significantly improve the manuscript on Fshr-ZsGreen reporter mice. We have seriously taken your comments to make a major revision of the manuscript, and here is a summary of the revision:

(1) New data on Fshr expression are input to the revised Manuscript:

a. Fshr expression in the testis and adipose tissues (WAT and BAT) of B6 mice;

b. Fshr expression in the testis of B6 by RNA-smFISH;

c. Comparison of Fshr expression in the testis and ovary between Fshr-ZsGreen and B8 mice by ddRT-PCR to prove Fshr expression without interruptions by insertion of P2A-ZsGreen vector;

d. Reduction of Fshr expression in osteocytes within the femoral sections from DMP1-CreERT2:Fshrfl/fl mice;

e. Fshr expression in an established Leydig cell line-TM3 by immunofluorescence and ddRT-PCR, also show Fshr located in the nuclei of TM3 cells;

f. Fshr expression at scRNA-seq level from 5 public single cell portals as Supplementary Data 3 to support our findings of the widespread expression pattern of Fshr, particularly in Leydig cells.

(2) Re-organization of Figure 2 with a new legend.

(3) A new paragraph is added to the Discussion Section of the revised MS to explain the function of P2A peptide in generation of GFP reporter mice and why Fshr express is not interrupted by the P2A-ZsGreen insertion in Fshr-ZsGreen reporter.

(4) Deletion of Figure 1-D-c, as it is not necessary.

(5) Replace of Figure 8-A (the left panel) with a reduced exposure time image.

(6) Amended parts of the revised MS are labeled in red.

A point by point response to the Reviewers’ comments:

**Reviewer 1:**
One of the shocking observations in this manuscript is the expression of FSHR in Leydig cells. Other observations are in the osteoblasts and endothelial cells as well as epithelial cells in different organs. The expression of ZsGreen in these tissues seems high and one shall start questioning if there are other mechanisms at play here.First, the turnover of fluorescent proteins is long, longer than 48h, which means that they accumulate at a different speed than the endogenous FSHR This means that ZsGreen will accumulate in time while the FSHR receptor might be degraded almost immediately. This correlated with mRNA expression (by the authors) but does not with the results of other studies in single-cell sequencing (see below).The expression of ZsGreen in Leydig cells seems much higher than in Sertoli cells, this is "disturbing" to put it mildly. This is visible in both the ZsGreen expression and the FISH assay (Figure 2 B-D).

Thank you for this valuable comments. We added new data on Fshr expression to prove the presence of Fshr in Leydig cells in B6 detected by immunofluorescence staining, RNA-smFISH and ddRT-PCR, as well as in TM3 cells-isolated Leydig cells from a male mice in the revise MS (Fig 2E, F and G), that demonstrate no interruptions of normal Fshr expression by insertion of P2A-ZsGreen vector into a locus located between exon10 and stop code. We use ZsGreen as an indicator for active Fshr promoter status, rather than a method to measure Fshr expression, which is done by ddRT-PCR. These data are shown in Figure 2G of the revised MS

In addition, we provide scRNA-seq based evidence on Fshr expression in human Leydig cells from two single cell portals (DISCO and BioGPS) as shown in Supplementary Data 3 in the revised MS. We also cited a recent report on scRNA-seq analysis of Fshr expression in Hu sheep in the revised MS as Reference 65 (PMID: 37541020) 1, which also clearly showed Fshr expression in Leydig cells at single cell level in Hu Sheep.

We believe that the lack of Fshr expression in some single cell databases may be due to the degradation of Fshr transcript in cells during the process of single cell populations. In our laboratory, we spent more than 6 months to optimize methods and reagents to perverse mRNA integrity more than 8 for RAN-seq.

The expression in WAT and BAT is also questionable as the expression of ZsGreen is high everywhere. That makes it difficult to believe that the images are truly informative. For example, the stainings of aorta show the ZsGreen expression where elastin and collagen fibres are - these are not "cells" and therefore are not expressing ZsGreen.FISH expression (for FSHR) in WT mice is missing.Also, the tissue sections were stained with the IgG only (neg control) but in practice both the KI and the WT tissues should be stained with the primary and secondary antibodies. The only control that I could think of to truly get a sense of this would be a tagged receptor (N-terminal) that could then be analysed by immunohistochemistry.

Reply 2 and 3: Thank you for these comments. New data on Fshr expression in WAT and BAT of B6 mice by immunofluorescence staining and in the testis of B6 mice by immunofluorescence staining and RNA-smFISH are added to the revised MS (Fig.2D and E, and Fig. 4G), showing similar patterns to that of Fshr-ZsGreen mice. Furthermore, we provide more evidences as Supplementary Data 3 on Fshr expression obtained from 4 public single cell portables, showing FSHR expression in a widespread organs and tissues (including different fractions of adipose cells) of human, mice and rat at single cell levels. Please also check Fshr expression pattern in adipose tissues by immunostaining for Fshr in previous reports (Fig. 3a of PMID: 28538730 and Fig. 2 of PMID: 25754247) 2 3, which showed a similar expression pattern to our finding. These data should address your concerns on Fshr expression in WAT and BAT and other organs/tissues.

Regard of “For example, the stainings of aorta show the ZsGreen expression where elastin and collagen fibres are - these are not "cells" and therefore are not expressing ZsGreen.” We believe that you referred to the image of the aorta in Supplementary Data2. However, Please take a look at the images of the aorta in Figure 5-C, which shows positively stained the layer of ‘elastin and collagen fibres’ for EMCN and a-SMA colocalized with Fshr expression with stained DAPI at a 1000X magnification, indicating endothelial cells and the cellular membrane presented in this layer, not just ‘elastin and collagen’.

The authors also claim:To functionally prove the presence of FSHR in osteoblasts/osteocytes, we also deleted FSHR in osteocytes using an inducible model. The conditional knockout of FSHR triggered a much more profound increase in bone mass and decrease in fat mass than blockade by FSHR antibodies (unpublished data).This would be a good control for all their images. I think it is necessary to make the large claim of extragonadal expression, as well as intragonadal such as Leydig cells.

Thank you for this very encouraging comment. As you suggested, we did add a result of reduced Fshr expression in osteocytes from DMP1-CreERT2+:Fshrfl/fl mice treated with tamoxifen to the revise MS, as shown in Figure 3D, demonstrating Fshr present in osteocytes and the specificity of Fshr antibody. Furthermore, we incorporated your advice on making ‘ large claim of extrogonadal and intragonadal expression of Fshr’ into the revised MS in red.

Claiming that the under-developed Leydig cells in FSHR KO animals are due to a direct effect of the FSHR, and not via a cross-talk between Sertoli and Leydig cells, is too much of a claim. It might be speculated to some degree but as written at the moment it suggests this is "proven".

Thank you for pointing out this incorrect claim and we apologized for it. In the revised MS, we deleted this claim.

We also do not know if this FSHR expressed is a spliced form that would also result in the expression of ZsGreen but in a non-functional FSHR, or whether the FSHR is immediately degraded after expression. The insertion of the ZsGreen might have disturbed the epigenetics, transcription, or biosynthesis of the mRNA regulation.

Thanks for this comment. In the revised MS, we added a new section to explain the function of P2A peptide in generation of a GFP reporter by sgRNA-guilded site specific knockin of P2A ZsGreen vector through CRISPRA/cas9 and provided a new result on comparison of Fshr expression in the testes and ovaries from Fshr-ZsGreen and B6 mice, showing equivalent Fshr expression between Fshr-ZsGreen and B6 mice (Figure 2G), which indicates no interruptions of Fshr expression by the insertion of P2A vector.

The authors should go through single-cell data of WT mice to show the existence of the FSHR transcript(s).For example here:
https://www.nature.com/articles/sdata2018192

Thank you so much for the valuable comment. Yes, we took you critical advice to check Fshr expression through 4 single cell portals, including DISCO, GTEx, BioGPS and Human single cell portal, and present the collected data as Supplementary Data 3 in the revised MS, that strongly support our findings of the wider Fshr expression. Particularly, Fshr expression in Leydig cells is proved by scRNA-seq studies of human cells from DISCO and BioGPS, as well as a recent study in Hu sheep (PMID: 37541020) 1 and we cited it in the revised MS.

**Reviewer 2:**
Is the FSHR expression pattern affected by the knockin mice (no side-by-side comparison between wt and GSGreen mice, using in situ hybridization and ddRTPCR, at least in the gonads, is provided)?

Thanks for the comment. In the revised MS, we provided a set of new data on Fshr expression in the testis, ovary, WAT and BAT of B6 mice by immunofluorescence staining and by RNA-smFISH for Fshr expression, showing similar expression patterns. Additionally, we also performed ddRT-PCT to compare Fshr expression in the testes and ovaries between Fshr-ZsGreen and B6 mice, demonstrating equivalent expression of Fshr expression between Fshr-ZsGreen and B6 mice. Interestingly, we also observed an significantly higher Fshr expression in the testis than that in the ovary (more than 30 folds).

Is the splicing pattern of the FSHR affected in the knockin compared to wt mice, at least in the gonads?

Thanks for the question. Please see our reply to the Reviewer 1 for the function of P2A peptide used for generation of GFP reporters. Although we didn’t directly assess the splicing pattern, we provide a result of comparison of Fshr expression in Figure 2F in the revised MS, indirectly showing no changes of the splicing pattern. We will assess the splicing pattern of Fshr in the future that has been neglected in the field.

Are there any additional off-target insertions of GSGreen in these mice?” and “Are similar results observed in separate founder mice?

Thanks for the questions. As we describe it in the method section in detail in the MS, Fshr-ZsGreen reporter was produced by the a site-specific long ssDNA recombination of the P2A-ZsGreen targeting vector to the locus between Exon10 and stop code by CRIPRA/cas9, which was guided by site-specific single guide RNA (sgRNA). We showed the results of Southern blot, DNA sequencing and site-specific PCR, proving the site-specific insertion of P2A-ZsGreen as shown in Figure 1. Because of the site-specific recombination, professionally, only one funder line is required for the study and there are no additional off-target insertions.

How long is GSGreen half-life? Could a very long half-life be a major reason for the extremely large expression pattern observed?

Thanks for the question. The half life of ZsGreen, also called ZsGreen1, is at least 26 h in mammalian cells or slightly longer due to its tetrameric structure, in contrast with the monomeric configuration of other well-known fluorescent proteins (PMID: 17510373) 4. The rationale for using this GFP protein is that ZsGreen is an exceptionally bright green fluorescent protein, which is up to 4X brighter than EGFP—and is ideally suited for whole-cell labelling, promoter-reporter studies, considering of the higher turnover and rapid degradation of Fshr transcript. In this study, we used ZsGreen as a monitor or an indicator of the active Fshr endogenous promoter, rather than a means for measuring the promoter activity. Therefore, regardless of its accumulation or not, ZsGreen driven by Fshr promoter, indicates the presence of active Fshr promoter in the defined cells. In stead, we used ddRT-PCR to measure Fshr expression degrees in this study. In addition, we also provide single cell sequence-based evidence from 4 public single cell portables to support our findings of the wide Fshr expression. Please see Supplementary Data 3 in the revised MS.

References:

(1) Su J, Song Y, Yang Y, et al. Study on the changes of LHR, FSHR and AR with the development of testis cells in Hu sheep. *Anim Reprod Sci*. Sep 2023;256:107306. doi:10.1016/j.anireprosci.2023.107306

(2) Liu P, Ji Y, Yuen T, et al. Blocking FSH induces thermogenic adipose tissue and reduces body fat. *Nature*. Jun 1 2017;546(7656):107-112. doi:10.1038/nature22342

(3) Liu XM, Chan HC, Ding GL, et al. FSH regulates fat accumulation and redistribution in aging through the Galphai/Ca(2+)/CREB pathway. *Aging Cell*. Jun 2015;14(3):409-20. doi:10.1111/acel.12331

(4) Bell P, Vandenberghe LH, Wu D, Johnston J, Limberis M, Wilson JM. A comparative analysis of novel fluorescent proteins as reporters for gene transfer studies. *J Histochem Cytochem*. Sep 2007;55(9):931-9. doi:10.1369/jhc.7A7180.2007